# Analyzing the brainstem circuits for respiratory chemosensitivity in freely moving mice

**Amol Bhandare[1], Joseph van de Wiel[1], Reno Roberts[1], Ingke Braren[2], Robert Huckstepp[1], Nicholas Dale[1]\***

[1]School of Life Sciences, University of Warwick, Coventry, United Kingdom; [2]University Medical Center Eppendorf, Vector Facility, Institute of Experimental Pharmacology and Toxicology, Hamburg, Germany

**Abstract** Regulation of systemic $PCO_2$ is a life-preserving homeostatic mechanism. In the medulla oblongata, the retrotrapezoid nucleus (RTN) and rostral medullary Raphe are proposed as $CO_2$ chemosensory nuclei mediating adaptive respiratory changes. Hypercapnia also induces active expiration, an adaptive change thought to be controlled by the lateral parafacial region ($pF_L$). Here, we use GCaMP6 expression and head-mounted mini-microscopes to image $Ca^{2+}$ activity in these nuclei in awake adult mice during hypercapnia. Activity in the $pF_L$ supports its role as a homogenous neuronal population that drives active expiration. Our data show that chemosensory responses in the RTN and Raphe differ in their temporal characteristics and sensitivity to $CO_2$, raising the possibility these nuclei act in a coordinated way to generate adaptive ventilatory responses to hypercapnia. Our analysis revises the understanding of chemosensory control in awake adult mouse and paves the way to understanding how breathing is coordinated with complex non-ventilatory behaviours.

**\*For correspondence:**
n.e.dale@warwick.ac.uk

**Competing interest:** The authors declare that no competing interests exist.

## Editor's evaluation

This paper presents a novel application of endoscopic imaging with miniscopes in awake-behaving mice to address the important problem of analyzing multicellular activity by calcium activity imaging within circumscribed regions of the medulla oblongata that are proposed to have chemosensory functions for the homeostatic regulation of breathing in response to elevated systemic CO2 (hypercapnia). The authors importantly demonstrate chemosensory responses of neurons in the retrotrapezoid nucleus-RTN, Raphe magnus and pallidus nuclei, and lateral parafacial region- pFL, and they describe regional heterogeneity of cellular responses to hypercapnia in these regions with important functional implications. Analyzing chemosensory properties of these medullary regions has been the focus of numerous studies, but the problem of analyzing regional multicellular chemosensory responses in the awake freely behaving rodent has not been previously addressed, so this paper represents an advance for the field. The authors present a novel catalog of neuronal responses, and they illustrate the feasibility of this imaging approach, paving the way for further development/application of this approach, while indicating its limitations.

## Introduction

The precise control of breathing is fundamental to the survival of all terrestrial vertebrates. Breathing fulfils two essential functions: provision of oxygen to support metabolism; and removal of the metabolic by-product, $CO_2$. Rapid and regulated removal of $CO_2$ is essential because its over-accumulation

in blood will result in death from the consequent drop in pH. Understanding of the complexity of the brain microcircuit regulating $CO_2$ and pH remains incomplete. Whilst the primacy of the ventral medulla surface (VMS) in central chemoreception was described almost 60 years ago, this region contains multiple chemosensory nuclei that are candidates to mediate adaptive control of breathing (*Nattie and Li, 1994*; *Richerson, 1995*). This has led to the idea that central chemosensitivity is mediated by a network of chemosensory nuclei distributed throughout the brainstem and extending as far as the limbic system (*Huckstepp and Dale, 2011b*). Furthermore, there are multiple phenotypes of neurons within single chemosensory nuclei (*Huckstepp et al., 2018*; *Iceman et al., 2014*; *Iceman and Harris, 2014*; *Johansen et al., 2015*), and even within previously well-defined subpopulations of neurons (*Shi et al., 2017*; *Stornetta et al., 2009*).

Recently, brain imaging techniques have been developed to allow recording of activity of defined cell populations in awake, freely moving animals even in deep brain structures such as the hypothalamus (*Ziv and Ghosh, 2015*). These methods require: the expression of genetically encoded $Ca^{2+}$ indicators such as GCaMP6 in the relevant neurons; the implantation of gradient refractive index (GRIN) lenses at the correct stereotaxic position; and a head-mounted mini-epifluorescence microscope to enable image acquisition during the free behaviour of the mouse. We have now adapted these methods to enable recording of defined neuronal populations in the rostral medulla oblongata of mice to analyze the chemosensory control of breathing.

In this paper, we study the activity of neurons in the retrotrapezoid nucleus (RTN) (*Mulkey et al., 2004*; *Nattie and Li, 1994*; *Ramanantsoa et al., 2011*) and the rostral medullary raphe (*Bradley et al., 2002*; *Ray et al., 2011*; *Richerson, 1995*). Evidence strongly favours their involvement in the central chemosensory response, but thorough understanding as to how these nuclei contribute to central chemosensitivity has been hampered by the inability to record the activity of their constituent cells in awake freely behaving adult rodents. Instead, neuronal recordings have been made from young (<14 d) or adult rodents under anaesthesia. Both of these methods have significant drawbacks -the chemosensory control of breathing matures postnatally; and anaesthesia is known to depress the activity of respiratory neurons and reconfigure the circuit.

We also assess the contribution of the lateral parafacial region (pF$_L$) to an often-overlooked aspect of the hypercapnic ventilatory response, active expiration. During resting eupneic breathing, there is little active expiration instead the expiratory step involves elastic rebound of the respiratory muscles to push air out of the lungs. However, when the intensity of breathing is increased for example during exercise or hypercapnia, active expiration (recruitment of abdominal muscles to push air out of the lungs) occurs. Mounting evidence suggests the pF$_L$ may contain the expiratory oscillator and that neurons in this nucleus are recruited to evoke active expiration (*Huckstepp et al., 2015*; *Huckstepp et al., 2016*; *Pagliardini et al., 2011*). Nevertheless, the key step of recording activity of these neurons in awake behaving animals and linking it to active expiration has not been achieved.

Our data show the pF$_L$ did not undergo sustained activation during hypercapnia, but instead contributed to acute transient high amplitude expiratory events. We found that neurons in the RTN and Raphe exhibited a range of responses to hypercapnia. Many RTN neurons exhibited fast adapting responses. This response type was less common in the Raphe, and many Raphe neurons exhibited a slower graded response. Our data are consistent with a tiered chemosensory network, with the RTN and Raphe responsible for detecting different aspects of the hypercapnic stimulus. These novel findings illuminate the fundamental functional components of chemosensory nuclei and show that neuronal responses to hypercapnia are considerably more complex than anticipated.

## Results
### Optical characteristics of the recording system
The GRIN lens has a diameter of 600 µm with a focal plane ~300 µm below the lens (*Jennings et al., 2015*; *Resendez et al., 2016*). However, the focal plane varies depending upon the distance between the camera and GRIN lens, which can be altered via a manual turret adjustment to optimise the focus on fluorescent cells. To document the range of focal plane depth in our system, we imaged fluorescent beads entrapped in agarose under two conditions: the turret at its lowest (i.e. camera is against the end of the GRIN lens, and the focal plane is at its closest to the lens) and its maximum setting (i.e. the camera is as far from the end of the GRIN lens as it can be, and the focal plane is at its deepest

relative to the lens) (*Figure 1—figure supplement 1*). The focal depth (in which the beads were sharply focussed) was 100 µm at any single setting and the full focal plane ranged from 150 to 450 µm below the end of the lens as the turret moved from its lowest to highest setting. The turret setting is given in all figure legends.

## Inclusion/exclusion criteria for cells in the study

Following assessment for inclusion/exclusion criteria, data from 18 mice comprising recordings from 194 cells were included for analysis (*Figure 1A*). As the mice were unrestrained and able to move freely during the recordings, to visualize physical movements during $Ca^{2+}$ analysis high-definition videos of mice were synchronized to the simultaneous plethysmograph and Inscopix $Ca^{2+}$ imaging recordings in Inscopix Data Processing Software and *Spike2*. Our recordings are performed at a single wavelength (*Figure 1B*). Therefore, it was essential to verify that any signals result from changes in $Ca^{2+}$ rather than from movement artefacts.

We examined the origins of the changes in fluorescence in three ways. Firstly, we transduced RTN neurons of three mice with AAV-syn-GFP and performed imaging during a hypercapnic challenge (*Figure 1C–D*, *Figure 1—video 1*). This allowed measurement of comparable levels of fluorescence to GCaMP6 to determine whether movement by itself, in the absence of any $Ca^{2+}$ reporter activity, could generate confounding fluorescent signals. Analysis of changes in fluorescence showed only small amplitude (often negative-going) fluctuations. In no case did we observe large increases in fluorescence of the type we routinely observed when GCaMP6 was expressed (e.g. *Figure 1E*), suggesting that movement per se cannot give fluorescence signals that resemble $Ca^{2+}$ signals.

Secondly, we analyzed the movement of the mouse simultaneously with the $Ca^{2+}$ recordings (*Figure 1E*) via two methods. The first was to analyze the headmount movements in the video recordings of the mouse in the plethysmography chamber using *AnyMaze* tracking software and convert these movements into a colour coded sonogram. The brighter colours in the sonogram indicate headmount movement. The second method analyzed the large excursions on the whole body plethysmography (WBP) trace. These arise artefactually from movement. Comparison of the sonogram based on the WBP trace (WBP Sonogram) and that based on head movement (Mov Sonogram) showed that the two measures correlated well and that large excursions of the WBP trace often coincided with head movement (compare sonograms in *Figure 1E*). When we examined the GCaMP6 fluorescence in conjunction with the sonograms of head and body movement, we observed that: (i) large movements of the head/body did not evoke noticeable changes in GCaMP6 fluorescence (*Figure 1Ea*,b); (ii) GCaMP6 $Ca^{2+}$ signals characterized by a fast rise and exponential decay occurred in the absence of any movement (*Figure 1Ec*,f); and (iii) similar shaped $Ca^{2+}$ signals occurred during movement (*Figure 1Ed*,e). Together this suggests that movement of the head or body of the animal does not prevent the recording of genuine changes in GCaMP6 fluorescence related to changes in intracellular $Ca^{2+}$.

Thirdly, we used the same paradigm of stimulation for all experiments as can be seen from the records in *Figure 1—figure supplements 2–4*. This enabled averaging of the responses aligned either to the onset of hypercapnia, or features of the WBP trace. This averaging brought out consistent features of the presumed $Ca^{2+}$ signals that were temporally related to either the onset of hypercapnia or alterations of breathing. While for the most part we could not record from the same neurons in different recording sessions from the same mouse, we did observe consistent patterns of neuronal firing between recording sessions (*Figure 1—figure supplement 5*). As movement that could give rise to artefact would not be expected to be similar from recording session to recording session, or from mouse to mouse, the consistency of the patterns of the signals between mice and recording sessions strongly suggests a true biological rather than artefactual origin of the recorded signals.

In some cases, it was possible to identify the same neuron from recording sessions on different days in the same mouse. This was only the case in a minority of sessions perhaps because the brainstem is extracranial and more mobile and thus it is harder to reproduce the same exact focal plane between different recording sessions. When we were able to identify the same neurons their activity patterns were remarkably similar between separate recording sessions (*Figure 1—figure supplement 6*). This gives further confidence that the observed patterns are a reflection of biological properties.

Finally, we made recordings from anaesthetized mice (*Figure 1F* and *Figure 4—figure supplement 1*). In this case movement artefacts cannot be a potential confounding factor. To elicit neural activity,

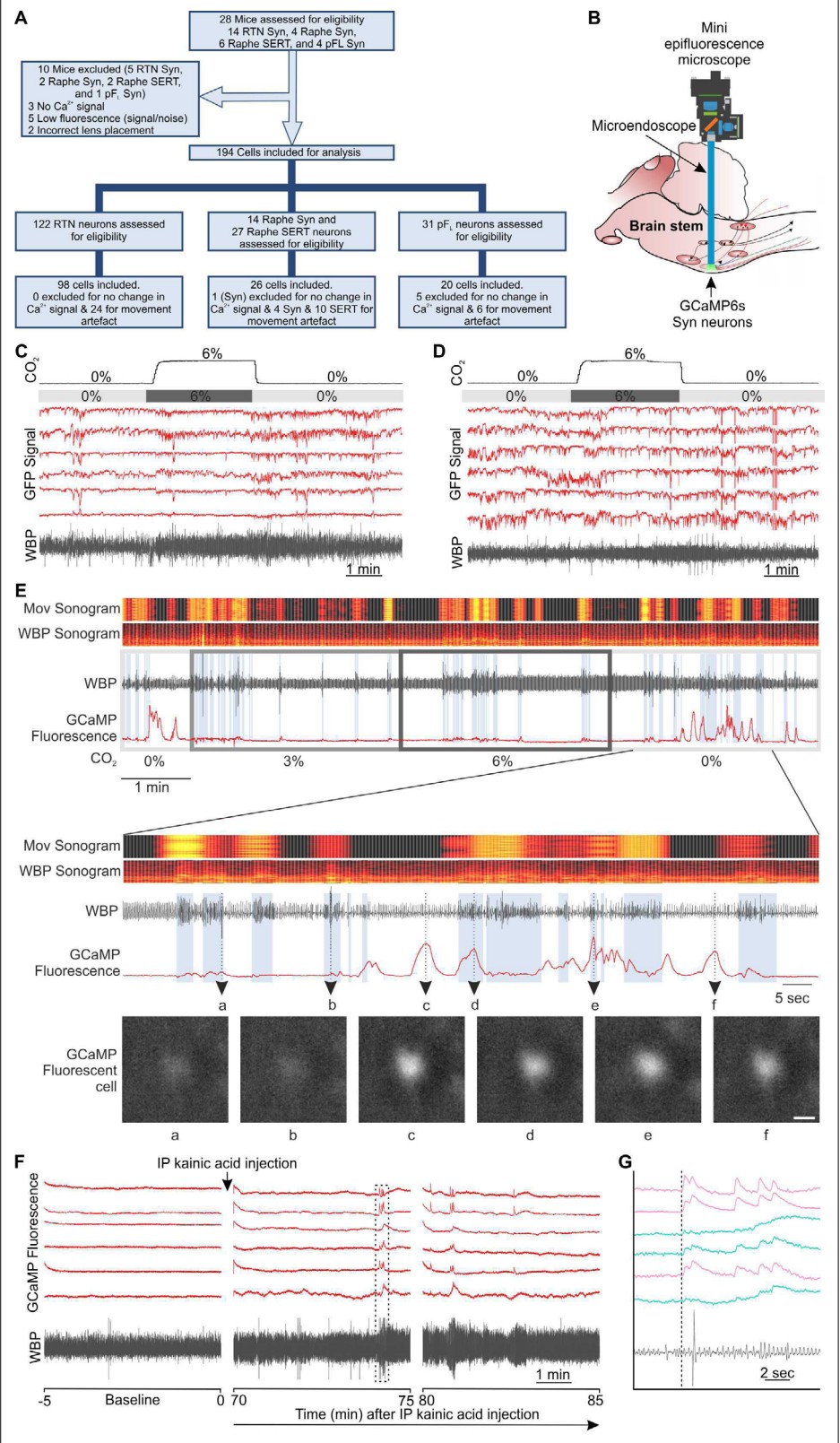

**Figure 1.** Experimental approach and movement artefact. (**A**) A CONSORT flow diagram for inclusion/exclusion of experiments and cells from the study. (**B**) Representation of GRIN lens (microendoscope), baseplate, and mini epifluorescence camera placement for recording of brainstem nuclei. (**C–D**) GFP control. RTN neurons were transduced with AAV-Syn-GFP (not Ca²⁺ sensitive) and GRIN lens implanted. Fluorescence was recorded going

*Figure 1 continued on next page*

*Figure 1 continued*

from 0 to 6% inspired $CO_2$ (gray bar). WBP: whole body plethysmography trace showing breathing movements. No signals are seen that resemble the $Ca^{2+}$ transients observed with GCaMP6. (**E**) Comparison of $Ca^{2+}$ fluorescence signals observed with GCaMP6 versus movement from WBP trace (WBP sonogram) and head movement as observed from video recording (Mov sonogram). In expanded traces- a,b, Examples of lack of fluorescence signal during movement of head/body; c,f, examples of fluorescence signal in the absence of movement; d,e, examples of fluorescence signals during movement. (Scalebar- 20 µm) (**F**) Seizures caused increase in activity of RTN neurons that was corelated with changes in breathing in anaesthetized mouse. Changes in mouse WBP before and after induction of non-behavioural seizures with intraperitoneal injection of kainic acid time matched with the activity of RTN neurons (red). The dotted square is changes in activity of neurons (ROIs) time matched with WBP and expanded in panel G. (**G**) Increase in the activity of RTN neurons correlated with changes in breathing. The start of the GCaMP6 transient is shown by the dotted line. $Ca^{2+}$ transients with a fast rise and exponential fall (pink traces); and slower sustained changes (turquoise traces).

The online version of this article includes the following video, source data, and figure supplement(s) for figure 1:

**Figure supplement 1.** The depth of focus of the GRIN lens.

**Figure supplement 2.** All RTN neurons responses to the hypercapnic challenge.

**Figure supplement 3.** All raphe neurons responses to the hypercapnic challenge.

**Figure supplement 4.** All $pF_L$ neuronal responses to the hypercapnic challenge (medium grey-6% $CO_2$, dark grey-9% $CO_2$).

**Figure supplement 5.** Consistent responses to hypercapnia in RTN neurons were recorded between sessions from the same mouse.

**Figure supplement 6.** Activity patterns of individual RTN neurons remains very similar across recording sessions.

**Figure supplement 7.** Two component analysis of neuronal categorisation in Raphe and RTN nuclei.

**Figure supplement 7—source data 1.** Data for *Figure 1—figure supplement 7*.

**Figure 1—video 1.** RTN neuronal responses to hypercapnia in mice transfected with synapsin GFP control (without GCaMP).
https://elifesciences.org/articles/70671/figures#fig1video1

**Figure 1—video 2.** Mouse undergoing kainate induced electrographic seizures in absence of physical seizures.
https://elifesciences.org/articles/70671/figures#fig1video2

---

we induced electrographic seizures via intraperitoneal kainic acid injection in mice in the absence of any behavioural seizure movements such as partial (including forelimb or hindlimb) or whole body continuous clonic seizures (*Figure 1—video 2*). After ~70 min this resulted in perturbations of the WBP trace and correlated $Ca^{2+}$ activity in the RTN neurons, presumably denoting the invasion of seizures into this nucleus. Detailed examination revealed that the fluorescence changes were of two types: (i) transients with a fast rise and exponential fall; and (ii) slower sustained changes (*Figure 1G*). This is an important observation as it shows that we expect to see two types of genuine $Ca^{2+}$ signal in recordings from freely-moving mice: fast transients with an exponential decay phase, that may sum together and slower more gradual rises.

Movements of the visual field, evident during most recordings, were corrected for via the Inscopix motion correction software to allow ROI-based measurement of fluorescence to remain in register with the cells during the recording. Where movement artefacts persisted following motion correction, in some cases it was possible to draw additional ROIs over a high contrast area devoid of fluorescent cells, such as the border of a blood vessel. This allowed for a null signal to be subtracted from the GCaMP signal (Figure 7E) thus providing the neuronal $Ca^{2+}$ transients free of movement artefact. If there was too much uncompensated motion artefact, so that we could not clearly analyze $Ca^{2+}$ signals, we excluded these recordings from quantitative analysis (details given in *Figure 1A*). GCaMP6 signals were only accepted for categorization if the following criteria were met: (1) the features of the cell (e.g. soma, large processes) could be clearly seen; (2) they occurred in the absence of movement of the mouse or were unaffected by mouse movement; (3) fluorescence changed relative to the background; (4) the focal plane had remained constant as shown by other nonfluorescent landmarks (e.g. blood vessels). Applying these criteria left 144/194 cells eligible for study. All of the recordings that were included are shown in *Figure 1—figure supplements 2–4*. The activity of the cells fell into the following categories:

### Inhibited (I)

Displayed spontaneous $Ca^{2+}$ activity at rest, which was greatly reduced during both 3 and 6% inspired $CO_2$ and could exhibit rebound activity following the end of the hypercapnic episode.

### Excited - adapting ($E_A$)

Silent or with low level activity at rest and showed the greatest $Ca^{2+}$ activity in response to a change in 3% inspired $CO_2$. Following an initial burst of activity, they were either silent or displayed lower level activity throughout the remainder of the hypercapnic episode. These cells often exhibited rebound activity following the end of the hypercapnic episode due to suppression by higher $CO_2$ levels. These cells essentially encoded the beginning and end of hypercapnia.

### Excited - graded ($E_G$)

Silent or with low level activity at rest and displayed an increase in $Ca^{2+}$ activity at 3% inspired $CO_2$ with a further increase in activity at 6% that returned to baseline upon removal of the stimulus. These cells encoded the level of inspired $CO_2$.

### Tonic (T)

Displaying spontaneous $Ca^{2+}$ activity throughout the recording that was unaffected by the hypercapnic episode but could provide tonic drive to the respiratory network.

### Sniff-coding (Sn)

Displayed elevated $Ca^{2+}$ signals correlated with exploratory sniffing.

### Expiratory (Exp)

Displayed elevated $Ca^{2+}$ signals correlated with large expiratory events.

### Non-coding (NC)

Displayed low frequency or sporadic $Ca^{2+}$ activity that was neither tonic in nature nor had any discernable activity that related to the hypercapnic stimulus or any respiratory event.

### Non-coding respiratory related (NC-RR)

Displayed $Ca^{2+}$ activity that did not code the hypercapnic stimulus, but it was instead related to variability in breathing frequency.

Our categorisation of neurons into these types was supported by a two-component analysis in which we plotted the change in $Ca^{2+}$ activity (from baseline) elicited by 6% inspired $CO_2$ versus the change in $Ca^{2+}$ activity elicited by 3% inspired $CO_2$. $E_A$ neurons as they showed more activity at 3% compared to 6% $CO_2$ should cluster below the line of identity (x=y), whereas $E_G$ neurons as they respond more strongly to 6% than 3% should cluster above the line of identity. $NC$ neurons as they did not show a response to hypercapnia should be clustered around the origin (x=y=0), and the $I$ neurons should be clustered in the quadrant where the change in $Ca^{2+}$ activity to both stimuli is negative. This analysis shows that the different classes of neurons in both the RTN and Raphe do indeed cluster in the appropriate regions of the graph (*Figure 1—figure supplement 7*).

## Chemosensory responses in the RTN

As the RTN contains chemosensitive glia and neurons (*Gourine et al., 2010*; *Mulkey et al., 2004*; *Nattie and Li, 1994*; *Ramanantsoa et al., 2011*), and is neuroanatomically diverse (*Shi et al., 2017*; *Stornetta et al., 2009*), we began by targeting all neurons of the RTN with a synapsin-GCaMP6s AAV (*Figure 2A–C*); a necessary step to enable their imaging during free behaviour via a mini-headmounted microscope (*Figure 1B*). The localization of successful and accurate transduction was confirmed posthoc by using choline acetyltransferase (ChAT) staining to define the facial nucleus and the location of the transduced cells (*Figure 2B*), with ~7% (9/132) of transduced neurons in the focal plane of the microscope co-labelled with ChAT (*Figure 2C*). The lens track terminated under the caudal pole of the facial nucleus, containing the highest number of neurons (*Figure 2B*), with

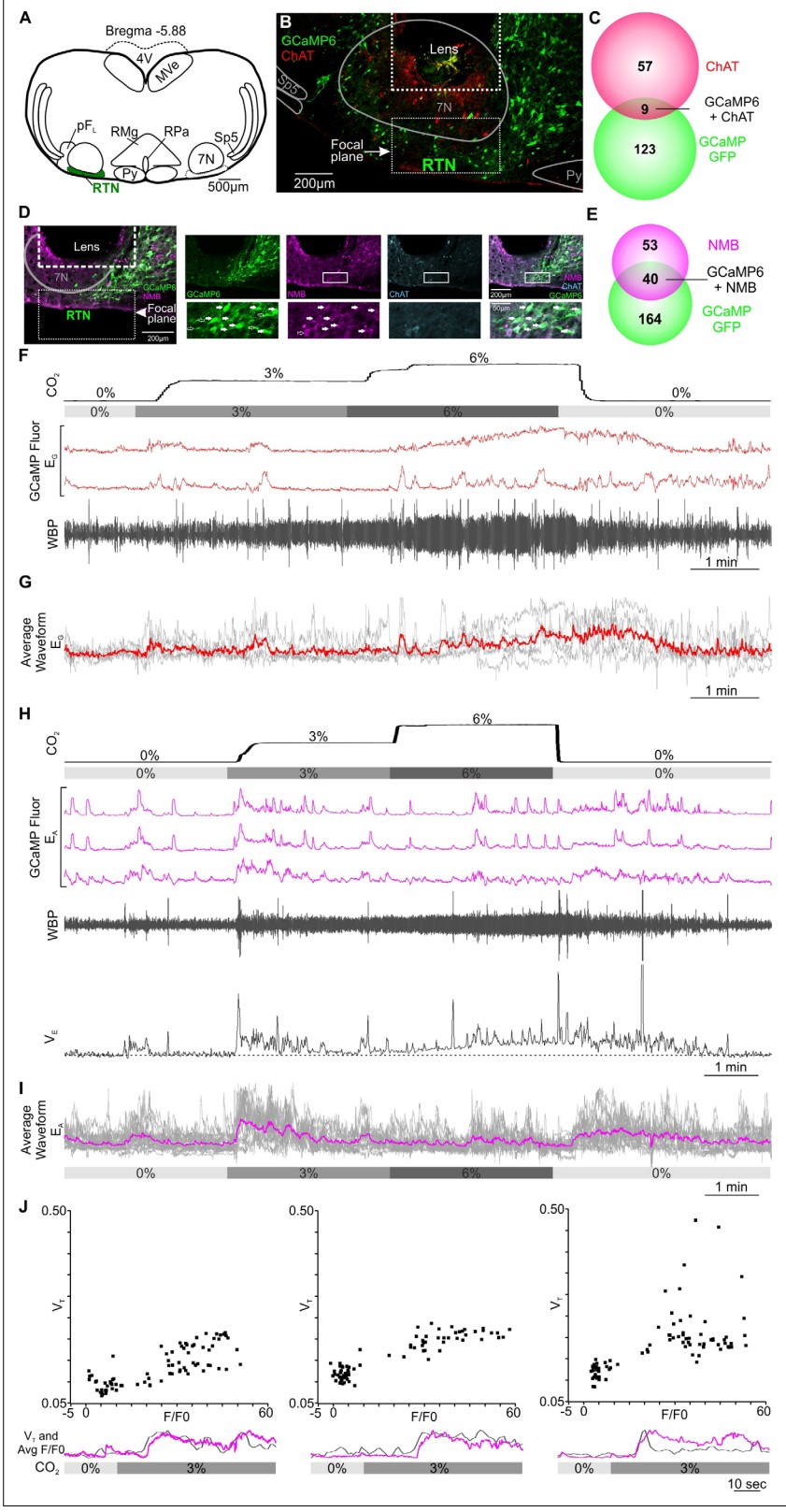

**Figure 2.** Excitatory chemosensory responses of RTN neurons in awake mice. (**A**) AAV9-Syn-GCaMP6s injection into the RTN. (**B**) Micrograph of lens placement and viral transduction of neurons (green) relative to the facial nucleus (ChAT +neurons red). (**C**) Venn diagram of cell counts of GCaMP6s transduced (green), and ChAT+ (red), neurons under the GRIN lens (1 representative section from each of 4 mice). (**D**) Neurons transduced

*Figure 2 continued on next page*

*Figure 2 continued*

with GCaMP6s also contain NMB confirming their identity as RTN neurons. (**E**) Venn diagram of cell counts of GCaMP6s transduced (green), and NMB+ (magenta), neurons under the GRIN lens (1 representative section from each of 3 mice). (**F**) Recording of mouse whole body plethysmography (WBP) in response to hypercapnia time-matched with Inscopix recorded GCaMP6 signals. These show neurons that gave a graded response to $CO_2$ ($E_G$). (**G**) Average of the graded neuronal responses aligned to the WBP trace in (F). (**H**) Examples of neurons that showed an adapting response to a change in inspired $CO_2$ ($E_A$). WBP trace aligned to the GCaMP6 fluorescence and instantaneous $V_E$ (minute ventilation) shown to demonstrate how the signals in the $E_A$ neurons closely correspond to $V_E$. (**I**) Average waveform of all $E_A$ neurons in the RTN aligned to 0, 3 and 6% inspired $CO_2$ (gray bar). (**J**) A plot of $V_T$ versus F/F0 for the $Ca^{2+}$ signal during the transition from 0 to 3% $CO_2$ in three individual mice showing positive correlation. Underneath each $V_T$ vs F/F0 graph is a plot of change in $V_T$ (grey) and average of $E_A$ neuronal $Ca^{2+}$ activity (magenta) from respective mouse during transition from baseline to 3% $CO_2$. Note that there is a transient increase in breathing at the beginning of hypercapnia that corresponds to the activation of the $E_A$ neurons. Abbreviations: 7 N, facial motor nucleus; Py, pyramidal tract; MVe, medial vestibular nucleus; sp5, spinal trigeminal nucleus: RTN, retrotrapezoid nucleus; RMg, raphe magnus; RPa, raphe pallidus; $pF_L$, parafacial lateral region.

The online version of this article includes the following video and figure supplement(s) for figure 2:

**Figure supplement 1.** Validation of NMB immunostaining patterns.

**Figure supplement 2.** The dynamics of individual $Ca^{2+}$ transients from two $E_A$ cells and two T cells recorded in the RTN before, during and after hypercapnia.

**Figure 2—video 1.** Excited graded ($E_G$) RTN neuronal response to hypercapnia in synapsin GCaMP6 transfected mice.

https://elifesciences.org/articles/70671/figures#fig2video1

**Figure 2—video 2.** Excited adapting ($E_A$) RTN neuronal response to hypercapnia in synapsin GCaMP6 transfected mice.

https://elifesciences.org/articles/70671/figures#fig2video2

---

the focal plane 150–450 µm below that, covering the dorso-ventral extent of the RTN (*Figure 1—figure supplement 1*). Therefore, our viral transduction and lens placement allowed us to record from the RTN. In some mice (n=3), we further checked the identity of transduced neurons by examining whether they expressed neuromedin B (NMB), a marker specific to Phox2b chemosensory neurons in the RTN (*Figure 2D*, *Figure 2—figure supplement 2*, *Li et al., 2016*; *Shi et al., 2017*). We found 43% (40/93) of NMB + cells in the RTN were also transduced with GCaMP6, and 20% (40/204) of GCaMP + neurons in the RTN were NMB+. Thus, we transduced nearly half of the NMB + neurons, and if the fluorescence imaging sampled transduced cells randomly and without bias to cell type, roughly 1 in 5 of cells recorded would have been of this phenotype.

After assessing the quality of recordings, we retained $Ca^{2+}$ activity traces from 98 RTN neurons in 9 mice (*Figure 1A*). There was a wide variety of neuronal responses to the hypercapnic challenge (*Figure 1—figure supplement 2*). While we observed 4/98 neurons exhibited a graded response to hypercapnia ($E_G$, *Figure 2F and G*, *Figure 2—video 1*), a much greater number (27/98) were of the adapting subtype ($E_A$, *Figure 2H1*, *Figure 2—video 2*). This adapting response may also be physiologically meaningful, as it matched the time course of changes in $V_E$ calculated from the WBP records (*Figure 2H*). In three mice the averaged $Ca^{2+}$ trace of the $E_A$ neurons in each mouse matched the changes in $V_T$ for the same mouse and a plot of $V_T$ versus $F/F_0$ for the $Ca^{2+}$ signal gave a positive correlation (*Figure 2J*, and *Supplementary file 1*). This correspondence between features of the responses in these neurons and the adaptive ventilatory response, supports the hypothesis that these are a physiologically important class of chemosensitive neurons. An alternative explanation that the rates of $Ca^{2+}$ sequestration greatly increased during hypercapnia such that $Ca^{2+}$ levels fell even although firing rates increased can be excluded by examining the dynamics of individual $Ca^{2+}$ transients. The rise time of these transients reflects the rate of $Ca^{2+}$ influx, and the exponentially falling decay phase the rate of $Ca^{2+}$ extrusion or sequestration. Comparison of these transients before, during and after the hypercapnic stimulus shows that these transients do not change in shape (*Figure 2H*, *Figure 2—figure supplement 2*).

A further 5/98 neurons were inhibited during hypercapnia (*Figure 3A and C*, blue traces and *Figure 3—video 1*). These neurons displayed spontaneous $Ca^{2+}$ activity during normocapnia, but

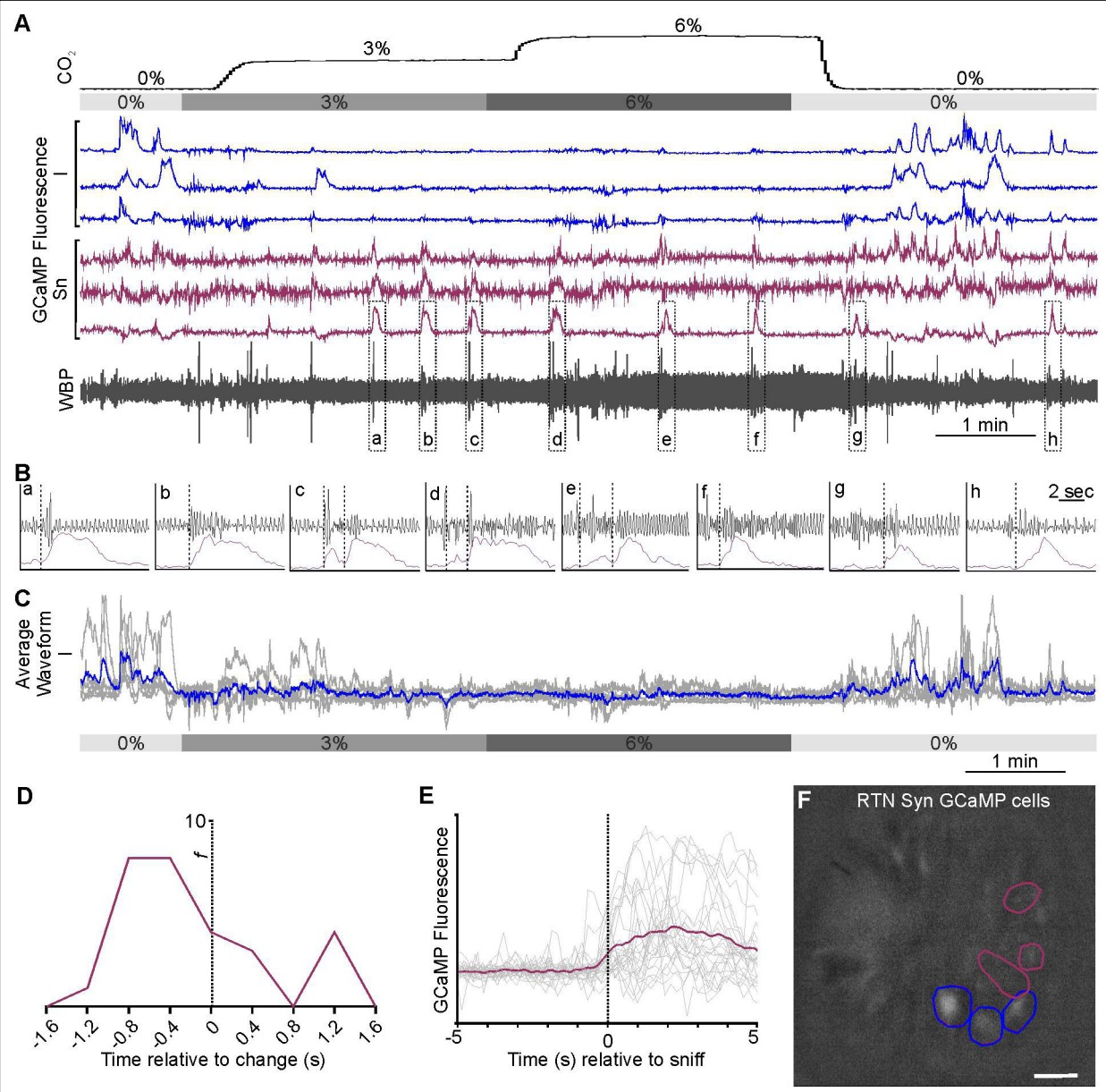

**Figure 3.** Sniff-coding, and CO₂-inhibited RTN neuronal responses in awake mice. (**A**) Recording of mouse WBP in response to hypercapnia time-matched with Inscopix recorded GCaMP6 signals. CO₂-Inhibited (blue) and sniff-coding (eggplant purple) calcium traces correspond to the neurons shown in panel F. (**B**) Expanded recordings from (**A**), the start of the Ca²⁺ transient is shown by the dotted line. (**C**) Average waveform of RTN inhibited neurons in response to hypercapnia. (**D**) Sniff-correlation histogram and (**E**) Spike triggered average (eggplant purple line) of all Ca²⁺ events (grey lines) temporally correlated to the beginning of sniff activity (dotted verticle line). (**F**) GCaMP6s fluorescence of transduced RTN neurons in freely behaving mice. Individual regions of interest (ROIs) drawn around CO₂-inhibited (blue) and sniff coactivated (eggplant purple) neurons. (Scalebar- 20 µm).

The online version of this article includes the following video and source data for figure 3:

**Source data 1.** Data for *Figure 3* panels D and E.

**Figure 3—video 1.** RTN sniff-coding (Sn) and inhibited (I) neuronal responses in mice transduced with synapsin GCaMP6.
https://elifesciences.org/articles/70671/figures#fig3video1

were abruptly silenced during hypercapnia (*Figure 3C*). We recorded from sniff-coding (Sn) neurons, 3/98, which showed elevated Ca²⁺ signals correlated with sniffing (*Figure 3A and B*). In sniff-coding neurons, Ca²⁺ activity coincided with perturbations of the plethysmography traces (increase in respiratory frequency, increase of tidal volume, presence of expiration, *Figure 3B* and *Figure 3—video 1*). Analysis of the onset of the Ca²⁺ signal relative to the start of the sniff, showed that, on average,

activity in these neurons preceded the sniff by 0.4–0.8 s (*Figure 3D*). The temporal correlation between the sniff-coding neurons and the sniff was further confirmed by averaging the $Ca^{2+}$ recordings from multiple sniffs (*Figure 3E*). Interestingly the different functional types of neuron are clustered adjacent to each other (*Figure 3F*).

Additionally, 5/98 neurons which were tonically active but did not encode any information about hypercapnia (T, *Figure 4A*, *Figure 4—video 1*). Tonically active neurons in the RTN could provide background excitation to the respiratory network. Of 98 neurons, 35 showed sporadic activity that did not correlate with $CO_2$ (NC, *Figure 4B*) and a further 19/98 neurons that did not encode $CO_2$ levels but displayed $Ca^{2+}$ signals that were correlated to changes in respiratory frequency and were generally silent when tidal volume was at its greatest (NC-RR, *Figure 4C*).

In summary for RTN neurons: ~40% (39/98) had activity patterns that were in some way modulated by $CO_2$. The majority of these neurons, 69% (27/39), displayed an adaptive response to $CO_2$, thus marking the onset of hypercapnia. Only a minority of neurons, 10% (4/39), encoded the magnitude of hypercapnia and a similar minority (13%; 5/39) were active and inhibited throughout the hypercapnic episode. Interestingly, we found a small number of sniff-activated neurons, which may be from the most rostral and ventral aspect of the retrofacial nucleus, which overlaps with the most caudal and dorsal aspect of the RTN (*Deschênes et al., 2016*; *Kurnikova et al., 2018*).

## Effect of anaesthesia on RTN chemosensory responses

Until now most investigation of RTN chemosensory responses at adult life stages has been in anaesthetised preparations. We therefore examined the effect of deep anaesthesia on the activity of RTN neurons before and during chemosensory stimuli (*Figure 4—figure supplement 1*). As might be expected, compared to the awake state, urethane anaesthesia had a deeply suppressive effect on the spontaneous activity of all RTN neurons. Whereas almost all neurons displayed spontaneous activity, there was hardly any activity under anaesthesia (*Figure 4—figure supplement 1A-B*). Furthermore, the responses to hypercapnia were greatly blunted, with many neurons simply being unresponsive. When we examined those neurons that retained a chemosensory response to hypercapnia in the anesthetised state, these responses differed from those that the same neurons showed in the awake state (*Figure 4—figure supplement 1C*). For example, cells displaying a graded response to hypercapnia under anaesthesia were classified as non-coding respiratory related (NC-RR) or non-coding (NC) in the awake state.

## Chemosensory responses in the medullary raphe

As the medullary Raphe contains multiple neuronal types which display functionally different $CO_2$ responses in vitro (*Bradley et al., 2002*; *Ray et al., 2011*; *Richerson, 1995*) and in vivo (*Veasey et al., 1995*), we next examined whether the activity of neurons in the rostral Raphe magnus and pallidus could be altered by $CO_2$ (*Figure 5*). We drove expression of GCaMP6s with a synapsin promoter (n=2 mice; *Figure 5A–C*). Assessing the ability of this construct to transduce serotonergic neurons in this nucleus, we found that 57/113 (50%) of the transduced neurons in the focal plane of the microscope were TPH+ (tryptophan hydroxylase; a marker of serotonergic neurons) (*Figure 5D*). As there are also GABAergic neurons (*Iceman et al., 2014*; *Iceman and Harris, 2014*), and non-serotonergic NK1R neurons in the Raphe (*Hennessy et al., 2017*; *Iceman and Harris, 2014*), we also specifically targeted the serotonergic neurons by driving GCaMP6 expression with a SERT (slc27a4) promoter (n=4 mice, *Figure 5E–F*). Post-hoc immunocytochemistry showed that all neurons in the optical pathway of the microscope (*Figure 5E*) which expressed GCaMP6s co-localised with TPH. We recorded 9 synapsin GCaMP6s neurons and 17 SERT GCaMP6s neurons.

As in the RTN, we found there were multiple categories of $CO_2$-dependent responses (*Figure 1—figure supplement 3*). A frequently observed class of neurons (8/26) exhibited a graded response to $CO_2$ ($E_G$, *Figure 5G and H* and *Figure 5—video 1*, synapsin). 4/26 neurons showed the adapting response ($E_A$, *Figure 5G*). The commonest class of neurons in our dataset was inhibited by $CO_2$ (10/26, all from mice transduced with GCaMP under the SERT promoter, *Figure 6A and B*, *Figure 6—video 1* SERT). The remaining 4/26 neurons displayed activity that was unaffected by hypercapnia (NC, *Figure 6C*).

In summary, 85% (22/26) of Raphe neurons had $Ca^{2+}$ activity patterns that were modulated by $CO_2$. Interestingly, unlike the RTN which showed considerable bias toward the $E_A$ functional type,

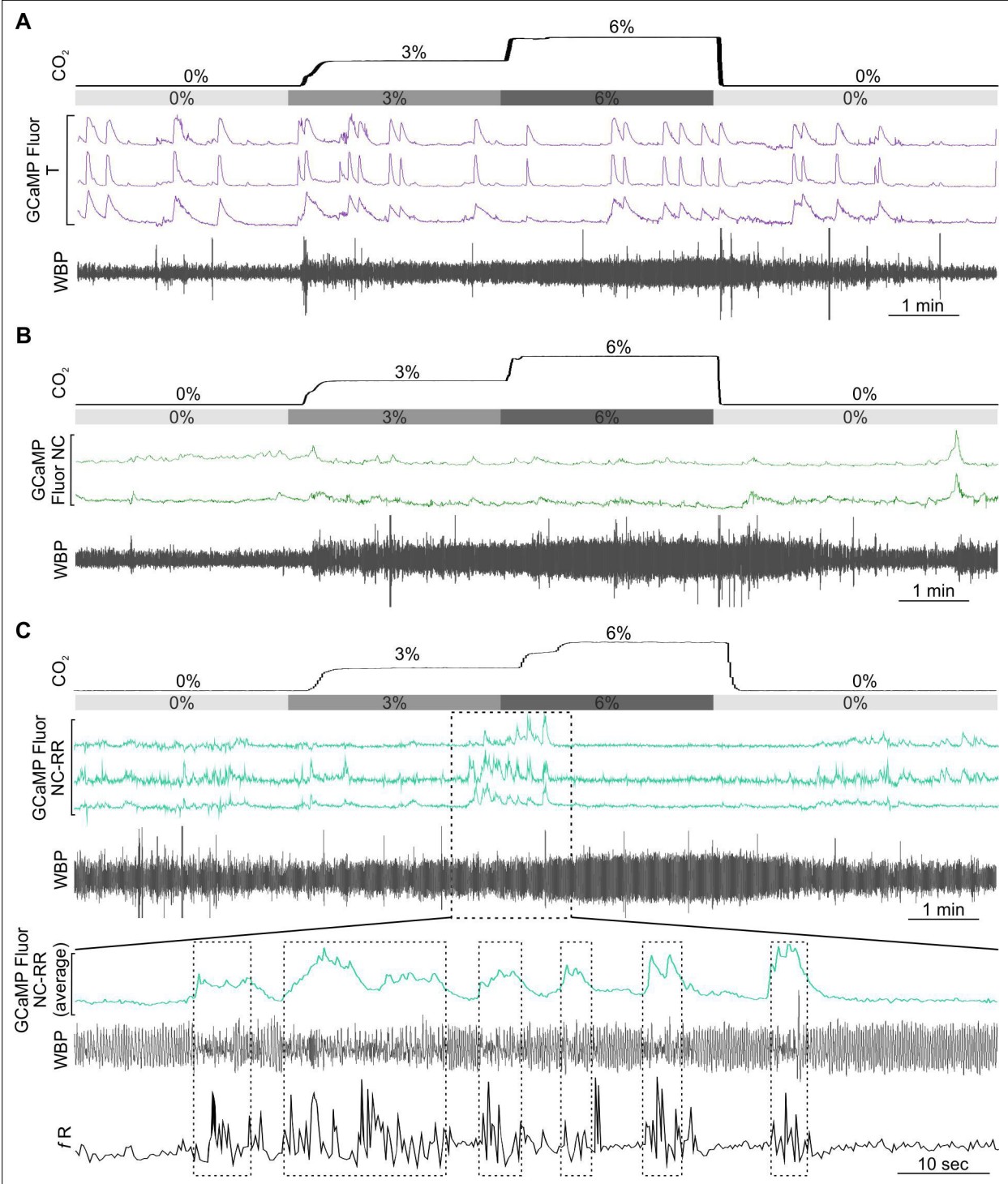

**Figure 4.** Tonically active, non-coding (NC) and NC-respiratory related (NC-RR) RTN neuronal responses in awake mice. Recording of mouse WBP in response to hypercapnia time-matched with Inscopix recorded GCaMP6 signals (**A, B, C**). GCaMP fluorescence of RTN (**A**) tonically active (T; violet blue) (**B**) non-coding (NC; green) and (**C**) non-coding respiratory related (NC-RR; shamrock green) neurons. Average of NC-RR neuronal activity (dotted box) expanded under it displayed $Ca^{2+}$ signals that were correlated to changes in respiratory frequency ($fR$) and were generally silent when tidal volume (WBP) was at its greatest.

The online version of this article includes the following video, source data, and figure supplement(s) for figure 4:

**Figure supplement 1.** Hypercapnia induced RTN neuronal responses in awake and anaesthesized mice.

*Figure 4 continued on next page*

*Figure 4 continued*

**Figure supplement 2.** Change in breathing frequency (fR), tidal volume (V_T) and minute ventilation (V_E) in response hypercapnia from mice recorded in the study.

**Figure supplement 2—source data 1.** Data for *Figure 4—figure supplement 2*.

**Figure 4—video 1.** Tonically active (T) RTN neurons (irrespective of hypercapnia) in synapsin GCaMP6 transfected mice.

https://elifesciences.org/articles/70671/figures#fig4video1

$E_A$ neurons were much less prevalent in the Raphe (only 15%), with $E_G$ (31%), and I (38%) being the most common types of response we observed. In the Raphe, there were no tonically active neurons or neurons that were co-active with any non-respiratory related orofacial movements such as sniffing.

## Expiratory activity of neurons in the pF$_L$

Neurons of the pF$_L$, lateral and adjacent to the RTN, have been proposed as the expiratory oscillator (*Huckstepp et al., 2015*; *Huckstepp et al., 2016*). Under resting conditions, eupneic breathing, with the exception of during REM sleep (*Andrews and Pagliardini, 2015*), does not involve active expiration, so the neurons of the pF$_L$ are largely silent. A hypercapnic stimulus causes an increase in respiratory frequency, tidal volume and also the onset of active expiration (*Huckstepp et al., 2015*). We therefore transduced neurons of the pF$_L$ in 3 mice with the synapsin-GCaMP6f construct (*Figure 7A–D*) and observed their responses during a hypercapnic stimulus (*Figure 7F*, *Figure 1—figure supplement 4*). We found that the $Ca^{2+}$ activity reflected changes in respiratory frequency and tidal volume with surprising fidelity (compare grey trace (fR) with GCaMP traces in *Figure 7F*). Closer inspection revealed the temporal relationship between $Ca^{2+}$ activity and changes in the plethysmographic traces (*Figure 7G*). Whilst $Ca^{2+}$ activity in the pF$_L$ neurons on average preceded an increase in tidal volume and respiratory frequency by ~0.4 s, some neurons showed $Ca^{2+}$ activity only after changes in tidal volume and respiratory frequency had occurred (*Figure 7H–J*). This time separation was independent of the level of inspired $CO_2$ (*Figure 7I and J*). These characteristics are more compatible with a correlative rather than a causal relationship between the activation of pF$_L$ neurons and changes in tidal volume and respiratory frequency.

When we specifically studied the incidence of active expiration, we found that $Ca^{2+}$ activity in pF$_L$ neurons always preceded a change in the expiratory activity (*Figure 7H–M*). Importantly, the time delay between activity in the pF$_L$ neurons and active expiration was sensitive to inspired $CO_2$ (*Figure 7K–M*). At 6% inspired $CO_2$ this time difference was ~0.5 s. (*Figure 7K and L*), while at 9% inspired $CO_2$ the time interval shortened to ~0.3 s (*Figure 7K and M*). In all 20 neurons in the pF$_L$ that exhibited activity, this activity preceded active expiration. This timing is consistent with a causal rather than a correlative link between neuronal activity in the pF$_L$ and induction of active expiration. Post-hoc immunocytochemistry showed that a small population of neurons which expressed GCaMP6 co-localised with ChAT (2/71, 1.5%, *Figure 7B and D*) and were located in the focal plane of the microscope (*Figure 7B*). These neurons either displayed expiratory-linked activity or more likely were silent and would have been excluded from our dataset, given no other responses other than expiratory activity were recorded in this region.

## Discussion

### Limitations

We have recorded the intracellular $Ca^{2+}$ signal from neurons present in the RTN and Raphe. The summation and temporal variation of the signal will depend on the $Ca^{2+}$ buffering/extrusion systems within the cell, the density and activation properties of $Ca^{2+}$ permeable channels, and whether $Ca^{2+}$ may additionally be released from intracellular stores. Therefore, this $Ca^{2+}$ signal cannot tell us the precise firing rates or dynamics of the neuronal activity, and so is an imprecise proxy of neuronal activity. Despite this imperfection, $Ca^{2+}$ activity of neurons is a widely used to assess their electrical activity (*Chen et al., 2013*; *Dana et al., 2019*; *Huang et al., 2021*). We verified that the rates of $Ca^{2+}$ influx and extrusion or sequestration do not change with the hypercapnic stimulus by examining individual $Ca^{2+}$ transients before, during and after hypercapnia to show that their rise times and decay

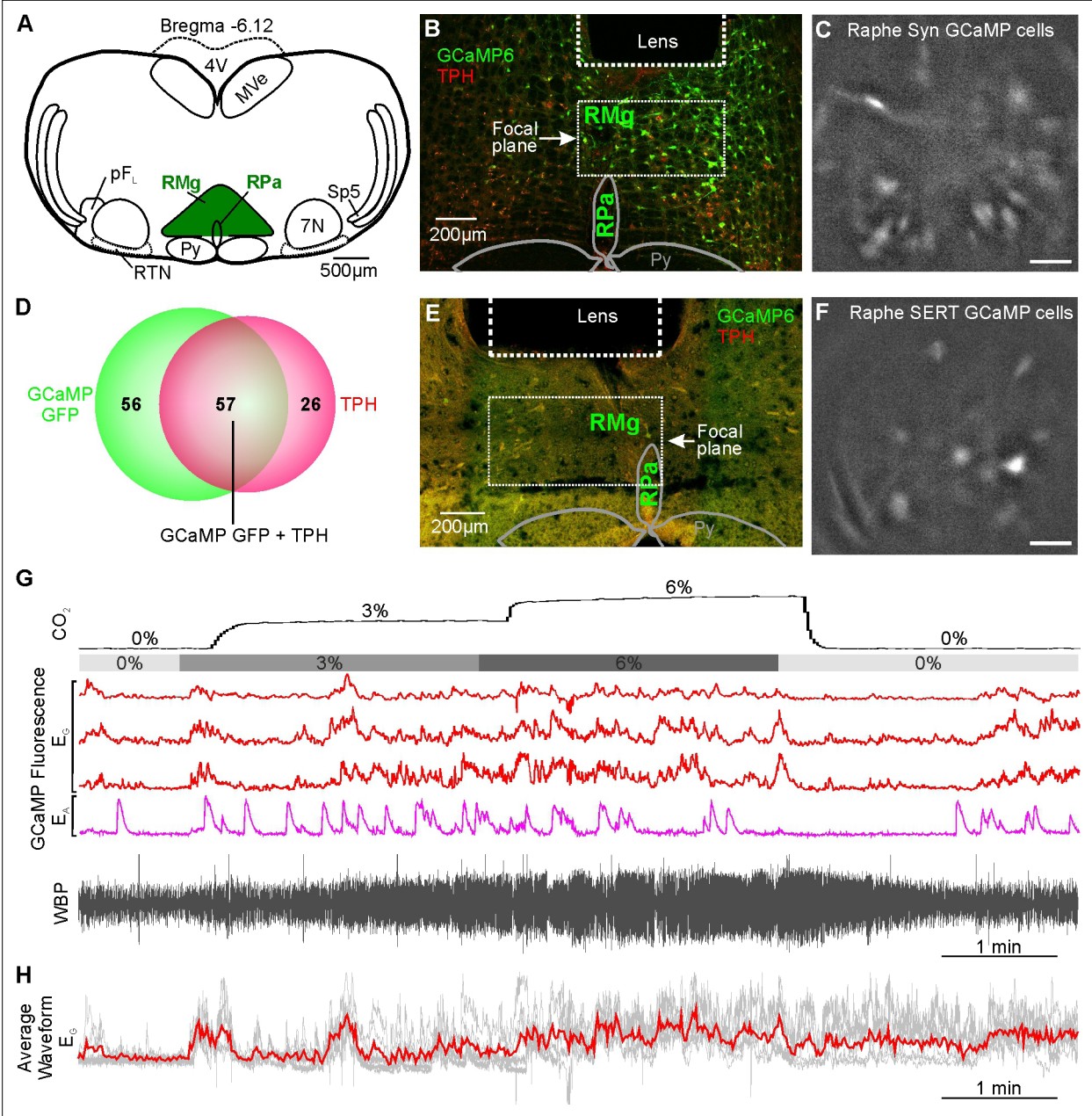

**Figure 5.** $CO_2$-excitated medullary raphe neurons in awake mice. (**A**) AAV injection into the Raphe. (**B**) Micrograph of lens placement and AAV9-Syn-GCaMP6s viral transduction of neurons (green) relative to the Raphe (TPH +neurons red). (**C**) GCaMP6s fluorescence signal from AAV9-Syn-GCaMP6s transduced raphe neurons in freely behaving mice (Scale bar- 50 µm). (**D**) Venn diagram of cell counts of AAV9-Syn-GCaMP6s transduced neurons (green) under the GRIN lens with TPH +neurons (red) in the raphe (1 representative section from each of 4 mice). (**E**) Micrograph of lens placement and AAV9-SERT-GCaMP6s viral transduction of neurons (green) relative to the Raphe (TPH +neurons red). (**F**) GCaMP6s fluorescence signal from AAV9-SERT-GCaMP6s transduced raphe neurons in freely behaving mice (Scale bar- 50 µm). (**G**) WBP recording in response to hypercapnia time-matched with Inscopix recorded GCaMP6 signals. GCaMP fluorescence of raphe excitatory graded ($E_G$), and excitatory adapting ($E_A$) neurons in response to hypercapnia. (**H–I**) Average waveform of raphe $E_G$ neuronal responses to hypercapnia aligned to the WBP trace in G. Abbreviations defined in *Figure 2*.

The online version of this article includes the following video for figure 5:

**Figure 5—video 1.** Excited graded ($E_G$) and adapting ($E_A$) Raphe neuronal responses to hypercapnia in synapsin GCaMP6 transfected mice.
https://elifesciences.org/articles/70671/figures#fig5video1

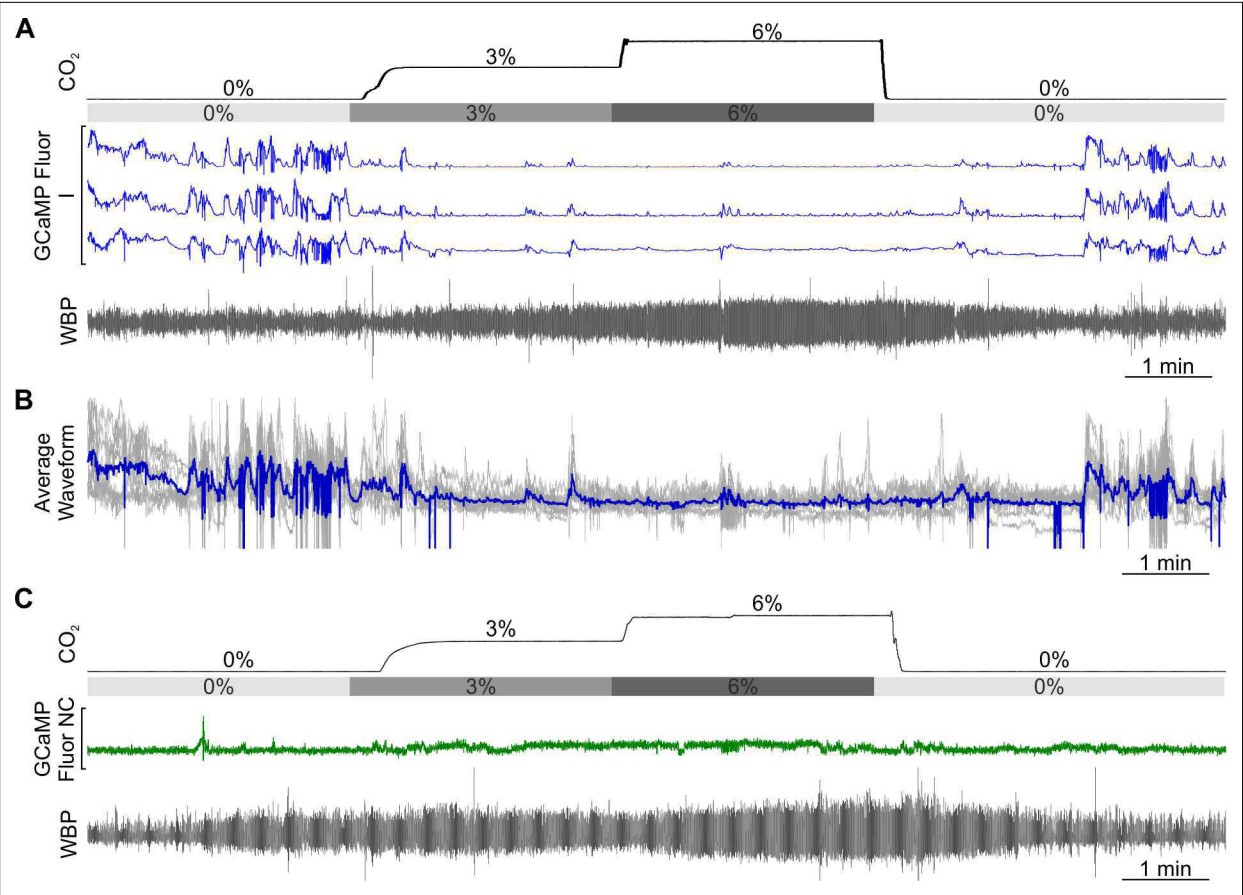

**Figure 6.** $CO_2$-inhibited and non-coding raphe neuronal responses in awake mice. Recording of mouse WBP in response to hypercapnia time-matched with Inscopix recorded GCaMP6 signals. GCaMP fluorescence of raphe $CO_2$ inhibited (I; blue) (**A**) and non-coding (NC; green) (**C**) neurons. (**B**) Average waveform of raphe inhibited neurons in response to hypercapnia aligned to the WBP trace in A.

The online version of this article includes the following video for figure 6:

**Figure 6—video 1.** Inhibited Raphe neurons due to hypercapnia in SERT GCaMP6 transfected mice.

https://elifesciences.org/articles/70671/figures#fig6video1

rates remain unaffected (*Figure 2—figure supplement 2*). This gives further confidence that the pattern of $Ca^{2+}$ activity reflects the pattern of neuronal firing.

We used GCaMP6s for recording from the RTN and Raphe as this is the most sensitive $Ca^{2+}$ sensor and we did not expect to need to resolve rapidly changing $Ca^{2+}$ signals. For the $pF_L$ we used faster responding, but less sensitive, GCaMP6f to attempt to document cycle by cycle changes in fluorescence, but these were not resolved in the current experiments.

We made our recordings at a single wavelength as this was the capability of the minimicroscopes available at the time of the study. Dual channel versions are now available and these would assist with the treatment and exclusion of movement artefacts e.g. through use of ratiometric imaging at two wavelengths. Nevertheless, our extensive controls for movement artefacts, and data analysis, show that we have resolved real $Ca^{2+}$ signals that reflect different patterns of neuronal firing that are reproducible between across the imaged neuronal population in different recording sessions from a single mouse and recordings between multiple mice. These signals are characterised by the well-documented pattern for true $Ca^{2+}$ signals of a fast onset and a slower exponentially-falling offset.

While we can be confident of the location of the GRIN lens and the volume of tissue in which labelled neurons could have contributed to the observed signals from posthoc analysis, it is important to note that we cannot identify the actual neurons that were recorded in each session. Related to this, we were unable, for the most part, to make consecutive recordings from the same neurons from session to session. We used a genetic targeting strategy that would give broad coverage of all

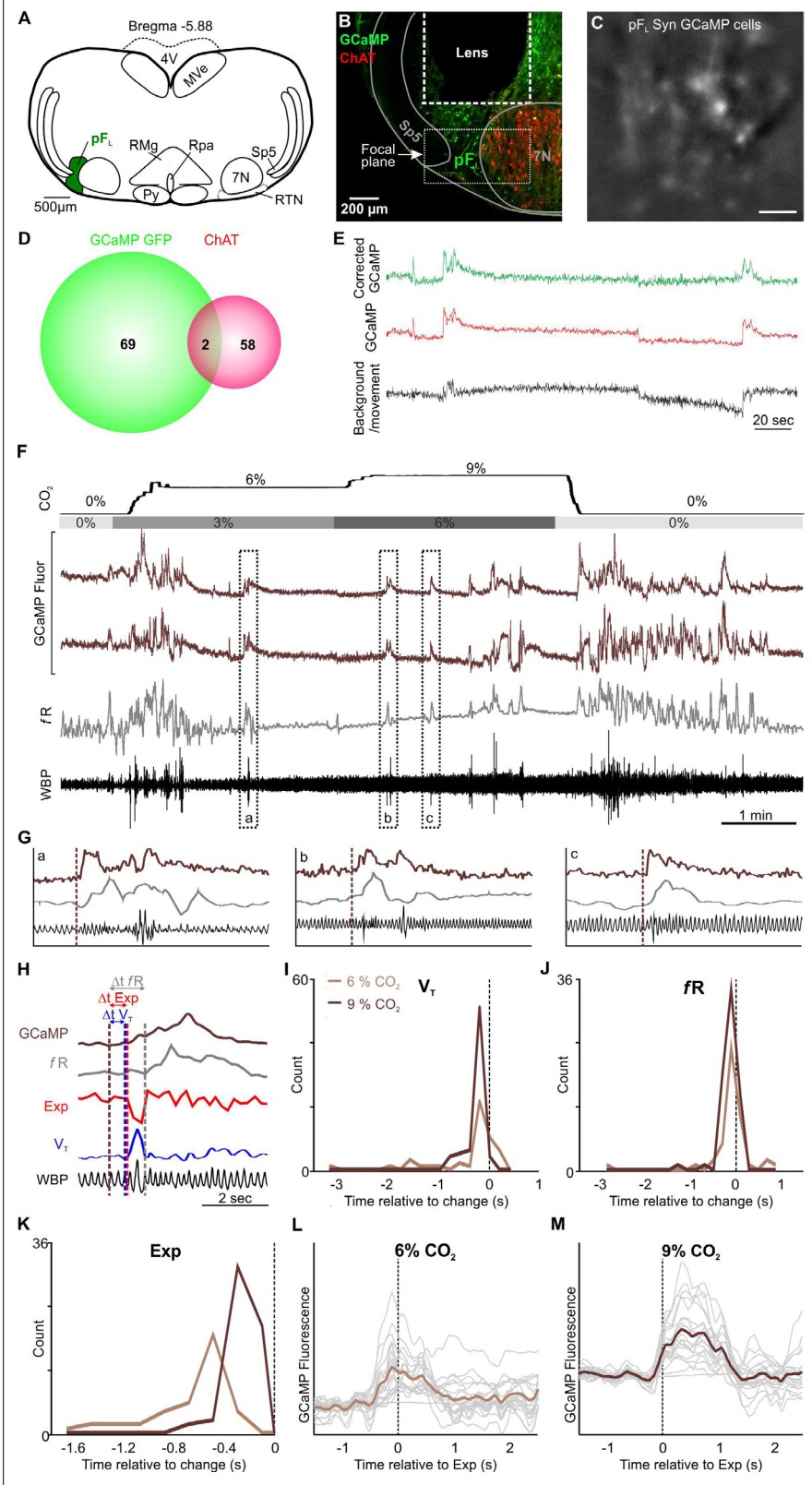

**Figure 7.** pF_L neurons drive active expiration. (**A**) AAV9-Syn-GCaMP6f injection into the pF_L. (**B**) Micrograph of lens placement and viral transduction of pF_L neurons (green) relative to the facial nucleus (ChAT +neurons red). (**C**) GCaMP6f fluorescence signal from transduced pF_L neurons in freely behaving mice (Scale bar- 50 µm). (**D**) Venn diagram of cell counts of GCaMP6f transduced (green) and ChAT+ (red) neurons under the GRIN lens (1

*Figure 7 continued on next page*

*Figure 7 continued*

representative section from each of 3 mice). (**E**) Movement artefact subtracted from GCaMP signal extracts a clear $Ca^{2+}$ transient from $pF_L$ neurons. An ROI was placed over the border of a blood vessel (background/movement; black) and the subsequent fluoresence recording was subtracted from the GCaMP signal (uncorrected GCaMP; red) giving rise to neuronal $Ca^{2+}$ transients of $pF_L$ neurons (corrected GCaMP; green) in awake mice that are free from the movement artefact. (**F**) WBP recording in response to hypercapnia time-matched with Inscopix recorded GCaMP signals. GCaMP6f fluorescence of transient expiratory (Exp; dark brown) $pF_L$ neurons. (**G**) Expanded traces from (**F**), the start of the $Ca^{2+}$ transient is shown by the brown dotted line. (**H**) Measurements of GCaMP6 fluorescence (dark brown) relative to tidal volume ($V_T$, blue), respiratory frequency ($fR$, grey), and expiration (Exp, red). Verticle dotted lines represent the start of the changes on their respective channels. (**I–K**) Frequency histograms of timing of changes in GCaMP fluorescence of $pF_L$ neurons relative to tidal volume ($V_T$, left), respiratory frequency ($fR$, centre), and expiration (Exp, right). (**L–M**) Spike triggered average (brown line) of all $Ca^{2+}$ events (grey lines) temporally correlated to the beginning of expiratory activity (dotted verticle line) at (**L**) 6% $CO_2$ and (**M**) 9% $CO_2$. Abbreviations defined in *Figure 2*.

The online version of this article includes the following source data for figure 7:

**Source data 1.** Data for *Figure 7* panels I-K.

neuronal types within the nuclei. This presents an overview of the different activity patterns that occur during responses to hypercapnia, however it does not allow identification of the responses of specific neuronal subtypes. Now that we have a library of all activity patterns in these regions, more precise genetic targeting of neurons is required to relate phenotype to firing pattern. Given that no single unique genetic marker has yet been identified to mark chemosensory neurons, such an approach would have to utilize intersectional genetics to achieve the necessary level of precision.

Our observations document the different types of neuronal responses to hypercapnia in awake mice. However, we cannot be sure whether these responses were a direct consequence of the chemosensory stimulus, i.e. a direct action of $CO_2$ or pH on the recorded neurons, or an indirect secondary outcome shaped by the synaptic networks present within the medulla. Additionally, recordings from the same neurons in response to repeated periods of hypercapnia would give further assurance that the activity patterns observed are reproducible responses to hypercapnia at a single cell as opposed to population level.

## Chemosensory activity in the rostral medulla of adult mice

For many years the RTN was considered to be a homogeneous population of chemosensory neurons (*Guyenet et al., 2010*; *Nattie and Li, 1994*). However, recent neuroanatomical (*Li et al., 2016*; *Shi et al., 2017*; *Stornetta et al., 2009*) and pharmacological (*Huckstepp et al., 2018*; *Li et al., 2016*) evidence suggests the region may be more heterogenous than first thought and that functional subdivisions of the RTN may exist in adult rodents. In accordance with this, we found 7 functional subpopulations of neurons. There were two classes of neurons that were excited by $CO_2$ and indicated an aspect of the hypercapnic stimulus (*Figure 2*): $E_A$ (the start and end of hypercapnia), and $E_G$ (the presence and magnitude of the hypercapnic stimulus). In addition, there were neurons that were inhibited by hypercapnia (I). Inhibition of RTN neuronal firing by hypercapnia has been previously observed (*Cleary et al., 2021*; *Nattie et al., 1993*; *Ott et al., 2011*).

We found the majority of neurons (from five different mice) in our recordings from the RTN were of the $E_A$ subtype that is they responded to the initial increase in inspired $CO_2$ but did not maintain their activation. In some of these neurons there could be a further small increase in activity at the transition of inspired $CO_2$ from 3 to 6%. This suggests that in these neurons some aspect of the sensory response adapts or fatigues, or they are subject to delayed $CO_2$-dependent feedback inhibition that depresses their activity. It is notable that in many of these neurons, activity increased following removal of the hypercapnic stimulus. This activation during transitions at the beginning and end of a stimulus is reminiscent of a multitude of rapidly adapting sensory neurons (e.g. rapidly adapting pulmonary receptors *Widdicombe, 1954*, Pacinian and Meissner corpuscles *Vallbo and Johansson, 1984*). This could be due to rapid removal of sensory adaptation followed reactivation and rebound activity as the arterial $CO_2$ levels moves back to resting levels.

While adapting responses to hypercapnia have not been recognised before, the Phox2b+neurons of the RTN have been subdivided into types 1 and 2 on the basis of their pH sensitivity (*Lazarenko*

*et al., 2009*). Examination of the response characteristics of the example type 1 neuron illustrated in this paper shows that it displayed an adapting response to acidification, taking 30–40 s to decline from peak firing to a steady state baseline (*Lazarenko et al., 2009*). By contrast the example type 2 neuron shows a sustained graded response (*Lazarenko et al., 2009*), similar to that reported for a Phox2B+neuron in an earlier publication (*Stornetta et al., 2006*). A similar subdivision has been made for acutely isolated Phox2b+neurons. In vitro, type 1 neurons also appear to display an adapting response to acidification suggesting that it may be an intrinsic property (*Wang et al., 2013*). We tentatively suggest that the adapting neurons ($E_A$) may therefore correspond to type 1 Phox2b+neurons, however this would require further direct experimental evidence to substantiate this point.

We found only 4 neurons (from 3 different mice) that exhibited a graded response to $CO_2$ ($E_G$). Although these neurons were in the minority, their graded responses to hypercapnia were very similar in time course to those previously described for Phox2b+ RTN neurons in the anaesthetised adult rat (*Stornetta et al., 2006*) and they might correspond to the type 2 Phox2b+ neurons (*Lazarenko et al., 2009*). The comparative rarity of the $E_G$ neurons was not due to weak responses to hypercapnia which were very robust in the awake mice (note the raw whole body plethysmography [WBP] traces in *Figures 2–4* and *Figure 4—figure supplement 2A*). Nor was it due to a technical issue that prevented us from seeing neural activity, as (1) the lens placement was sufficient to image the most superficial aspect of the RTN where the Phox2b+ neurons lie (*Mulkey et al., 2004*; *Shi et al., 2017*; *Stornetta et al., 2009*), and (2) induction of a seizure in anesthetised mice gave clear $Ca^{2+}$ activity in the RTN neurons when seizures occurred and altered respiratory activity (*Figure 1F and G*). The possibility that the serotype of the AAV (AAV-9) did not transduce the chemosensory neurons of the RTN seems remote, as others have demonstrated its efficacy for these neurons (*Hérent et al., 2021*). Although we recorded from 98 RTN neurons in 9 mice, we cannot completely exclude that some unknown aspect of the recording set up may have prevented us from seeing all types of neurons and might have led to comparative under-representation of the $E_G$ subtype in our dataset.

The respiratory network, and with it the hypercapnic ventilatory response, changes with development (*Huckstepp and Dale, 2011b*). Chemosensory responses become less dependent on Phox2B+ neurons of the RTN by 3 months of age when those neurons are genetically ablated (*Ramanantsoa et al., 2011*). Acute chemical lesions of the RTN that remove almost all NMB+ neurons, greatly perturbed central chemosensory responses at adult stages, however this was not seen with smaller lesions that preserved a little under half of the NMB+ neurons (*Souza et al., 2018*). A lesioning strategy, be it chemical or genetic, cannot discriminate between a direct chemosensory function and a relay function of this nucleus. The majority of evidence for the responses of RTN neurons, and in particular the Phox2B+ neurons, to hypercapnic stimuli comes from neonatal or young juvenile animals (*Mulkey et al., 2004*; *Ramanantsoa et al., 2011*). Here we are investigating chemosensory mechanism in the adult after the system has fully matured. It is possible that the nature of the chemosensory responses change, and the $E_A$ neuronal phenotype emerges at adult stages. It is notable that even in the cells classified as $E_G$ there is a transient enhancement of activity immediately following the imposition of hypercapnia (*Figure 2F and G*) suggesting that detection of change in $PCO_2$ is an important role for RTN neurons.

A second possibility is that much of prior evidence for RTN chemosensory responses depends heavily on recordings from anaesthetized animals and that anaesthesia alters the dynamics of neuronal responses to $CO_2$. When we compared RTN responses in awake and anaesthetized animals we found that deep urethane anaesthesia dramatically changed neuronal firing patterns before and during hypercapnia (*Figure 4—figure supplement 1*). While the choice of anaesthetic agent and depth of anaesthesia is likely to alter the degree to which RTN neurons retain their natural activity, our recordings suggest that studies of chemosensory responses in anaesthetised preparations should be interpreted with caution.

These possibilities are not mutually exclusive and it is likely that they may all contribute to the relatively low proportion of neurons of the $E_G$ subtype in our dataset. Unexpectedly, whilst we recorded from sniff activated neurons, we did not find any neurons with sigh related activity, which may be due to their sparsity in number (*Li et al., 2016*). Therefore, more recordings may be necessary to uncover further functional subpopulations within the RTN region.

Several lines of evidence have suggested the importance of medullary Raphe serotonergic neurons in mediating respiratory chemosensitivity: the proximity of raphe neuron processes to blood vessels

(*Bradley et al., 2002*); the correspondence of the location of $CO_2$-dependent ATP release at the medullary surface and the raphe neurons (*Bradley et al., 2002*; *Gourine et al., 2005*); suppression of the medullary raphe by isofluorane (*Johansen et al., 2015*; *Ray et al., 2011*) which matches the reduction in the overall hypercapnic ventilatory response; and most compellingly the observation that inhibition of their activity via inhibitory DREADD receptors removes ~40% of the total adaptive ventilatory response in awake mice (*Ray et al., 2011*). There were two classes of neuron excited by $CO_2$ and these marked the presence and magnitude of the hypercapnic stimulus ($E_G$) and the start of the hypercapnic episode ($E_A$), supports the hypothesis that the Raphe acts as a primary chemosensory nucleus.

Of the 2 known serotonergic subtypes in the Raphe, NK1R-negative serotonergic neurons project to areas responsible for $CO_2$ integration (*Brust et al., 2014*), whereas NK1R-positive serotonergic neurons project to motor nuclei responsible for airway patency and co-ordination, and diaphragmatic movements (*Hennessy et al., 2017*). Interestingly both subtypes project to the preBötC (*Brust et al., 2014*; *Hennessy et al., 2017*). Furthermore, both serotonergic and GABAergic raphe neurons project to the $pF_L$ (*Silva et al., 2020*), with serotonergic neurons also innervating areas that influence expiration, namely the C1 (*Malheiros-Lima et al., 2020*), and NTS (*Silva et al., 2019*). Therefore, Raphe neurons are well placed to have far-reaching effects on breathing.

While both the RTN and Raphe have a high proportion of neurons that can be activated by $CO_2$, there are interesting differences in the nature of the responses that we observed (*Figure 8A*). The majority of RTN neurons that we observed were of the $E_A$ subtype, thus they will encode a change in inspired $CO_2$ rather than the presence of the entire stimulus. Even the $E_G$ neurons responded to a change in inspired $CO_2$ with a transient change in $Ca^{2+}$ activity in addition to the graded response. By contrast the situations was reversed in the Raphe: we observed a greater number of Raphe neurons were of the $E_G$ subtype than the $E_A$ subtype. This may suggest a complementarity between the RTN and Raphe, such that the more superficially located RTN rapidly encodes changes in $PCO_2$, with the deeper sitting Raphe providing a more sustained and graded response to encode the magnitude. It is noteworthy that plethysmographic changes in breathing are well-known to partially adapt to step changes in inspired $CO_2$ (illustrated in *Figure 2H and J*, see also *Bravo et al., 2016*) suggesting that the $E_A$ subtype is physiologically important. The rapid response to a sudden change in inspired $CO_2$ could plausibly be regarded as a trigger to quick behavioural adaptation that could remove the animal from the source of hypercapnia, or trigger a rapid adjustment in breathing that is sufficient to restore normocapnia.

$CO_2$-inhibited neurons in the RTN and Raphe have been previously described (*Iceman et al., 2014*; *Nattie et al., 1993*). Our recordings give further evidence for these subtypes in both nuclei. In the Raphe, $CO_2$ inhibited neurons were thought to be exclusively GABAergic. Our finding that serotonergic Raphe neurons can be inhibited by $CO_2$ is unexpected. The mechanism of $CO_2$-dependent inhibition is unclear. One possibility is that a $CO_2$ activated $K^+$ channel is present in these cells. An inward rectifier $K^+$ channel that is activated by $CO_2$ has been described in HeLa cells (*Huckstepp and Dale, 2011a*), indicating that this mechanism of intrinsic sensitivity is possible. However, a likelier alternative hypothesis is that the inhibitory action of $CO_2$ on the serotonergic and GABAergic neurons is indirect and depends upon synaptic connectivity -for example via $CO_2$-dependent activation of local GABAergic interneurons that are present in the medullary Raphe (*Figure 8B*).

While the emphasis of chemosensory mechanisms has been on the $CO_2$/pH-dependent activation of excitatory neurons, $CO_2$/pH-dependent inhibition of neurons could also be a powerful contributor to the hypercapnic ventilatory reflex (*Figure 8A*). While we have not demonstrated synaptic connections from any of the classes of neurons we describe, if $CO_2$-inhibited neurons were to tonically excite neurons that normally inhibit the preBötzinger complex, a process of disinhibition during hypercapnia could result in greater excitation of preBötzinger complex neurons (*Figure 8A*). RTN and Raphe neurons are known to project to the Bötzinger complex (*Morinaga et al., 2019*; *Rosin et al., 2006*), which contains inhibitory neurons (*Ezure and Manabe, 1988*), whilst RTN neurons are also known to project to inhibitory cells of the preBötC (*Yang et al., 2019*), which regulate the synchronisation of excitatory bursting and the emergence of the inspiratory rhythm (*Ashhad and Feldman, 2020*). The $CO_2$-dependent disinhibition of the inhibitory interneurons in the preBötC could contribute to the emergence of more powerful inspiratory bursts and hence an increase in tidal volume (*Figure 8A*).

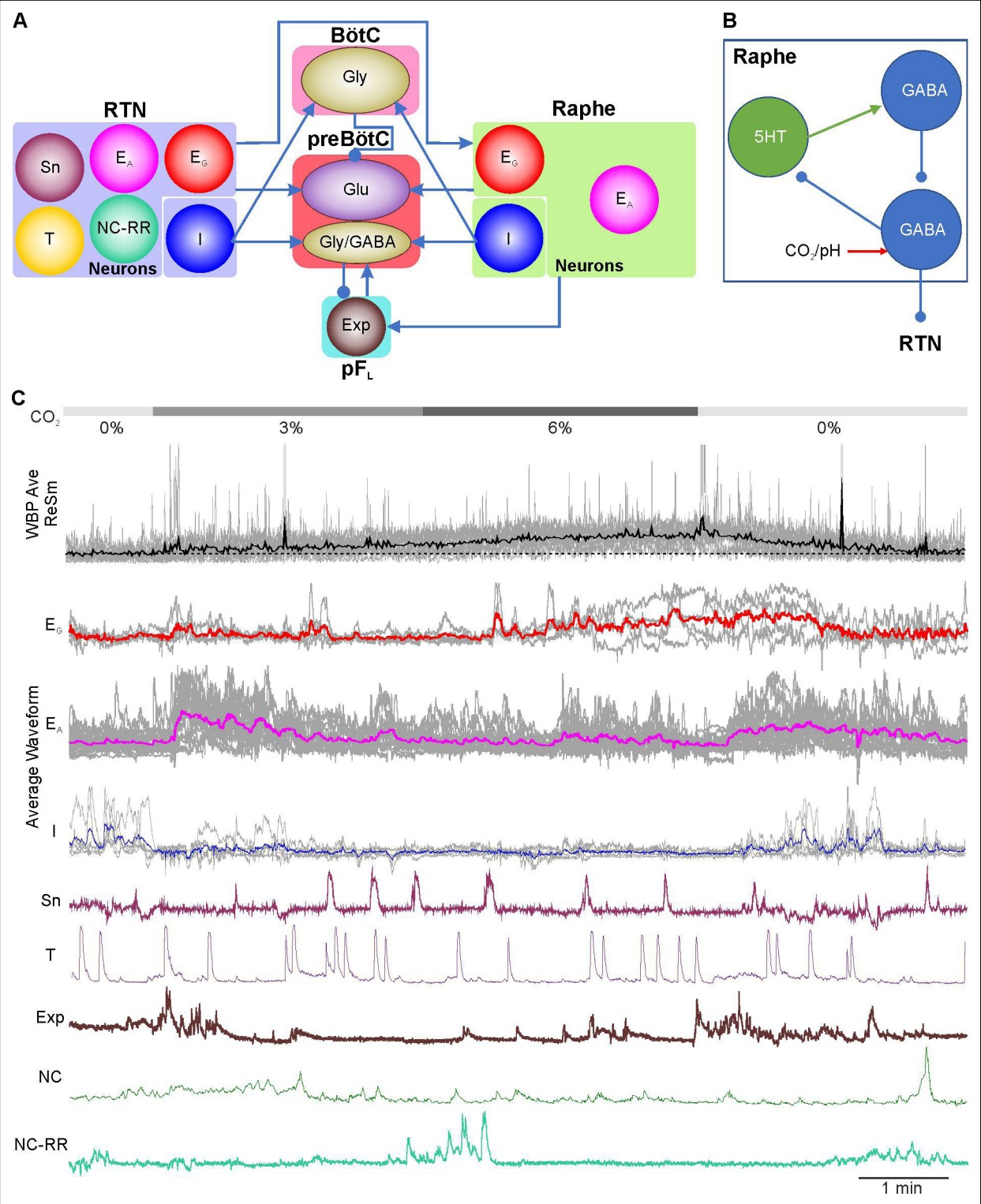

**Figure 8.** Summary diagram of findings in the paper showing contribution of the RTN, Raphe and pF$_L$ neurons to respiratory chemosensitivity and active expiration. (**A**) Block diagram showing the types of activity pattern observed in neurons of the RTN, pF$_L$ and Raphe. Proposed potential interconnections to the preBötzinger and Bötzinger complexes are speculative and based on regional interactions reported in the literature. (**B**) Hypothesis for generating CO$_2$ dependent inhibition of serotonergic neurons. CO$_2$ is proposed to activate GABAergic neurons that inhibit tonically active serotonergic neurons which are not directly CO$_2$ sensitive. If these were to activate a separate population of non-CO$_2$ sensitive GABAergic neurons, these could normally silence the CO$_2$ sensitive neurons until an episode of hypercapnia. This mechanism could explain an indirect inhibitory action of CO$_2$ on both serotonergic and GABAergic neurons. (**C**) Averages of the different types of neuronal activity seen in the RTN, Raphe and pF$_L$ during hypercapnia aligned to an average of the WBP trace.

## Non-chemosensory mechanisms in the RTN

The RTN showed several forms of $Ca^{2+}$ activity that did not respond to hypercapnia. In support of previous findings (*Nattie et al., 1993*), we saw a small population of tonically firing neurons that were unaffected by elevating inspired $CO_2$. These neurons might provide tonic drive to the respiratory network that drives tidal volume, which is known to originate at least in part from the RTN (*Huckstepp et al., 2015*). However, in 67/98 of the RTN neurons, we observed some tonic activity during room air breathing. Therefore, about 2/3 of the RTN neurons we imaged could plausibly contribute to tonic drive at rest. In addition to providing drive to basal breathing, tonic activity in neurons may also provide a level of baseline excitation that permits for the system to be more easily manipulated, allowing other respiratory features to be expressed. For example there is an absolute requirement for network wide activity to enable expiratory motor output to occur (*Huckstepp et al., 2016*). Furthermore, it is well documented that as respiratory network excitation decreases during sleep, sensitivity to $CO_2$ diminishes leading to lower minute ventilation, a higher resting $PaCO_2$ (*Douglas et al., 1982a*), and reduced hypercapnic ventilatory responses (*Douglas et al., 1982b*). Thus, tonic activity in the RTN may be vital to allow a greater sensitivity to $CO_2$ during wakefullness.

We also found non-coding neurons in the RTN. These neurons could be subdivided into those that displayed $Ca^{2+}$ activity not related to the level of $CO_2$, nor any abstractable feature on the respiratory recording (NC), and those that had $Ca^{2+}$ activity that matched variations in respiratory output (NC-RR). This latter population were largely silent when $V_T$ was at its greatest but displayed activity when the respiratory frequency was high and the faster ventilatory cycle meant that $V_T$ was smaller. The NC-RR neurons might therefore provide excitation to increase respiratory frequency. The NC sub-population could represent a novel neuronal type in the RTN, or could be a known non-$CO_2$ coding subpopulation that are likely to have other functions such as those that express high levels neuromedin B (*Shi et al., 2017*).

Sniffing is a mechanism that allows for olfactory sampling of the air to discern location of smell and to test for irritants before air is taken into the lung. Sniffing is obligatory for odour perception (*Mainland and Sobel, 2005*), and thus activates the piriform cortex (*Sobel et al., 1998a*). However, sniffing also causes neuronal activity in the absence of odorants (*Sobel et al., 1998a*). It activates the hippocampus (*Vanderwolf, 2001*), and the cerebellum (*Sobel et al., 1998b*), respectively, likely to prime odour-related memory recall, and co-ordination of movement toward or away from specific odours. Whilst the activation of these brain regions during sniffing has been well documented, the pattern generating microcircuit for sniffing has only recently been identified. Sniffing requires the activation of two sets of muscles (i) the nasal muscles to open the airway and decrease resistance, allowing more rapid movement of air through the nose, and (ii) the respiratory (inspiratory and expiratory) muscles, to draw air in and out of the nasal cavity. The RTN is located adjacent to, and innervates, the facial motor nucleus (*Deschênes et al., 2016*; *Kurnikova et al., 2018*). These RTN projections to the facial are directly involved in sniffing (*Deschênes et al., 2016*) and control of nasal direction during the odour response (*Kurnikova et al., 2018*). In conjunction the RTN provides significant drive to the preBötC (*Rosin et al., 2006*), and the $pF_L$ (*Zoccal et al., 2018*), both of which are integral parts of the sniffing microcircuit (*Deschênes et al., 2016*; *Moore et al., 2013*). Therefore, it is not surprising that the sniff pathway originates in, or passes through, the RTN.

## Active expiration

The $pF_L$ is thought to be a subsidiary conditional oscillator for expiration that is normally suppressed at rest (*Huckstepp et al., 2016*). That all $pF_L$ neurons showed the same discharge pattern, supports the hypothesis that the $pF_L$ is comprised of a single homogenous group of neurons acting with a single purpose (*Huckstepp et al., 2018*). Even though the nucleus appears to integrate information from many nuclei -RTN (*Zoccal et al., 2018*), C1, (*Malheiros-Lima et al., 2020*), NTS, rostral pedunculopontine tegmental (*Silva et al., 2019*) and the preBötC (*Huckstepp et al., 2016*) -the $pF_L$ appears to be a simple on-off switch, rather than forming complex outputs as we describe in the RTN.

Surprisingly, we found that $pF_L$ neurons were only transiently active during large expiratory efforts, (such as sniffing, sighing, or other forms of brief deep breathing), in contrast to the sustained active expiration previously reported (*Huckstepp et al., 2015*; *Leirão et al., 2018*). This may be due to our use of brief periods of graded $CO_2$ in conscious animals, rather than sustained $CO_2$ (~1 hr) (*Leirão et al., 2018*) or use of the vagotomised anaesthetised preparation in prior studies (*Huckstepp et al.,*

*2015*). Importantly, in the un-anaesthetised, vagi intact, in situ preparation of young adult rats, $pF_L$ neurons only displayed a small number of action potentials (~10) during the latter portion of expiratory phase 2 (approximately the final 20% of the expiratory period) (*Magalhães et al., 2021*). In our recordings the expiratory period was ~100ms at the highest level of $CO_2$, allowing for 20ms of $pF_L$ neuronal discharge during times of active expiration. Thus $Ca^{2+}$ signals in $pF_L$ neurons were likely too small to elicit a recordable change in GCaMP fluorescence, except when expiratory efforts were notably large in terms of both duration and amplitude. Furthermore, in contrast to our expectation that $pF_L$ neurons would be phasically active between inspiratory bursts, $pF_L$ neurons exhibited elevated intracellular calcium beginning before, and spanning, the expiratory period.

## Concluding remarks

Endoscopic imaging of the activity of the brainstem neurons involved in the control of breathing provides a new way to investigate the neural circuitry for the chemosensory control of breathing in awake adult mice, and to understand how breathing is coordinated with complex non-ventilatory behaviours. Some components, such as expiratory activity of the $pF_L$ or sniff activity in the RTN, are easy to interpret, whilst others are more complicated. The responses to $CO_2$ were more heterogeneous in both hSyn+ RTN and hSyn+ and serotonergic Raphe neurons than would be expected from the prior literature. Notably, our dataset of RTN chemosensitive neurons contained a preponderance of the $E_A$ subtype. The $E_A$ neuronal responses are potentially well suited to detect small increases of inspired $CO_2$ outside the normal range that can occur during normal behavioural conditions. In contrast, the Raphe neurons tended to be active during the entire $CO_2$ stimulus and conceivably these neurons could take over from RTN neurons during more sustained episodes of hypercapnia. These data provide an intriguing possibility that the chemosensory network is arranged in a hierarchy with each layer conveying a unique range of $CO_2$.

# Materials and methods

**Key resources table**

| Reagent type (species) or resource | Designation | Source or reference | Identifiers | Additional information |
|---|---|---|---|---|
| Gene (*Mus musculus*) | Promoter of mouse *Slc6a4*, Gene Accession: NM_010484 | Genecopoiea | MPRM41232-PG02 | |
| Strain, strain background (*Mus musculus*, male) | wild-type | Jackson Laboratories | | C57BL/6 J background |
| Cell line (*Homo sapiens*) | 293AAV Cell Line | CellBioLabs | AAV-100 | HEK293-T |
| Antibody | (Goat polyclonal) anti- cholineacetyl transferase antibody | Millipore | RRID:AB_262156 | (1:100 dilution) |
| Antibody | (Rabbit polyclonal) anti- Tryptophan hydroxylase 2 antibody | Millipore Sigma | RRID:AB_10806898 | (1:500 dilution) |
| Antibody | (Rabbit polyclonal) anti- NMB (CENTER) antibody | Sigma-Aldrich | RRID:AB_2619620 | (1:1000 dilution) |
| Antibody | (donkey polyclonal) anti- rabbit IgG (H+L) Alexa Fluor 680 | Jackson Immunoresearch | RRID:AB_2340627 | (1:250 dilution) |
| Antibody | (donkey polyclonal) anti- rabbit IgG (H+L) Alexa Fluor 594 | Jackson Immunoresearch | RRID:AB_2340621 | (1:250 dilution) |
| Antibody | (donkey polyclonal) anti- goat IgG (H+L) Alexa Fluor 568 | Abcam | RRID:AB_2636995 | (1:250 dilution) |

*Continued on next page*

*Continued*

| Reagent type (species) or resource | Designation | Source or reference | Identifiers | Additional information |
|---|---|---|---|---|
| Antibody | (donkey polyclonal) anti- goat IgG (H+L) Alexa Fluor 680 | Jackson Immunoresearch | RRID:AB_2340432 | (1:250 dilution) |
| Recombinant DNA reagent | AAV-9: pGP-AAV-syn-GCaMP6s-WPRE.4.641 | Addgene, Watertown, MA, USA | BS1-NOSAAV9 | Dilution 1:10 |
| Recombinant DNA reagent | AAV-DJ: pAAV-SERT-GCaMP6s | This paper | Raphe neuronal specific virus with calcium indicator | Dilution 1:10 |
| Recombinant DNA reagent | AAV-9: pGP-AAV-syn-GCaMP6f-WPRE.24.693 | Addgene, Watertown, MA, USA | BS3-NXFAAV9 | Dilution 1:10 |
| Recombinant DNA reagent | pAAV.Syn.GCaMP6s.WPRE.SV40 | Addgene, Watertown, MA, USA | Plasmid#100843 | |
| Recombinant DNA reagent | AAV packaging plasmid pAAV-DJ, AAV-DJ Helper Free Expression System | Cellbiolabs | VPK-410-DJ | |
| Recombinant DNA reagent | AAV packaging plasmid pHelper, AAV-DJ Helper Free Expression System | Cellbiolabs | VPK-410-DJ | |
| Sequence-based reagent | VC327-4 6 s HindIII for | This paper | PCR primers | 5´- TTGACTGCCTAAGCTTgccaccatgcatcatcatcatcatg 300 nM |
| Sequence-based reagent | VC327-4 6 s Afe rev | This paper | PCR primers | 5'-GATCTCTCGAGCAGCGCTtcacttcgctgtcatcatttgtacaaact 300 nM |
| Peptide, recombinant protein | PrimeSTAR GXL DNA Polymerase | Takara Bio Europe SAS | R050A | 1 U·µg-1 |
| Commercial assay, kit | In-Fusion HD Cloning Kit | Takara Bio Europe SAS | 639642 | |
| Chemical compound, drug | OptiPrep | PROGEN Biotechnik GmbH | 1114542 | Iodixanol |
| Commercial assay, kit | 2 x qPCRBIO SyGreen Mix Hi-ROX | PCRBiosystems | PB20.12 | |
| Chemical compound, drug | kainic acid | Fisher Scientific | 15467999 | 8 mg·kg$^{-1}$ IP |
| Chemical compound, drug | atropine | Westward Pharmaceutical Co. | 0641-6006-10 | 120 µg·kg$^{-1}$ |
| Chemical compound, drug | meloxicam | Norbrook Inc. | | 2 mg·kg$^{-1}$ |
| Chemical compound, drug | buprenorphine | Reckitt Benckiser | | 100 µg·kg-1 |
| Chemical compound, drug | PEI MAX Hydrochloride Transfection Grade Linear 40,000 mwco | Polysciences Europe GmbH | 24765–1 | 1 µg·µL-1 |

*Continued on next page*

*Continued*

| Reagent type (species) or resource | Designation | Source or reference | Identifiers | Additional information |
|---|---|---|---|---|
| Software, algorithm | Inscopix Data Processing Software (IDPS) | Inscopix | Python Tools 2.2.6 for Visual Studio 2015 | Version: 1.6.0.3225 |
| Software, algorithm | Spike2 | Cambridge Electronic Design | RRID:SCR_000903 | Version: 8.23 |
| Software, algorithm | Prism 9 | GraphPad | RRID:SCR_002798 | Version 9.1.0 |
| Software, algorithm | ZEN | Carl Zeiss | RRID:SCR_013672 | Version 11.0.0.190 |

Experiments were performed in accordance with the European Commission Directive 2010/63/EU (European Convention for the Protection of Vertebrate Animals used for Experimental and Other Scientific Purposes) and the United Kingdom Home Office (Scientific Procedures) Act (1986) with project approval from the University of Warwick's AWERB.

## Cell lines

To produce AAV particles, we obtained a proprietary cell line, HEK293AAV cells, directly from Cell-BioLabs for this study. The certification from CellBioLabs stated the cells' identity and that they were free of microbial contamination. These cells were only used to produce the AAV particles and were not used by themselves to generate any of the data in this study.

## AAV-DJ: pAAV-SERT-GCaMP6s vector particle production and purification

The CMV promoter region was excised from pAAV-GFP (Cellbiolabs) by restriction digest with MluI and ClaI, blunting with Klenow Fragment (TheroScientific) and re-ligation with T4 DNA ligase (ThermoScientific) to generate pAAV-GFP without promoter. A SERT promoter clone was purchased from Genecopoiea (Product ID: MPRM41232-PG02, Symbol: Slc6a4, Species: Mouse, Target Gene Accession: NM_010484, Alias: 5-HTT, AI323329, Htt, SertA). The promoter region was cut out with EcoRI and HndIII and introduced by T4 DNA ligation into the previously modified pAAV-GFP to generate pAAV-SERT. Subsequently, a PCR performed using PrimeStar GLX Polymerase (Takara Clontech) with pAAV.Syn.GCaMP6s.WPRE.SV40 (Addgene Plasmid#100843) as a template (forward primer 5´- TTGACTGCCTAAGCTTgccaccatgcatcatcatcatcatg and reverse primer 5´- GATCTCTCGAGCAGCGCTtcacttcgctgtcatcatttgtacaaact) to amplify the GCaMP6s fragment. pAAV-SERT was digested with HindIII and AfeI and the PCR product was inserted by InFusion Cloning (Takara Clontech) to generate pAAV-SERT His-GCaMP6s. Accuracy of all cloning steps was verified by PCR, restriction digests and DNA sequencing analysis.

AAV-DJ pseudotyped vectors were generated by co-transfection of HEK293-AAV cells (Cellbiolabs) with the AAV transfer plasmid pAAV-SERT His-GCaMP6s and the AAV packaging plasmid pAAV-DJ and pHelper (both Cellbiolabs). HEK293AAV cells (Cellbiolabs) were cultivated in Dulbecco's modified Eagle's medium (DMEM, High Glucose, Glutamax) supplemented with 10% (v/v) heat-inactivated fetal calf serum, 0.1 mM MEM Non-Essential Amino Acids (NEAA), 100 U/ml penicillin and 100 µg/ml streptomycin. Tissue culture reagents were obtained from Life Technologies. Briefly, $1 \times 10^7$ HEK293-AAV cells were seeded one day before transfection on 15 cm culture dishes and transfected with 7.5 µg pAAV-DJ, 10 µg pHelper and 6.5 µg pAAV plasmid per plate complexed with Max-polyethylenimine (PEI, Polysciences) at a PEI:DNA ratio (w/w) of 3:1. After 72 hr cells were harvested and resuspended in 5 ml lysis buffer (50 mM Tris base, 150 mM NaCl, 5 mM MgCl$_2$, pH 8.5). After three freeze-thaw cycles, benzonase (Merk; final concentration 50 U/ml) was added and the lysates were incubated for 1 hr at 37 °C. Cell debris was pelleted and vector containing lysates were purified using iodixanol step gradients. Finally, iodixanol was removed by ultrafiltration using Amicon Ultra Cartridges (50 mwco) and three washes with DPBS.

The genomic titers of DNase-resistant recombinant AAV particles were determined after alkaline treatment of virus particles and subsequent neutralization by qPCR using the qPCRBIO SY Green Mix Hi-Rox (Nippon Genetics Europe GmbH) and an ABI PRISM 7900HT cycler (Applied Biosystems).

Vectors were quantified using primers specific for the GCaMP6s sequence (5'- CACAGAAGCAGA GCTGCAG and 5'- actggggaggggtcacag). Real-time PCR was performed in a total volume of 10 μl with 0.3 μM for each primer. The corresponding pAAV transfer plasmid was used as a copy number standard. A standard curve for quantification was generated by serial dilutions of the respective plasmid DNA. The cycling conditions were as follows: 50 °C for 2 min, 95 °C for 10 min, followed by 35 cycles of 95 °C for 15 s and 60 °C for 60 s. Calculations were done using the SDS 2.4 software (Applied Biosystems).

## Virus handling

Raphe and RTN neurons - AAV-9: pGP-AAV-syn-GCaMP6s-WPRE.4.641 at a titre of $1 \times 10^{13}$ GC·ml$^{-1}$ (Addgene, Watertown, MA, USA); Raphe: AAV-DJ: pAAV-SERT-GCaMP6s at a titre of $1.8 \times 10^{13}$ GC·ml$^{-1}$ (University hospital Hamburg-Eppendorf, Hamburg, Germany). pF$_L$ - AAV-9: pGP-AAV-syn-GCaMP6f-WPRE.24.693 at a titre of $1 \times 10^{13}$ GC·ml$^{-1}$ (Addgene, Watertown, MA, USA).

Viruses were aliquoted and stored at –80 °C. On the day of injection, viruses were removed and held at 4 °C, loaded into graduated glass pipettes (Drummond Scientific Company, Broomall, PA, USA), and placed into an electrode holder for pressure injection. The AAV-syn-GCaMP6s and AAV-syn-GCaMP6f vectors use the synapsin promoter, and therefore transduced neurons showing higher tropism for the AAV 2/9 subtype, and to a much lesser extent neurons that show low tropism for AAV 2/9 within the injection site, e.g. facial motoneurons, they do not transduce non-neuronal cells. Vector AAV SERT-GCaMP6s was specific for serotonergic neurons.

## Viral transfection of RTN, Raphe and pF$_L$ neurons

Adult male C57BL/6 mice (20–30 g) were anaesthetized with isoflurane (4%; Piramal Healthcare Ltd, Mumbai, India) in pure oxygen (4 L·min$^{-1}$). Adequate anaesthesia was maintained with 0.5–2% Isoflu-orane in pure oxygen (1 L·min$^{-1}$) throughout the surgery. Mice received a presurgical subcutaneous injection of atropine (120 μg·kg$^{-1}$; Westward Pharmaceutical Co., Eatontown, NJ, USA) and meloxicam (2 mg·kg$^{-1}$; Norbrook Inc, Lenexa, KS, USA). Mice were placed in a prone position into a digital stereo-taxic apparatus (Kopf Instruments, Tujunga, CA, USA) on a heating pad (TCAT 2-LV: Physitemp, Clifton, NJ, USA) and body temperature was maintained at a minimum of 33°C via a thermocouple. The head was levelled at bregma, and 2 mm caudal to bregma, and graduated glass pipettes containing the virus were placed stereotaxically into either the RTN, rostral medullary raphe or pF$_L$ (*Figures 2A, 5A and 7A*). The RTN was defined as the area ventral to the caudal half of the facial nucleus, bound medi-ally and laterally by the edges of the facial nucleus (coordinates with a $9^0$ injection arm angle: –1.0 mm lateral and –5.6 mm caudal from Bregma, and –5.5 mm ventral from the surface of the cerebellum; *Figure 2A*). The Raphe was defined as the medial, TPH containing regions level with the caudal face of the facial nucleus: the Raphe Magnus (RMg) was directly above the pyramidal tracts (Py), and the Raphe Pallidus (RPa) was bound laterally by the Py (coordinates with a $9^0$ injection arm angle: 0 mm lateral and –5.8 mm caudal from Bregma, and –5.2 mm ventral from the surface of the cerebellum; *Figure 5A*). The pF$_L$ was defined as the neurons bound laterally by the spinothalmic tract and medially by the lateral edge of the facial motor nucleus (coordinates with a $9^0$ injection arm angle: –1.8 mm lateral and –5.55 mm caudal from Bregma, and –4.7 mm ventral from the surface of the cerebellum; *Figure 7A*). The virus solution was pressure injected (<300 nL) unilaterally. Pipettes were left in place for 3–5 min to prevent back flow of the virus solution up the pipette track. Postoperatively, mice received intraperitoneal (IP) injections of buprenorphine (100 μg·kg$^{-1}$; Reckitt Benckiser, Slough, UK). Mice were allowed 2 weeks for recovery and viral expression, with food and water ad libitum.

## GRIN lens implantation

Mice expressing GCaMP6 were anesthetized with isoflurane, given pre-surgical drugs, placed into a stereotax, and the head was levelled as described above. To widen the lens path whilst producing the least amount of deformation of tissue, a graduated approach was taken; firstly a glass pipette was inserted down the GRIN lens path to a depth 200 μm above where the lens would terminated and left in place for 3 min; this procedure was then repeated with a blunted hypodermic needle. The GRIN lens (600 μm diameter, 7.3 mm length; Inscopix, Palo Alto, CA, USA) was then slowly inserted at a rate of 100 μm·min$^{-1}$ to a depth ~1300 μm above the target site, then lowered at a rate of 50 μm·min$^{-1}$ to a depth ~300 μm above the RTN, Raphe or pF$_L$ (coordinates with a $9^0$ injection arm angle: RTN – 1.1 mm

lateral and –5.75 mm caudal from Bregma, and –5.3 mm ventral from the surface of the cerebellum; Raphe – 0 mm lateral and –5.95 mm caudal from Bregma, and –5.1 mm ventral from the surface of the cerebellum; $pF_L$ – 1.7 mm lateral and –5.7 mm caudal from Bregma, and –4.6 mm ventral from the surface of the cerebellum). The lens was then secured in place with SuperBond (Prestige Dental, Bradford, UK). Postoperatively, mice received buprenorphine, and were allowed 2 weeks for recovery, with food and water ad libitum.

## Baseplate installation

Mice expressing GCaMP6 and implanted with GRIN lens were anesthetized with isofluorane, given pre-surgical drugs, and placed into a stereotax as described above. To hold the miniaturized microscope during recordings, a baseplate was positioned over the lens and adjusted until the cells under the GRIN lens were in focus. The baseplate was then secured with superbond, and coated in black dental cement (Vertex Dental, Soesterberg, the Netherlands) to stop interference of the recording from ambient light. Mice were allowed 1 week for recovery, with food and water ad libitum.

## Ca²⁺ imaging in freely moving mice

All mice were trained with dummy camera and habituated to plethysmography chamber at least twice before imaging. The miniature microscope with integrated 475 nm LED (Inscopix, Palo Alto, CA, USA) was secured to the baseplate. GCaMP6 fluorescence was visualised through the GRIN lens, using nVista 2 HD acquisition software (Inscopix, Palo Alto, CA, USA). Calcium fluorescence was optimised for each experiment so that the histogram range was ~150–600, with average recording parameters set at 10–20 frames/sec with the LED power set to 10–20 mW of light and a digital gain of 1.0–4.0. A TTL pulse was used to synchronize the calcium signalling to the plethysmography trace. All images were processed using Inscopix data processing software (Inscopix, Palo Alto, CA, USA). GCaMP6 movies were ran through: preprocessing algorithm (with temporal downsampling), crop, spatial filter algorithm (0.005–0.5 Hz), motion correction and cell identification through manual regions of interest (ROIs) operation to generate the identified cell sets. Cell sets were imported into *Spike2* software for processing.

## Plethysmography

Mice were placed into a custom-made 0.5 L plethysmography chamber, with an airflow rate of 1 l·min⁻¹. The plethysmography chamber was heated to 31 °C (thermoneutral for C57/BL6 mice). $CO_2$ concentrations were sampled via a Hitech Intruments (Luton, UK) GIR250 Dual Sensor Gas analyzer or ML206 gas analyzer (ADinstruments, Sydney, Australia) connected to the inflow immediately before entering the chamber. The analyser had a delay of ~15–20 sec to read-out the digital output of gas mixture. Pressure transducer signals and $CO_2$ measurements were amplified and filtered using the NeuroLog system (Digitimer, Welwyn Garden City, UK) connected to a 1401 interface and acquired on a computer using *Spike2* software (Cambridge Electronic Design, Cambridge, UK). Video data was recorded with *Spike2* software and was synchronised with the breathing trace. Airflow measurements were used to calculate: tidal volume ($V_T$: signal trough at the end of expiration subtracted from the peak signal during inspiration, converted to mL following calibration and normalized to body weight), and respiratory frequency (*f*R: breaths per minute). Minute ventilation ($V_E$) was calculated as $V_T$ x *f*R.

## Hypercapnia in freely behaving mice

Instrumented mice were allowed ~30 mins to acclimate to the plethysmograph. The LED was activated through a TTL pulse synchronised with the *Spike2* recording and 1 or 3 min of baseline recordings were taken (gas mixture: 0% $CO_2$ 21% $O_2$ 79% $N_2$). The mice were then exposed to 3 min epochs of hypercapnic gas mixture at different concentrations of $CO_2$: RTN and Raphe transduced mice were exposed to 3% followed by 6% $CO_2$, and $pF_L$ transduced mice were exposed to 6% followed by 9% $CO_2$. All gas mixtures contained 21% $O_2$ balanced $N_2$. Following exposure to the hypercapnic gas mixtures, $CO_2$ levels were reduced back to 0% and calcium signals were recorded for a further 3–4 minutes recovery period.

## Hypercapnia and seizure induction in urethane anaesthetised mice

Instrumented mice were anaesthetised with an IP injection of 1.2–1.5 g·kg-1 urethane (Sigma-Aldrich, St Louis, MO, USA) and placed into the plethysmograph. For recording responses of RTN neurons to hypercapnia, the LED was activated and 1 or 3 minute of baseline recording was taken. The mice were then exposed to 2 or 3 minute epochs of hypercapnic gas mixture 3%, 6% and 9% $CO_2$ (in 21% $O_2$ balanced $N_2$) sequentially, as experiments into the chemosensitivity of RTN neurons in mice often use 6% $CO_2$ in freely behaving experiments but 9% $CO_2$ in anaesthetised animals. Following exposure to the hypercapnic gas mixtures, $CO_2$ levels were reduced back to 0% and calcium signals were recorded for a further 3 minute recovery period.

For neurons recorded in the RTN, after completion of hypercapnic responses and a period of rest, baseline $Ca^{2+}$ activity was recorded for 5 min. The mice were then injected with a dose of kainic acid (8 mg·kg$^{-1}$ IP) sufficient to induce an electrographic seizures in the absence of any movement from behavioural seizures. Following injection of kainic acid, calcium activity was recorded every alternate 5 min to avoid the fluorophore bleaching and concurrently record the calcium activity for a long enough to evidence the effect of induction of seizure on the activity of RTN neurons (at least for 90 min post kainic acid injection).

## Preparation of fixed brain slices

Mice were humanely killed by pentobarbital overdose (>100 mg·kg$^{-1}$) and transcardially perfused with paraformaldehyde solution (4% PFA; Sigma-Aldrich, St Louis, MO, USA). The head was removed and postfixed in PFA (4 °C) for 3 days to preserve the lens tract. The brains were removed and postfixed in PFA (4 °C) overnight. Brainstems were serially sectioned at 50–70 μm.

## Immunohistochemistry

Free-floating sections were incubated for 1 hr in a blocking solution (PBS containing 0.1% Triton X-100 and 5% BSA). The tissue was then incubated overnight at room temperature in primary antibodies: goat anti-choline acetyl transferase [ChAT; 1:100; Millipore, Burlington, MA, USA] alone for co-labelling with GCaMP6, or rabbit anti-tryptophan hydroxylase [TPH; 1:500; Sigma-Aldrich, St Louis, MO, USA] alone for co-labelling with GCaMP6.

Slices were washed in PBS (6 × 5 min) and then incubated in a blocking solution to which secondary antibodies were added; donkey anti-rabbit Alexa Fluor 594 (1:250; Jackson Laboratory, Bar Harbor, ME, USA) for co-labelling with GCaMP6, or donkey anti-goat Alexa Fluor 594 (1:250; Jackson Laboratory, Bar Harbor, ME, USA; RTN) for co-labelling with GCaMP6 in RTN tissue or donkey anti-goat Alexa Fluor 568 (1:250; Jackson Laboratory, Bar Harbor, ME, USA) for co-labelling with GCaMP6 in pF$_L$ tissue. The tissue was then incubated 2–4 hr at room temperature. Tissue was washed in PBS (6 × 5 min). Slices were mounted on polysine adhesion slides and were coverslipped with Vectashield Antifade Mounting Medium with DAPI (Vectorlabs, Burlingame, CA, United States).

## Heat-induced epitope retrieval and immunohistochemistry for NMB

Slices were mounted onto poly-lysine coated microscope slides and allowed to dry and adhere. Mounted brain sections were added to the pre-heated sodium citrate antigen retrieval buffer (10 mM sodium citrate, 0.05% Tween 20, pH 9.0) and incubated for 15 min. The mounted brain sections were then removed and washed in PBS.

The tissue was then incubated for 1 hr in a blocking solution. Without washing, the tissue was then incubated overnight at room temperature in blocking solution containing primary antibodies: (goat anti-ChAT [same as above] alone for co-labelling with GCaMP6, or in conjunction with rabbit anti-Neuromedin-B [NMB; 1:100; SAB1301059; Sigma-Aldrich, St Louis, MO, USA] antibody) for co-labelling of ChAT and NMB.

The tissue was then washed in PBS (6 × 5 min) and incubated for 1 hr in a blocking solution. Without washing, the tissue was then incubated 2 hr at room temperature in blocking solution containing secondary antibodies: donkey anti-rabbit Alexa Fluor 568 (1:250; Invitrogen, Waltham, MA, United States) for co-labelling with GCaMP6, or donkey anti-rabbit Alexa Fluor 488 (1:250; Invitrogen, Waltham, MA, United States) and donkey anti-goat Alexa Fluor 594 (1:250; Invitrogen, Waltham, MA, United States) for co-labelling of ChAT and NMB. Tissue was washed in PBS (6 × 5 min).

Slides were examined using a Zeiss 880 confocal microscope with ZEN acquisition software (Zeiss, Oberkochen, Germany).

## Antibody specificity

The antibodies that we used have been independently validated by others in the field: anti-ChAT (*Dempsey et al., 2015*; *Saunders et al., 2015*; *Zhang et al., 2020*); anti-TPH (*Pitzer et al., 2015*; *Quina et al., 2020*; *Zhong et al., 2017*) 1; and anti-NMB (*Li et al., 2016*). The staining patterns we report are highly consistent with these prior studies indicating that these antibodies are specifically recognising their targets with the tissue we have examined. Additionally, we compared NMB immunostaining with in situ hybridization patterns for NMB documented in the Allen Brain Atlas and found it to be very similar (*Figure 2—figure supplement 1*).

## Acknowledgements

This work was supported by an MRC Discovery Award MC_PC_15070. ND is a Royal Society Wolfson Research Merit Award Holder. We thank Tamara Sotelo-Hitschfeld for help in the early stages of developing the method.

## Additional information

### Funding

| Funder | Grant reference number | Author |
|---|---|---|
| Medical Research Council | MC_PC_15070 | Nicholas Dale |
| Royal Society | | Nicholas Dale |

The funders had no role in study design, data collection and interpretation, or the decision to submit the work for publication.

### Author contributions

Amol Bhandare, Data curation, Formal analysis, Investigation, Methodology, Writing - review and editing; Joseph van de Wiel, Investigation, Methodology; Reno Roberts, Investigation; Ingke Braren, Resources, Methodology; Robert Huckstepp, Formal analysis, Supervision, Investigation, Methodology, Writing - review and editing; Nicholas Dale, Conceptualization, Supervision, Funding acquisition, Writing - original draft, Project administration, Writing - review and editing

### Author ORCIDs

Amol Bhandare (iD) http://orcid.org/0000-0002-5214-9355
Robert Huckstepp (iD) http://orcid.org/0000-0003-4410-3397
Nicholas Dale (iD) http://orcid.org/0000-0003-2196-2949

### Ethics

Experiments were performed in accordance with the European Commission Directive 2010/63/EU (European Convention for the Protection of Vertebrate Animals used for Experimental and Other Scientific Purposes) and the United Kingdom Home Office (Scientific Procedures) Act (1986) with project approval from the University of Warwick's AWERB.

### Decision letter and Author response

Decision letter https://doi.org/10.7554/eLife.70671.sa1
Author response https://doi.org/10.7554/eLife.70671.sa2

## Additional files

### Supplementary files

• Supplementary file 1. A user-modifiable worksheet that uses simple mathematical functions to explore the predicted correlation between $V_T$ and $F/F_0$ for $E_A$ neurons under two assumptions. (i) no

adapting component in the $V_T$ response to hypercapnia; and (ii) an adapting $V_T$ component in the response to hypercapnia is present. This should be compared to the data shown in *Figure 2J*.

- Transparent reporting form

### Data availability

All data generated or analysed during this study are included in the MS and supporting files. Source data files have been provided for Figure 3D, Figure 7I–K, Figure 1—figure supplement 7, and Figure 4—figure supplement 2.

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
