## [Editor Report]

This paper presents a novel application of endoscopic imaging with miniscopes in awake-behaving mice to address the important problem of analyzing multicellular activity by calcium activity imaging within circumscribed regions of the medulla oblongata that are proposed to have chemosensory functions for the homeostatic regulation of breathing in response to elevated systemic CO2 (hypercapnia). The authors importantly demonstrate chemosensory responses of neurons in the retrotrapezoid nucleus-RTN, Raphe magnus and pallidus nuclei, and lateral parafacial region- pFL, and they describe regional heterogeneity of cellular responses to hypercapnia in these regions with important functional implications. Analyzing chemosensory properties of these medullary regions has been the focus of numerous studies, but the problem of analyzing regional multicellular chemosensory responses in the awake freely behaving rodent has not been previously addressed, so this paper represents an advance for the field. The authors present a novel catalog of neuronal responses, and they illustrate the feasibility of this imaging approach, paving the way for further development/application of this approach, while indicating its limitations.

---

## [Decision Letter]

**Decision letter after peer review:**

Thank you for submitting your article "Analyzing the brainstem circuits for respiratory chemosensitivity in freely moving mice" for consideration by *eLife*. Your article has been reviewed by 3 peer reviewers, including Jeffrey C Smith as Reviewing Editor and Reviewer #1, and the evaluation has been overseen by Catherine Dulac as the Senior Editor.

Essential revisions:

1) The authors should discuss more the technical limitations of their imaging approach. A major concern is whether the targeted regional neuronal and glial populations have been adequately sampled to reasonably understand the spatial and functional heterogeneity of the hypercapnic responses. There are important optical limitations as well as potential limitations associated with efficacy of cellular labeling with the vector constructs used. These are major caveats that need to be more thoroughly discussed in the context of technical limitations. The authors state for example that the GRIN lens placement was within the RTN (Discussion p. 10, para. 2) or terminated under the caudal pole of the facial nucleus (p. 6, para. 1) but the anatomical reconstructions consistently show the end of the lens in facial motor nucleus and in Methods (p. 17, para. 1), so as the authors discuss the coverage of the RTN populations may not be sufficient. The authors state that generally the lens was positioned ~300 µm above the regions to be imaged. Please provide information on the depth of field of the GRIN lens used and the typical field of view (including providing length scale bars on the cellular fluorescence images presented in the main text). Additional information on the viral transduction efficacy with the GCaMP constructs used is also essential.

2) Cell identification is another technical issue that needs to be addressed. It is critically important that the authors provide more convincing additional experimental data on the molecular identity of RTN neurons imaged to help with the interpretation of the results on this component. This should be done by employing an experimental strategy that allows for cell-type specific expression of GCaMP in RTN neurons (e.g., by using a Phox2b promoter), and/or by providing convincing additional quantification verifying that a high percentage of the GCaMP-expressing neurons imaged in the RTN region also express NMB and VGLUT2. For astrocytes and serotonin neurons, an attempt was made to restrict GCaMP expression to the targeted cell groups by using viruses including GFAP (gfaABC1D) or SERT promoters, but a non-specific neuronal promoter (synapsin) was used to drive GCaMP expression in RTN neurons or the pFL, with the obvious problem that the recordings were made from unidentified cells. Granted there is no specific marker for the pFL, but this approach is not justified for the RTN neurons. The molecular phenotype of CO2-sensitive RTN neurons has been well-established, and viral vectors using a Phox2b promoter have been employed for many years to preferentially target those RTN neurons. There are not a lot of RTN neurons (maybe 700-800 in mice) and there are many other neurons in the same region that likely were transduced with GCaMP and may account for the measured responses to CO2.

3) Information about specificity of the antibodies used should be provided, which is particularly important for the immunostaining for NMB and VGLUT2 used to suggest that some of the GCaMP-expressing parafacial cells were bona fide RTN neurons. There are no controls provided for the specificity of the antibody staining, and no references attesting to the quality of the antibodies, which should be standard practice (see Rhodes and Trimmer, J Neuroscience 2006; Saper, JCN, 2005; Saper and Sawchenko, ibid, 2003). Even if the staining were validated as specific for the relevant antigens, demonstrating that some of the GCaMP-expressing neurons were VGLUT2- and NMB-positive RTN neurons does not guarantee that the recorded cells were also RTN chemosensory neurons. This important caveat needs to be amplified in the text with appropriate disclaimers.

4) The authors need to explain more their interpretations of the measured GCaMP signals. For various neurons there may be a nonlinear relationship between theses signals and neuronal firing. For glia it is not clear that "activation" by hypercapnia would always cause a change in intracellular calcium, and yet signaling pathways could still be engaged that influence neighboring neurons. In addition, the authors need to explain why GCaMP6s was used for calcium activity imaging of RTN and Raphe neurons, whereas GCaMP6f was used for imaging astrocytes and pFL neurons. For comparisons of calcium dynamics between regions it is important that the same sensor is utilized which was done for the RTN region and Raphe neurons, but interpretation of what is being encoded by a particular sensor needs to take into account the sensor signaling kinetics. GCaMP6s kinetics produce long (many seconds) calcium signal decays after signal onset, which can for example contribute to features such as adaptation of the calcium dynamics imaged and would not accurately reflect neuronal activity profiles per se.

5) The authors should discuss more how they adequately ruled out movement artifacts affecting the calcium signals. The authors provide some evidence that the measured Ca transients are independent of movement, but the GFP signal measured in different mice is not the best and adequate control for movement artifacts in other GCaMP-expressing mice. Inscopix miniscopes are capable of dual color imaging to detect a co-expressed fluorescent marker in the same cell, which would be a better control.

6) The calcium signals are interpreted by the authors to reflect a variety of cellular response patterns, but there is no indication of how the authors verified consistency of these patterns, and this important issue needs to be addressed. Did the authors apply the CO2 challenges in repeated or alternately ordered fashion (0-3-6, 0-6-3) to more convincingly verify a consistent response in each individual cell? This is important information to include. This approach would also aid in dissociating common CO2-associated response patterns from any more randomly associated movement artifacts, for example by the use of peristimulus averaging to tease out signal from noise.

7) The authors suggest that tonic RTN neurons "provide tonic drive to the respiratory network." However, it is possible that these neurons are not respiratory. Just because they are in/near the RTN does not mean they are chemoreceptors. Please discuss more carefully.

8) There was little effect of CO2 on calcium signals in RTN astrocytes. The authors need to discuss if they imaged the astrocytes near the ventral medullary surface that have been most associated with CO2 activation and control of breathing. The authors also should indicate if they have independent histochemical evidence that the gfaABC1D promoter indeed restricted GCaMP expression to glial cells. Also, it is important to explain how they can conclude, based on the strength of the calcium signal, whether or not the astrocytes could play a functional role – at what threshold would they decide the signal is big enough, and on what would they base that decision. Is there any evidence that glial calcium levels need to rise in order for glia to influence neighboring neurons? Please discuss these issues.

9) The authors argue (page 6, 2nd paragraph) that the adapting responses of RTN neurons to hypercapnia match adaptation of VE in the WBP trace. This issue should be addressed in more detail. The data for this are very unconvincing. Instead there is a gradually increasing level of respiratory activity after switching to 3% and 6% CO2. In Figure 2H the WBP Av ReSm shows a similar result, with gradually increasing amplitude as CO2 level increases. The authors should expand the "WBP Av ReSm" trace in the vertical direction to more clearly show the increase in amplitude with time, and a horizontal line should be drawn at the baseline level to allow the reader to visualize the change in amplitude more clearly. When this is done, there is a clear ramp up of VE after changing to 3% CO2, and an even steeper increase with 6% CO2. This is opposite of the clear adaptation of calcium level in the "Average Waveform" trace in Figure 2H.

10) The imaging results for pFL neurons suggest that these cells did not exhibit sustained expiratory-related oscillatory activity in response to hypercapnia that would be predicted based on the concept of a conditional expiratory oscillator in the pFL. The nature of the transient expiratory events described and whether they are in anyway related to the hypercapnic challenge in unclear from the results presented in Figure 8. This issue should be addressed more thoroughly in the manuscript.

11) The authors conclude (page 15) that: "Raphe neurons tended to be active during the entire CO2 stimulus and conceivably these neurons are likely to take over from RTN neurons under pathophysiological levels of CO2." The logic of how the authors come to this conclusion should be explained. The data are more supportive of the opposite conclusion. 42% of raphe neurons respond with a graded or sustained increase in firing in response to a rise in CO2 of as little as 3%, whereas most RTN neurons rapidly adapt (in only 1-2 minutes) to 3% CO2 and don't increase to 6% CO2 (a level that strongly stimulates breathing). The data are more consistent with the conclusion that at low levels of CO2, RTN neurons would have little or no effect, while raphe neurons would be expected to provide continuous activation of the respiratory network across the whole CO2 range studied. But there are potentially major problems with identifying the relevant RTN chemosensory neurons as indicated above that are thought to generate graded and sustained responses to CO2. This issue should be discussed more thoughtfully, particularly in the context of new experimental data that may be obtained regarding the molecular phenotypic identity of the RTN neurons analyzed.

12) There should be an explicit discussion of the role of synaptic connectivity, and the authors should discuss the relationship of their results with data from more reduced preparations (about which there is little discussion). Synaptic inputs were intact and all responses that were measured were due to a combination of intrinsic responses, synaptic input and glial modulation. Some of the text seems to imply that the responses are all intrinsic, but this issue should be clarified.

13) There are numerous reporting details that should be included. How many mice were used for cell counts and from how many sections? Did cells with particular response patterns (or sniffs) come from different mice or the same mouse? How were they distributed among the mice used? Which results were from synapsin and SERT mice?

14) In the text (p. 4) description of Figure 1, please correct the references to panels D-F. The actual figure panels do not correspond to the text descriptions. Similarly, in the figure legend, correctly match the references to panels D-G. Please also check the y-axis labeling in panel D.

*Reviewer #1 (Recommendations for the authors):*

1) The authors have worked out the essential technical approaches required for deep brain fluorescence imaging of targeted regions in the medulla of freely behaving mice. The authors are attempting to draw functional conclusions with this imaging approach about chemosensory responses of neurons and glia in multiple important medullary regions that have been the focus of numerous studies. Understanding the details of how these types of cells respond in any of these regions in vivo, particularly in awake behaving animals, requires extensive work with this imaging approach for any region studied, beyond what the authors have presented. The observations presented here are nonetheless important for the field and provocative, but the major concern with this approach is whether the targeted regional neuronal and glial populations have been adequately sampled to reasonably understand the spatial and functional heterogeneity of the hypercapnic responses. There are important optical limitations as well as potential limitations associated with efficacy of cellular labeling with the vector constructs used. These are major caveats that need to be more thoroughly discussed in the context of technical limitations. The authors state that the GRIN lens placement was within the RTN (Discussion p. 10, para. 2) or terminated under the caudal pole of the facial nucleus (p. 6, para. 1) but the anatomical reconstructions consistently show the end of the lens in facial motor nucleus and in Methods (p. 17, para. 1), so as the authors discuss the coverage of the RTN populations may not be sufficient. The authors state that generally the lens was positioned ~300 µm above the regions to be imaged. Please provide information on the depth of field of the GRIN lens used and the typical field of view (including providing length scale bars on the cellular fluorescence images presented in the main text). Additional information on the viral transduction efficacy with the GCaMP constructs used would be helpful.

2) Another technical issue relates to the genetically encoded calcium sensors used for different experiments. The authors need to explain why GCaMP6s was used for calcium activity imaging of RTN and Raphe neurons, whereas GCaMP6f was used for imaging astrocytes and pFL neurons. For comparisons of calcium dynamics between regions it is important that the same sensor is utilized which was done so this is not an issue, but interpretation of what is being encoded by a particular sensor needs to take into account the sensor signaling kinetics. GCaMP6s kinetics produce long (many seconds) calcium signal decays after signal onset, which can for example contribute to features such as adaptation of the calcium dynamics imaged and would not accurately reflect neuronal activity profiles per se.

3) The imaging results for pFL neurons are of considerable interest since these cells did not exhibit sustained expiratory-related oscillatory activity in response to hypercapnia that would be predicted based on the concept of a conditional expiratory oscillator in the pFL. The nature of the transient expiratory events described and whether they are in anyway related to the hypercapnic challenge in unclear from the results presented in Figure 8. This issue needs to be addressed more thoroughly in the manuscript.

4) In the text (p. 4) description of Figure 1, please correct the references to panels D-F. The actual figure panels do not correspond to the text descriptions. Similarly, in the figure legend, correctly match the references to panels D-G. Please also check the y-axis labeling in panel D.

*Reviewer #2 (Recommendations for the authors):*

The authors should provide more discussion of the evidence supporting the conclusion that a high percentage of the neurons they recorded from were in the RTN and were Phox2b+/Neuromedin B+/vGluT2^+^.

Page 6, 2nd paragraph: The authors argue that the adapting responses of RTN neurons to hypercapnia match adaptation of VE in the WBP trace. The data for this are very unconvincing. Instead there is a gradually increasing level of respiratory activity after switching to 3% and 6% CO2. In Figure 2H the WBP Av ReSm shows a similar result, with gradually increasing amplitude as CO2 level increases. The authors should expand the "WBP Av ReSm" trace in the vertical direction to more clearly show the increase in amplitude with time, and a horizontal line should be drawn at the baseline level to allow the reader to visualize the change in amplitude more clearly. When this is done, there is a clear ramp up of VE after changing to 3% CO2, and an even steeper increase with 6% CO2. This is opposite of the clear adaptation of calcium level in the "Average Waveform" trace in Figure 2H.

None of the other WBP traces appear to show any evidence of adaptation. To better illustrate if there is adaptation, or conversely if there is a graded increase, the authors should include more examples of rectified and smoothed WBP traces (or VE) with expanded vertical scales and horizontal lines marking the baseline level.

The traces in Figures4H and 4I should be expanded in the y axis to better visualize the increase in calcium levels, and a horizontal line should be added denoting the baseline.

Page 8, 4th paragraph: The number of raphe glial cells with various types of responses should be stated.

*Reviewer #3 (Recommendations for the authors):*

There are numerous reporting details that should be included. How many mice were used for cell counts? and from how many sections? did cells with particular response patterns (or sniffs) come from different mice? or the same mouse? how were they distributed among the mice? which results were from synapsin and SERT mice?

[Editors’ note: further revisions were suggested prior to acceptance, as described below.]

Thank you for resubmitting your work entitled "Analyzing the brainstem circuits for respiratory chemosensitivity in freely moving mice" for further consideration by *eLife*. Your revised article has been evaluated by Catherine Dulac (Senior Editor) and a Reviewing Editor.

Your revisions have improved the manuscript, but some remaining critical issues need to be rigorously addressed in a more thoroughly revised manuscript, as outlined below:

Essential Revisions.

(1) The critical issue of identifying the recorded neurons as RTN neurons according to the current definition of these neurons based on their established molecular properties (particularly see Reviewer #3) remains a major problem. As indicated in the reviews, there are other approaches to address this issue. Despite the new neuromedin B immunostaining results, the authors have not implemented a strategy to establish that any particular GCaMP-imaged cell was an actual RTN neuron. Accordingly, the authors cannot directly relate the measured responses of individual neurons to any specific type of neuron in the region studied. The authors should employ an approach that defines the molecular identity of the neurons imaged. Without additional convincing neuronal labeling/identification information that RTN neurons (as currently defined in the literature) were imaged, the authors need to use different terminology for what they are calling "RTN neurons" (e.g., ventral/ventromedial parafacial neurons) to avoid confusion and misinterpretation by readers. This issue also needs to be thoroughly discussed in the manuscript.

(2) The authors should include a more thorough discussion of the problem of precisely interpreting what the measured calcium signals are encoding due to uncertain, possibly nonlinear, relationships between the signals recorded from various neurons and their neuronal firing patterns. This discussion should include the rationale for their selection of the GCaMP sensors used. The authors need to have a section at the beginning of the Discussion dealing with these technical aspects/limitations that may affect interpretations of their observations, including further addressing any potentially confounding effects of insufficient correction of signals for movement artifacts- an issue raised again in this review.

(3) The issue of the consistency of calcium dynamic response patterns for a given neuron with repeated measurements to establish whether a given neuron truly belongs to any particular group as classified. How the authors establish consistency by applying some specific metrics should be more thoroughly addressed. A more rigorous assessment of the response consistency in individual neurons would go a long way to address technical and interpretive problems (including those associated with movement artifacts).

(4) The authors need to rectify inconsistencies in the data presented and address questions about how they interpret some of their observations, as noted in the reviews.

(5) The authors should thoroughly address all other issues raised by the reviewers.

*Reviewer #1 (Recommendations for the authors):*

This revised paper provides supporting evidence demonstrating the technical application of deep endoscopic fluorescence imaging with head-mounted miniscopes and genetically encoded calcium sensors in awake behaving mice for multicellular calcium activity imaging from key medullary structures involved in respiratory control in freely behaving mice, which is novel and important for the respiratory neurobiology field. The authors address some of the major problems and concerns about characterizing multi-neuronal activity by dynamic calcium imaging within medullary regions (RTN, raphe magnus and pallidus nuclei, pFL) proposed to have important chemosensory functions for the regulation of respiratory responses and homeostatic control of breathing with hypercapnia.

They have revised the paper in a number of important aspects in response to the original reviews: (1) they have removed the data on astroglial calcium dynamics and now just focus on the neuronal datasets obtained from these regions; (2) they have addressed the technical issues related to the depth of field with the GRIN lens and miniscope system used, which is critical for understanding the adequacy of sampling calcium dynamics in the targeted neuronal populations; (3) they have provided additional information on the neuromedin B identity of the possibly imaged neurons, although there are still concerns about the specificity of the antibody labeling and how the molecular identity has been established for the imaged neurons; and (4) they have discussed more directly how their regional calcium imaging data may be reconciled with available information on electrophysiological behavior and its variability in different experimental approaches, including comparisons with measurements in their awake freely behaving mice vs measurements in anesthetized and in vitro cellular measurements.

*Reviewer #2 (Recommendations for the authors):*

This revised manuscript reports the results of GCaMP imaging of neurons from the RTN and raphe of unanesthetized mice using a mini-microscope to determine how hypercapnia affects neuronal calcium levels. The authors have made a number of major changes, including removing the data on glia.

There is a lot of potential in this approach to determine how neurons in these two regions respond to hypercapnia in vivo. However, as currently presented some errors have been introduced, there are some inconsistencies in the data, and there are new concerns about some artifacts contaminating some of the results. The authors also make some conclusions that are not supported by their data.

It is important for the authors to specifically state that for the two specific types of neurons of most interest here the authors cannot convert the size of the calcium signals measured to firing rate. The relationship between calcium levels and firing rate is unlikely to be linear and is likely to be different for the two neuron types. One or both neuron types may not increase calcium levels very much in response to an increase in firing rate. The relationship could only be determined by simultaneous imaging and extracellular recordings of spikes. The authors should discuss this issue.

Movement artifacts influencing the interpretation of results

There is serious concern that movement has confounded some of the data. The authors provide arguments against the role of movement in altering the results, but many neurons had to be excluded due to movement artifacts (Figure 1A), indicating it is sometimes a problem.

Line 155 – "[under anesthesia]…movement artifacts cannot be a potential confounding factor."

This statement is not true. Convulsive seizures involve a large amount of movement. This experiment was done without a paralytic agent, so it does not rule out movement as a cause of the calcium signal recorded in the neurons during seizures.

Figure 1 Figure Supplement 2 and 3

For several groups of neurons recorded from the same mice, the pattern of calcium transients is so nearly identical that it does not seem biologically plausible. For example, that is true for neurons from mouse 7 for the RTN EA group. It is also true for the top two traces for Figure 2H (both extracted from mouse 5 in Suppl figure2), and for mouse 6 for the Raphe I group. How can the authors verify that the calcium activity is due to an increase in firing rate in these examples, and not movement or other artifact shared across neurons recorded at the same time? It would be possible for neurons to have similar patterns of activity, but not to be essentially identical.

There are a number of inconsistencies in the data

Figure 1 Figure Supplement 2

The trace labeled "Average EA" (7th trace down from the top in 1st column) does not correspond with the individual traces shown, where there should be a transient increase in calcium upon exposure to CO2 instead of occurring midway through the 3% CO2 exposure.

Figure 2H – Why does the Average Waveform EA look different than the Average EA in Figure 1 Figure Supplement 2? Shouldn't they be the same?

There are several observations that are not consistent with the literature.

In Figure 2H, the VE trace shows adaptation in 3% CO2, but not in 6% CO2. In Figures 2H and 9C of the previous version of this paper, there was no adaptation at either CO2 level. The authors state on line 452 that adaptation to step changes in inspired CO2 is well known to occur in the literature, but they do not give a reference. I don't think that is true, with the existing literature instead showing a sustained increase in CO2 with continuous exposure to hypercapnia. The authors should cite literature if it exists.

Line 257 – "Only a minority of neurons, 13% (5/38), encoded the magnitude of hypercapnia."

The sizes of the responses were much less than expected based on previous literature. This should be mentioned.

Line 367 "…type 1 neurons … display an adapting response to acidification."

The data in Lazarenko et al., 2009 is very unconvincing for an adapting response, and therefore the analogy with the EA neurons is as well. Coupled with the fact that ventilation does not adapt in most literature in response to sustained hypercapnia, and other chemoreceptors do not adapt to CO2, I am not convinced that the EA neurons are important for chemoreception. There is also no clear mechanism for why neurons should adapt to 3% CO2 but not 6%.

Line 569 – "[raphe neurons] are likely to take over from RTN neurons under pathophysiological levels of CO2."

That conclusion is not consistent with the observation that nearly half of raphe neurons have a graded or sustained response to CO2 of only 3%. That is not a pathophysiological stimulus.

It should be pointed out that sampling of both groups of neurons was limited and the authors may have not recorded from all subtypes of neuron.

The authors left labels off of the axes of a number of figures, e.g. 3B, 4C (fR), 7G.

Figure 1 Figure Suppl 3 – The EA trace does not look convincingly adapting.

*Reviewer #3 (Recommendations for the authors):*

The authors have attempted to address the concerns from the previous reviews.

1. In dealing with the major issue of identifying the recorded neurons as RTN cells, the authors argue that the use of a PRSX8 promoter-driven viral approach was ineffective. But, this is not the only solution. An alternative is using a Phox2b-Cre mouse and a DIO virus for GCaMP delivery. These reagents are readily available and should be attempted if they want to say anything about RTN neurons.

2. The authors need to use different terminology for what they are calling RTN. The initial description of RTN neurons in cats was based on their projections to VRG (and a lesser extent, DRG; see Smith et al., 1989). Since then, the RTN cell group has been defined functionally and neurochemically with increasingly greater precision (e.g., Guyenet et al., 2019). In the current study, these authors have not convincingly demonstrated that the actual recorded cells meet any of the established criteria for being called RTN neurons. Therefore, they should follow the more general "parafacial" terminology to describe this region, and refer to these as "unidentified neurons located in the parafacial region."

3. There are issues remaining with respect to Nmb antibody validation and immunostaining. It is still the case that no experimental controls for the Nmb immunohistochemistry are provided in this paper.

Instead, two references are cited. In Yamagata et al. (*eLife* 2021), however, it appears that a different antibody was used (monoclonal antibody from Developmental Studies Hybridoma Bank) – rendering this an irrelevant citation. The Li et al. (Nature, 2014) citation indeed reports on the same polyclonal antibody used here, but they used it to detect Nmb-IR terminals in the preBötC (see Ext. Data Figure 2 in Li et al.) and not cell somata in the RTN. The problems of detecting neuropeptides in neuronal cell bodies by antibody staining are well recognized and the images provided are unconvincing. For example, there appears to be staining where one would expect to find few, if any, Nmb-expressing neurons – e.g., see upper right in current Figure 2D, just below the lens lesion. This is not to say that this Ab staining approach is impossible, but at least some validation is required. For example, the authors might combine IHC with FISH to demonstrate that the Nmb antibody specifically recognizes neurons that also express Nmb transcripts and, more importantly, that it does not non-specifically stain cells that do not express Nmb.

(It should be noted that, even with more appropriate validation, this strategy of post hoc identification will not be able to ensure that any particular GCaMP-imaged cell was an actual RTN neuron – i.e., no attempt is made to relate individual responsive neurons to the cells that were subsequently immunostained.)

4. The issue of response "patterns" remains. The post hoc grouping of different neurons selected for similar responses to use for averaging (Figure 1, Supp. 2) does not say anything about whether the response in an individual neuron represents its own characteristic "pattern." It is circular reasoning to select neurons based on a type of response to the stimulus, and then view the averaged data from those pre-selected neurons aligned to the stimulus as evidence for a consistent "pattern."

Moreover, the new anecdotal data presented of recordings from the same cell over multiple days (Figure 1, Supp. 5) with no analysis beyond the "eye test" is unconvincing and does not adequately address this issue. What precisely does it mean to say that "activity patterns were remarkably similar between separate recording sessions" when no quantification is provided to assess that professed similarity. Why not simply repeat the stimulus protocol and verify a consistent response pattern?

5. Aside from issues with the data mentioned above, it is worth pointing out that some of the interpretations presented in the Discussion are questionable and/or fail to adequately consider the existing literature. A few examples are provided below.

As mentioned in the previous review, the responses recorded (and any associated response properties such as graded, adapting, etc.) could represent either an intrinsic sensory response or a complex amalgam of the presumed intrinsic property with extrinsic inputs to the recorded neuron. Although the authors claim to have clarified this at various points in the text (e.g., one instance is noted on p. 13, when discussing CO2-inhibited raphe neurons), a reader is likely to get the overall sense from the presentation that the responses are mostly presumed to be intrinsic (e.g., with phrasing such as "cells encoded the level of inspired CO2.") An explicit acknowledgement of this interpretive confound, which is unavoidably inherent to studies that correlate neuronal activity with behavior, should be provided at the outset of the Discussion (preferably in a general standalone section that deals with the various experimental and interpretive limitations.)

The statements that: "The majority of evidence for the involvement of the RTN, and in particular the Phox2B+ neurons, in CO2 chemoreception comes from neonatal or young juvenile animals" and "much of prior evidence for RTN chemosensory responses depends heavily on recordings from anaesthetized animals" ignores strong evidence from the Guyenet group for RTN involvement in CO2 responses of unanesthetized adult rodents (e.g., Basting et al., 2015).

Further, the suggestion that a normal developmental feature is that "chemosensory response becomes less dependent on Phox2B+ neurons of the RTN by 3 months of age" is based on a study that examined adaptations following genetic RTN destruction – hardly a normal developing mouse – and it ignores the profound effects of acute RTN neuron ablation in adult rats on chemosensory responses (Souza et al., 2018). A more balanced interpretation of the present results that considers all the relevant literature is advisable.

As a final example, it would appear to be a stretch to relate E-A and E-G responses in vivo to two types of RTN neurons recorded in vitro, based apparently on a visual inspection ("careful examination") of a few example cells presented in figures from other papers in which those particular response characteristics were neither identified nor characterized. It does not seem appropriate to build a substantial and speculative part of the discussion on this subjective interpretation from those example cells.

[Editors’ note: further revisions were suggested prior to acceptance, as described below.]

Thank you for resubmitting your work entitled "Analyzing the brainstem circuits for respiratory chemosensitivity in freely moving mice" for further consideration by *eLife*. Your revised article has been evaluated by Catherine Dulac (Senior Editor) and a Reviewing Editor.

The manuscript has been improved but there are some remaining issues that need to be addressed, as outlined below:

Essential Revisions:

The reviewers appreciate that the authors have responded to many of the previous points of major concern raised. The authors' novel catalog and analyses of multicellular neuronal calcium dynamics in the freely behaving mouse during hypercapnia are a valuable contribution to the field, including by illustrating the feasibility and paving the way for further development/application of this imaging approach, while indicating its limitations. However, there are some concerns that should be further addressed.

1) There is still a serious concern that there is a problem with precisely interpreting what the measured calcium signals are encoding due to uncertain, possibly nonlinear, relationships between calcium imaging from various neurons and their firing rate/pattern. The authors respond to this criticism by saying: "… the dynamics of the Ca^2+^ signal are likely to reflect the dynamics of firing. We can be confident that when the firing rate increases, intracellular Ca^2+^ will also increase." While this has been shown for some types of neurons (especially hippocampal and neocortical neurons), the reviewers are not aware that it has been shown for RTN or raphe neurons, and the authors offer no evidence for that. There are many mechanisms by which calcium levels could become dissociated from the firing rate. For example, in some neurons calcium levels may be relatively insensitive to an increase in firing rate due to low calcium current density or a high level of calcium buffering. An increase in firing rate or intracellular acidosis could have a stimulatory effect on calcium extrusion or sequestration. The "adaptation" of calcium levels the authors point out may reflect augmentation of calcium regulation rather than adaptation of firing rate. The response/decay kinetics of GCaMP6s is also a problem in terms of encoding the temporal characteristics of neuronal firing (e.g., Dana et al. Nat Methods 16, 649-657, 2019). A disconnect between calcium levels and firing rate could also explain why some neurons did not have a linear response to a graded increase to CO2 – their firing rate may have increased linearly while calcium levels fell off. The authors say they "have never attempted to convert Ca^2+^ signals to firing rate," but they lead the reader to believe that their measurements reflect neuronal firing rate/patterns throughout the paper by describing their measurements of calcium fluorescence as "activity of neurons", "neuronal firing", "activity patterns", or "activity." Neurons were categorized as Inhibited, Excited, or Tonic. Neurons are described as being "silenced." All of these terms lead the reader to think that the measurements presented are reliable surrogates of neuronal electrophysiological activity. The authors state that "the use of Ca^2+^ as a proxy for activity is widely accepted." That doesn't make it right. The authors need to be more precise in their terminology throughout the text. They are measuring calcium signals, not necessarily firing rate/activity patterns.

2) The authors' responses have not adequately addressed the major and overarching concerns that: a) the recordings were ultimately from unidentified RTN neurons even if we allow that they were in the region of the RTN; and, b) we cannot know whether the recorded responses were a true characteristic CO2 response of the recorded neurons without repeated measurements in the same neurons (as opposed to some random fluorescence changes that were found represented among the population of recorded cells in different mice). The authors need to emphasize these problems more clearly in the manuscript.

3) Reviewer #3 still questions the specificity of the NMB immunostaining. While it is appreciated that the authors tried to match their immunolabeling with the Allen Brain Atlas, this is unconvincing and appears to be mostly comparisons of the type of non-specific labeling that is often seen in areas of high cell density (e.g., Figure 2, Figure Suppl. 1, Panel Ei, cortex; Panel Eii, piriform cortex). The most obvious example they provide of "real" strong in situ labeling from the Allen Atlas is in Panel Eii (in the hilus of the dentate gyrus?) – and in this case, the NMB immunostaining they show is not any stronger than the non-specific labeling noted above.

Other specific issues that should be addressed:

1) Abstract, lines 26-28. Given the authors' acknowledgements of the potential limitations of their calcium signal measurements in terms of encoding temporal patterns of neuronal activity, the statement that their "analysis revises understanding of chemosensory control in awake adult mouse" should be modified.

2) Introduction: "brain imaging techniques have been developed to allow recording of activity of defined cell populations in awake, freely-moving animals … require: the expression of genetically encoded Ca^2+^ indicators such as GCaMP6 in the relevant neurons …" This specific requirement was not met for the RTN since there was no "defined cell population" targeted with the GCaMP6. Consider rewording.

3) Page 5, lines 151-153: "When we were able to identify the same neurons their activity patterns were remarkably similar between separate recording sessions (Figure 1—figure supplement 6)." This cursory analysis, based on a few neurons, is not very convincing. The need for repeated measurements of the same neuron to be sure about the characteristic dynamic pattern of the calcium signal should be clearly stated in the limitations section in the Discussion.

4) Line 369: "are reproducible between recording sessions from a single mouse …" Change to "are reproducible across the imaged neuronal population between recording sessions from a single mouse" to clarify that these general activity patterns were observed, but not within the same individual neurons.

5) There seem to be two different definitions applied for the EA subtype of response. For one, they responded to the initial increase in inspired CO2 but did not maintain their activation, and for the other, they respond to 3% CO2 more robustly than 6% CO2. Are these the same?

6) Lines 447-448: "Chemosensory responses become less dependent on Phox2B+ neurons of the RTN by 3 months of age (Ramanantsoa et al., 2011) … should add "when those neurons are genetically ablated" or some such qualifier.

7) Line 515: This statement should include a reference to Cleary et al. (PMID: 34013884), in which CO2-inhibited SST-expressing interneurons were recorded in the parafacial (RTN) region. This paper should also be cited with reference to the diversity of neuronal subtypes in that region, many of which may have been sampled in the current GCaMP6 recordings.

8) Page 15, para. 1 and 2 and Figure 8. The calcium signal measurements presented in this paper and summarized in panel C do not directly provide information on neuronal firing patterns and associated synaptic interactions implied in these diagrams. Given the uncertainty, the authors should emphasize that the regional interactions postulated are based on what is generally proposed in the literature, and it is currently unknown if any specific type of neuron that the authors have classified from the calcium signals and represented in the diagram has the connections indicated. Some readers may find these diagrams excessively speculative.

9) Line 619: "The neuronal responses to CO2 were more heterogeneous in both the RTN and Raphe than would be expected from the prior literature" should be clarified to state that "The neuronal responses to CO2 were heterogeneous for unidentified neurons in the RTN and for serotonergic neurons in Raphe."

---

## [Author Response]

Essential revisions:1) The authors should discuss more the technical limitations of their imaging approach. A major concern is whether the targeted regional neuronal and glial populations have been adequately sampled to reasonably understand the spatial and functional heterogeneity of the hypercapnic responses. There are important optical limitations as well as potential limitations associated with efficacy of cellular labeling with the vector constructs used. These are major caveats that need to be more thoroughly discussed in the context of technical limitations. The authors state for example that the GRIN lens placement was within the RTN (Discussion p. 10, para. 2) or terminated under the caudal pole of the facial nucleus (p. 6, para. 1) but the anatomical reconstructions consistently show the end of the lens in facial motor nucleus and in Methods (p. 17, para. 1), so as the authors discuss the coverage of the RTN populations may not be sufficient. The authors state that generally the lens was positioned ~300 µm above the regions to be imaged. Please provide information on the depth of field of the GRIN lens used and the typical field of view (including providing length scale bars on the cellular fluorescence images presented in the main text). Additional information on the viral transduction efficacy with the GCaMP constructs used is also essential.

We thank the reviewers for raising these important issues.

Optical characteristics of GRIN lens

We appreciate that the optical characteristics are a key point in the interpretation of our data.

The GRIN lens has a diameter of 600 µm and ample evidence in the literature shows that it has a focal plane that is ~300 µm below the lens as described in these references:

Resendez SL, Jennings JH, Ung RL, Namboodiri VM, Zhou ZC, Otis JM, et al. Visualization of cortical, subcortical and deep brain neural circuit dynamics during naturalistic mammalian behavior with head-mounted microscopes and chronically implanted lenses. Nat Protoc. 2016;11(3):566-97. Table-2

Jennings JH, Ung RL, Resendez SL, Stamatakis AM, Taylor JG, Huang J, et al. Visualizing hypothalamic network dynamics for appetitive and consummatory behaviors. Cell. 2015;160(3):516-27. Figure-4

However, given this is such a key point, we have imaged fluorescent beads to document this for the lenses we used in this study (now included as Figure 1 Figure Supplement 1). It is possible to adjust the focal plane of the lens by winding the camera back on its turret from the end of the lens (which moves the focal plane deeper into the tissue). We have therefore documented the focal plane and depth under two conditions -a turret setting of 0 turns (i.e. camera is against the end of the GRIN lens, and the focal plane is at its closest to the lens) and a maximum turret setting of 4 turns (i.e the camera is far from the end of the GRIN lens as it can be, and the focal plane is at its farthest). The imaging of the beads shows that at a turret setting of 0, the focal depth (in which the beads are sharply focussed) extends from 150-250 µm below the end of the lens, and for a turret setting of 4 this has been moved from 350-450 µm below the end of the lens. We kept a record of the turret settings for every recording and have added this into the data description and this gives the approximate depth of focus for every image. The text on pp 3 has been modified to explain this.

Even though anatomical reconstructions show the end of the lens in facial motor nucleus, the lens extends on either end of the shown anatomical reconstruction and covers the caudal pole of the facial nucleus, and the focal plane will be ventral to the facial nucleus. In the figure panels where we show lens placement we now include a box which shows the range of focal depths in which we expect to image cells. Our additional analysis suggests that in most cases we were indeed able to image from the ventral surface of the medulla.

Field of view was same in all the recordings- "width" 1440 and "height" 1080 pixels. We have added scale bars as requested to the GRIN lens images.

Transduction efficacy

We have addressed this by giving Venn diagrams for GCaMP expression versus different markers in Figures2, 5 and 9. We show a low percentage of ChAT (9/66), and a high percentage of NMB (40/93) and TPH (57/83) cells transduced by GCaMP.

Additionally, the numbers of cells that we recorded (i.e. within the imaging field) were as follows:

9 mice for RTN with hSyn promoter 122 neurons transduced ~14 neurons/mouse

6 mice for Raphe with hSyn or SERT promoter 41 neurons transduced ~7 neurons/mouse

3 mice pF_L_ with hSyn promoter, 31 neurons transduced ~10 neurons/mouse

2) Cell identification is another technical issue that needs to be addressed. It is critically important that the authors provide more convincing additional experimental data on the molecular identity of RTN neurons imaged to help with the interpretation of the results on this component. This should be done by employing an experimental strategy that allows for cell-type specific expression of GCaMP in RTN neurons (e.g., by using a Phox2b promoter), and/or by providing convincing additional quantification verifying that a high percentage of the GCaMP-expressing neurons imaged in the RTN region also express NMB and VGLUT2. For astrocytes and serotonin neurons, an attempt was made to restrict GCaMP expression to the targeted cell groups by using viruses including GFAP (gfaABC1D) or SERT promoters, but a non-specific neuronal promoter (synapsin) was used to drive GCaMP expression in RTN neurons or the pFL, with the obvious problem that the recordings were made from unidentified cells. Granted there is no specific marker for the pFL, but this approach is not justified for the RTN neurons. The molecular phenotype of CO2-sensitive RTN neurons has been well-established, and viral vectors using a Phox2b promoter have been employed for many years to preferentially target those RTN neurons. There are not a lot of RTN neurons (maybe 700-800 in mice) and there are many other neurons in the same region that likely were transduced with GCaMP and may account for the measured responses to CO2.

The PRSX8 promoter can only be used with lentiviral (LV) vectors. Unlike AAV vectors which are expressed in cells without causing cellular insults, LVs integrate into the cellular genome and can have off target effects. While we have had great success with the AAV approach, our use of LVs and the PRSX8 promoter to drive GCaMP expression in 10 mice on 3 separate occasions has not been successful. This strategy did indeed transduce NMB+ neurons. However, the GCaMP fluorescence was unvarying to any stimulus. We think this an artefact as these cells did not respond to KA injection. Thus, for unknown technical reasons this approach has not worked.

As the PRSX8 approach did not work we performed further experiments with hSyn-driven transduction and use NMB immuno-labelling to verify correct placement of lens and the presence of these cells in the field of view. These additional recordings replicated our prior findings, provided evidence for some more neurons with graded CO_2_ responses, provided evidence for a further subclass of CO_2_ non-coding neurons that exhibit Ca^2+^ signals related to variability in the breathing traces. These additional experiments have also allowed us to document more thoroughly how anaesthesia alters the responses of these neurons. We now have 98 neurons included. Furthermore, the proportion of NMB+ neurons transduced was approx. 43%.

3) Information about specificity of the antibodies used should be provided, which is particularly important for the immunostaining for NMB and VGLUT2 used to suggest that some of the GCaMP-expressing parafacial cells were bona fide RTN neurons. There are no controls provided for the specificity of the antibody staining, and no references attesting to the quality of the antibodies, which should be standard practice (see Rhodes and Trimmer, J Neuroscience 2006; Saper, JCN, 2005; Saper and Sawchenko, ibid, 2003). Even if the staining were validated as specific for the relevant antigens, demonstrating that some of the GCaMP-expressing neurons were VGLUT2- and NMB-positive RTN neurons does not guarantee that the recorded cells were also RTN chemosensory neurons. This important caveat needs to be amplified in the text with appropriate disclaimers.

We thank the reviewers for raising this point. The antibodies we used are those that have been used by others and validated. We give the details and references in the Methods:

Goat anti-choline acetyltransferase (AB144P); Millipore, Burlington, MA, USA- PMIDs- 33013329, 25723967, 25602013

Rabbit anti-tryptophan hydroxylase (ABN60); Σ-Aldrich, St Louis, MO, USA- PMIDs- 32332079, 25732261, 28821671

Rabbit anti-Neuromedin-B (SAB1301059); SigmaAldrich, St Louis, MO, USA- PMID- 26855425, 33393903

We agree that it is important to add caveats to our ability to identify the actual cells that we recorded from. However, the RTN contains a multitude of cells, and that will include chemosensory and non-chemosensory neurons. We have unbiasedly report on the activity of all neurons in the nucleus. We have been careful to avoid stating that we only record from “chemosensory” RTN neurons -clearly some of the recorded neurons are insensitive to the chemosensory stimuli.

4) The authors need to explain more their interpretations of the measured GCaMP signals. For various neurons there may be a nonlinear relationship between theses signals and neuronal firing. For glia it is not clear that "activation" by hypercapnia would always cause a change in intracellular calcium, and yet signaling pathways could still be engaged that influence neighboring neurons. In addition, the authors need to explain why GCaMP6s was used for calcium activity imaging of RTN and Raphe neurons, whereas GCaMP6f was used for imaging astrocytes and pFL neurons. For comparisons of calcium dynamics between regions it is important that the same sensor is utilized which was done for the RTN region and Raphe neurons, but interpretation of what is being encoded by a particular sensor needs to take into account the sensor signaling kinetics. GCaMP6s kinetics produce long (many seconds) calcium signal decays after signal onset, which can for example contribute to features such as adaptation of the calcium dynamics imaged and would not accurately reflect neuronal activity profiles per se.

We agree that the relationship between intracellular Ca^2+^ accumulation and neuronal firing may be nonlinear. Nevertheless, it is well established that neuronal firing does indeed lead to accumulation of intracellular Ca^2+^ and has been used as a proxy of neural activity in many imaging studies (and underlies other techniques such as cFos expression to assess neural activation).

The choice of GCaMP6s and GCaMP6f was predicated on their sensitivity and speed of kinetics (PMID: 23868258). We used GCaMP6f in the pF_L_ to attempt to resolve their firing during active expiration, whilst this may be a different fluorophore, we do not make any comparisons between the pF_L_ and any other site. However, we used GCaMP6s in both the RTN and the Raphe, so these measurements are directly comparable. GCaMP6s being a slower responding GCaMP is more sensitive to smaller signals and seemed the correct choice for our studies. The inability to resolve individual spikes does not seem to us a shortcoming, as the Ca^2+^ will accumulate and likely show the envelope of firing in the neurons. We note that the graded responses that we observe in the RTN neurons are remarkably similar to the chemosensory response of a Phox2b+ neuron documented in Stornetta et al. 2006.

As the paper is now considerably longer due to the revisions and extra documentation, we have decided to focus only on the neuronal recordings and have taken out the data on glial cells. This has the benefit of allowing us to present the neuronal data much more fully and simplifies the discussion of our results.

5) The authors should discuss more how they adequately ruled out movement artifacts affecting the calcium signals. The authors provide some evidence that the measured Ca transients are independent of movement, but the GFP signal measured in different mice is not the best and adequate control for movement artifacts in other GCaMP-expressing mice. Inscopix miniscopes are capable of dual color imaging to detect a co-expressed fluorescent marker in the same cell, which would be a better control.

With respect we think that our treatment of this has been very thorough. Dual wavelength ratiometric imaging is of course desirable but such microscopes were not available at the time we performed this study. With appropriate controls single wavelength imaging has been used to study neural activity successfully in many brain areas and we believe that our work conforms to the best practice published in the field. We have emphasized that we have used a consistent experiment protocol and have used averaging to document consistent Ca^2+^ signals (see point below).

6) The calcium signals are interpreted by the authors to reflect a variety of cellular response patterns, but there is no indication of how the authors verified consistency of these patterns, and this important issue needs to be addressed. Did the authors apply the CO2 challenges in repeated or alternately ordered fashion (0-3-6, 0-6-3) to more convincingly verify a consistent response in each individual cell? This is important information to include. This approach would also aid in dissociating common CO2-associated response patterns from any more randomly associated movement artifacts, for example by the use of peristimulus averaging to tease out signal from noise.

We of course agree that a consistent paradigm of stimulation is critical to document reliable responses to hypercapnia. With respect, we point out that this is exactly what we did and that we provided all the records of all the recordings in Figure 1 Figure Supplement 2 which documents the systematic nature of our study and the reproducibility of what we observed across neurons and mice. Our paradigm was to go from 0 to 3 to 6 and back to 0% inspired CO_2_. This was chosen to minimise bleaching of the GCaMP and allow repeated imaging sessions of cells on different days. We also point out that we averaged the responses of each type of neuron aligned to hypercapnic stimulus, and for neurons that displayed activity related to breathing we averaged the Ca^2+^ signals triggered to the relevant features of the WBP trace (a form of peristimulus averaging). We have modified the text on pp4 to make this clearer.

In addition, we now document in Figure 1 Supplement 3 that, when we can identify the same neurons between recording sessions on different days, their activity patterns are remarkably similar (text on pp 5).

7) The authors suggest that tonic RTN neurons "provide tonic drive to the respiratory network." However, it is possible that these neurons are not respiratory. Just because they are in/near the RTN does not mean they are chemoreceptors. Please discuss more carefully.

We have carefully reviewed our wording and modified any ambiguities. However, the RTN is well known to provide tonic drive to the preBotC so this suggestion is hardly contentious. We have also been careful to avoid describing all the RTN neurons are chemosensitive. Like others (e.g. PMIDs 29873079, 31635852, 26968853 and 24107938) we found neurons apparently insensitive to changes in PCO_2­_. We have added some additional text to pp 15.

8) There was little effect of CO2 on calcium signals in RTN astrocytes. The authors need to discuss if they imaged the astrocytes near the ventral medullary surface that have been most associated with CO2 activation and control of breathing. The authors also should indicate if they have independent histochemical evidence that the gfaABC1D promoter indeed restricted GCaMP expression to glial cells. Also, it is important to explain how they can conclude, based on the strength of the calcium signal, whether or not the astrocytes could play a functional role – at what threshold would they decide the signal is big enough, and on what would they base that decision. Is there any evidence that glial calcium levels need to rise in order for glia to influence neighboring neurons? Please discuss these issues.

As mentioned above we have decided to take out the glial recordings so that this paper focusses more clearly on neuronal responses. We shall publish the glial responses in a different paper.

9) The authors argue (page 6, 2nd paragraph) that the adapting responses of RTN neurons to hypercapnia match adaptation of VE in the WBP trace. This issue should be addressed in more detail. The data for this are very unconvincing. Instead there is a gradually increasing level of respiratory activity after switching to 3% and 6% CO2. In Figure 2H the WBP Av ReSm shows a similar result, with gradually increasing amplitude as CO2 level increases. The authors should expand the "WBP Av ReSm" trace in the vertical direction to more clearly show the increase in amplitude with time, and a horizontal line should be drawn at the baseline level to allow the reader to visualize the change in amplitude more clearly. When this is done, there is a clear ramp up of VE after changing to 3% CO2, and an even steeper increase with 6% CO2. This is opposite of the clear adaptation of calcium level in the "Average Waveform" trace in Figure 2H.

We agree that we had not clearly documented this point. To rectify this, we have now provided for three mice, in which this phenomenon was particularly clear, a correlation plot (V_T_ vs F/F0) at the beginning of the 3% hypercapnia to show that there is a positive correlation between these two variables i.e. there is an adapting change in V_T_ that matches the Ca^2+^ signal. To aid interpretation of this plot we also provide an Excel worksheet that uses simple mathematical functions to model an adapting F/F0 response together with either a gradual increase in V_T_ or an adapting change in V_T_ plus a gradual increase. This is an interactive spreadsheet so the reader can alter the values of the parameters to explore further. The key point is that if there were no adapting change in V_T_, the correlation plot would have negative slope, whereas with an adapting change in V_T_ it gives a positive slope -which is what we observe in the real experiment.

10) The imaging results for pFL neurons suggest that these cells did not exhibit sustained expiratory-related oscillatory activity in response to hypercapnia that would be predicted based on the concept of a conditional expiratory oscillator in the pFL. The nature of the transient expiratory events described and whether they are in anyway related to the hypercapnic challenge in unclear from the results presented in Figure 8. This issue should be addressed more thoroughly in the manuscript.

We agree and have added more discussion of this on pp16. Most studies on this are from anaesthetised and vagotomised rodents, which elongates and enhances expiratory output. Data from pF_L_ neurons in unanaesthetised unvagotomised in situ preparation shows pF_L_ neurons only discharge in the final 20% of the expiratory period. In freely moving mice this equates to approximately only 20 ms, which may not be enough to give a sufficient signal using GRIN lens technology and the GCaMP6f construct. This technique does however still allow for recording of large expiratory events, which support the role of the pF_L_ as an expiratory oscillator. Therefore, this finding is still of great importance. We now discuss this in greater detail on pp16 and 17.

11) The authors conclude (page 15) that: "Raphe neurons tended to be active during the entire CO2 stimulus and conceivably these neurons are likely to take over from RTN neurons under pathophysiological levels of CO2." The logic of how the authors come to this conclusion should be explained. The data are more supportive of the opposite conclusion. 42% of raphe neurons respond with a graded or sustained increase in firing in response to a rise in CO2 of as little as 3%, whereas most RTN neurons rapidly adapt (in only 1-2 minutes) to 3% CO2 and don't increase to 6% CO2 (a level that strongly stimulates breathing). The data are more consistent with the conclusion that at low levels of CO2, RTN neurons would have little or no effect, while raphe neurons would be expected to provide continuous activation of the respiratory network across the whole CO2 range studied. But there are potentially major problems with identifying the relevant RTN chemosensory neurons as indicated above that are thought to generate graded and sustained responses to CO2. This issue should be discussed more thoughtfully, particularly in the context of new experimental data that may be obtained regarding the molecular phenotypic identity of the RTN neurons analyzed.

Thank you for raising this point. We have looked closely at the chemosensory responses documented for Phox2b neurons in the published literature. As mentioned above the time course of the Ca^2+^ signal (including its recovery back to baseline in return to 0%) of the graded neurons (E_G_) is remarkably similar to the response documented by Stornetta et al. 2006. We note that Phox2b+ chemosensory neurons have been subdivided into type 1 and type 2 categories based on the characteristics of their pH sensitivity. Interestingly type 1 neurons display an adapting response to acidification -this is clearly seen in Figure 6B1 of Lazarenko et al. 2009 doi: 10.1002/cne.22136, where the adaptation is particularly prominent on going from pH 7.3 to pH 7.0, and was also evident (but slower) with lesser acidification. The type 2 neurons (panel C1) show a sustained response. This adapting response in type 1 neurons is also evident in the isolated Phox2b neurons reported in Wang et al. 2013 J Neurosci doi: 10.1523/JNEUROSCI.5550-12.2013 e.g. Figure 2A. Although the acid stimulus is only 1 minute long, allowing for the time taken for the stimulus to wash on and wash off (about 30s), the firing appears to adapt.

We now discuss these points and suggest that the adapting and graded neuronal subtypes we observe could be equivalent to these previously described subtypes. In total we observe that ~20% of the neurons in the RTN exhibit a graded or adapting response, this matches reasonably well with the proportion of GCaMP6 transduced neurons that are also NMB+.

Our additional experiments have allowed us to document more fully the effect of anaesthesia on the chemoresponsiveness of RTN neurons. The over-riding result is that anaesthesia very effectively silences the vast majority of RTN neurons -both their baseline activity and their response to hypercapnia. We find that the chemosensory phenotype of the neurons that retain activity under anaesthesia changes between the awake and anesthetised state. For example, neurons that exhibit a graded chemosensory response in the anaesthetised state would be classified as “non-coding respiratory related” in the awake state. Neurons that exhibited a graded response when awake, became non-coding in the anaesthetized state. Adapting neurons in the awake state became non-coding or inhibited neurons in the anaesthetized state. We think there is very compelling evidence that some RTN neurons are intrinsically pH sensitive (e.g. the Wang et al. study on acutely isolated neurons). However, our data would suggest that caution in interpreting the activity patterns of these neurons in the presence of anaesthesia is warranted, especially as the type of anaesthetic agent and depth of anaesthesia are likely to be confounding variables.

12) There should be an explicit discussion of the role of synaptic connectivity, and the authors should discuss the relationship of their results with data from more reduced preparations (about which there is little discussion). Synaptic inputs were intact and all responses that were measured were due to a combination of intrinsic responses, synaptic input and glial modulation. Some of the text seems to imply that the responses are all intrinsic, but this issue should be clarified.

Thank you for pointing this out. We have clarified this matter at various points in the text.

13) There are numerous reporting details that should be included. How many mice were used for cell counts and from how many sections? Did cells with particular response patterns (or sniffs) come from different mice or the same mouse? How were they distributed among the mice used? Which results were from synapsin and SERT mice?

We have added these details to Figure 1 Supplement 2.

14) In the text (p. 4) description of Figure 1, please correct the references to panels D-F. The actual figure panels do not correspond to the text descriptions. Similarly, in the figure legend, correctly match the references to panels D-G. Please also check the y-axis labeling in panel D.

Now corrected.

Reviewer #1 (Recommendations for the authors):1) The authors have worked out the essential technical approaches required for deep brain fluorescence imaging of targeted regions in the medulla of freely behaving mice. The authors are attempting to draw functional conclusions with this imaging approach about chemosensory responses of neurons and glia in multiple important medullary regions that have been the focus of numerous studies. Understanding the details of how these types of cells respond in any of these regions in vivo, particularly in awake behaving animals, requires extensive work with this imaging approach for any region studied, beyond what the authors have presented. The observations presented here are nonetheless important for the field and provocative, but the major concern with this approach is whether the targeted regional neuronal and glial populations have been adequately sampled to reasonably understand the spatial and functional heterogeneity of the hypercapnic responses. There are important optical limitations as well as potential limitations associated with efficacy of cellular labeling with the vector constructs used. These are major caveats that need to be more thoroughly discussed in the context of technical limitations. The authors state that the GRIN lens placement was within the RTN (Discussion p. 10, para. 2) or terminated under the caudal pole of the facial nucleus (p. 6, para. 1) but the anatomical reconstructions consistently show the end of the lens in facial motor nucleus and in Methods (p. 17, para. 1), so as the authors discuss the coverage of the RTN populations may not be sufficient. The authors state that generally the lens was positioned ~300 µm above the regions to be imaged. Please provide information on the depth of field of the GRIN lens used and the typical field of view (including providing length scale bars on the cellular fluorescence images presented in the main text). Additional information on the viral transduction efficacy with the GCaMP constructs used would be helpful.

We have covered these important matters in Essential Revisions point 1 above, provided more data and revised the text.

2) Another technical issue relates to the genetically encoded calcium sensors used for different experiments. The authors need to explain why GCaMP6s was used for calcium activity imaging of RTN and Raphe neurons, whereas GCaMP6f was used for imaging astrocytes and pFL neurons. For comparisons of calcium dynamics between regions it is important that the same sensor is utilized which was done so this is not an issue, but interpretation of what is being encoded by a particular sensor needs to take into account the sensor signaling kinetics. GCaMP6s kinetics produce long (many seconds) calcium signal decays after signal onset, which can for example contribute to features such as adaptation of the calcium dynamics imaged and would not accurately reflect neuronal activity profiles per se.

We have covered these concerns in our response to Essential Revisions point 2 above.

3) The imaging results for pFL neurons are of considerable interest since these cells did not exhibit sustained expiratory-related oscillatory activity in response to hypercapnia that would be predicted based on the concept of a conditional expiratory oscillator in the pFL. The nature of the transient expiratory events described and whether they are in anyway related to the hypercapnic challenge in unclear from the results presented in Figure 8. This issue needs to be addressed more thoroughly in the manuscript.

We agree, the recent paper by Magalhaes et al. 2021 (PMID 34510468) shows firing during the expiratory phase in rat (in situ preparation). These are recordings have a very elongated expiratory period. They only show a smally flurry of bursts just before inspiration and our mice are breathing 16 times fast than this. This is probably too short lasting for us to resolve. We discuss this in the paper and in point 10 above.

4) In the text (p. 4) description of Figure 1, please correct the references to panels D-F. The actual figure panels do not correspond to the text descriptions. Similarly, in the figure legend, correctly match the references to panels D-G. Please also check the y-axis labeling in panel D.

Corrected

Reviewer #2 (Recommendations for the authors):The authors should provide more discussion of the evidence supporting the conclusion that a high percentage of the neurons they recorded from were in the RTN and were Phox2b+/Neuromedin B+/vGluT2^+^.

We have now done this see response to Essential Revisions point 1 above.

Page 6, 2nd paragraph: The authors argue that the adapting responses of RTN neurons to hypercapnia match adaptation of VE in the WBP trace. The data for this are very unconvincing. Instead there is a gradually increasing level of respiratory activity after switching to 3% and 6% CO2. In Figure 2H the WBP Av ReSm shows a similar result, with gradually increasing amplitude as CO2 level increases. The authors should expand the "WBP Av ReSm" trace in the vertical direction to more clearly show the increase in amplitude with time, and a horizontal line should be drawn at the baseline level to allow the reader to visualize the change in amplitude more clearly. When this is done, there is a clear ramp up of VE after changing to 3% CO2, and an even steeper increase with 6% CO2. This is opposite of the clear adaptation of calcium level in the "Average Waveform" trace in Figure 2H.

We agree that we did not clearly present the evidence. We have now performed further analysis to thoroughly document the adapting response in the plethysmography trace and correlate it with the F/F0 trace from the adapting neurons. We show this correlation in Figure 2J for the E_A_ neurons in three mice. See Essential Revisions point 9.

We agree that V_E_ shows a ramp upwards, but this is in addition to a transient enhancement at the very beginning of hypercapnia.

None of the other WBP traces appear to show any evidence of adaptation. To better illustrate if there is adaptation, or conversely if there is a graded increase, the authors should include more examples of rectified and smoothed WBP traces (or VE) with expanded vertical scales and horizontal lines marking the baseline level.

We have revised Figure 2 to make this clearer.

The traces in Figures4H and 4I should be expanded in the y axis to better visualize the increase in calcium levels, and a horizontal line should be added denoting the baseline.

This has been done.

Page 8, 4th paragraph: The number of raphe glial cells with various types of responses should be stated.

The glial data has been removed now. See response to Essential Revisions point 8 above.

Reviewer #3 (Recommendations for the authors):There are numerous reporting details that should be included. How many mice were used for cell counts? and from how many sections? did cells with particular response patterns (or sniffs) come from different mice? or the same mouse? how were they distributed among the mice? which results were from synapsin and SERT mice?

We have provided more details on these points most notably in Figure 1 Supplement 2 and in the relevant figure legends.

[Editors’ note: further revisions were suggested prior to acceptance, as described below.]

Essential Revisions.(1) The critical issue of identifying the recorded neurons as RTN neurons according to the current definition of these neurons based on their established molecular properties (particularly see Reviewer #3) remains a major problem. As indicated in the reviews, there are other approaches to address this issue. Despite the new neuromedin B immunostaining results, the authors have not implemented a strategy to establish that any particular GCaMP-imaged cell was an actual RTN neuron. Accordingly, the authors cannot directly relate the measured responses of individual neurons to any specific type of neuron in the region studied. The authors should employ an approach that defines the molecular identity of the neurons imaged. Without additional convincing neuronal labeling/identification information that RTN neurons (as currently defined in the literature) were imaged, the authors need to use different terminology for what they are calling "RTN neurons" (e.g., ventral/ventromedial parafacial neurons) to avoid confusion and misinterpretation by readers. This issue also needs to be thoroughly discussed in the manuscript.

In the interests of transparency, we have revised the text to make it clear that we cannot identify the neurons to a specific subtype within the RTN.

However, we disagree that we are not using the correct nomenclature. The neurons are in the correct location underneath the caudal portion of the facial nucleus, in an area that contains NMB labelled neurons. The RTN, as implied by its name, is a nucleus, and nuclei consist of multiple neuronal subpopulations. It is inappropriate to define the RTN as a single anatomically defined neuronal subtype, as is being requested. Furthermore, calling this “ventral/ventromedial parafacial” is more likely to confuse readers. We have responded to this point more fully in point 2 of reviewer 3.

(2) The authors should include a more thorough discussion of the problem of precisely interpreting what the measured calcium signals are encoding due to uncertain, possibly nonlinear, relationships between the signals recorded from various neurons and their neuronal firing patterns. This discussion should include the rationale for their selection of the GCaMP sensors used. The authors need to have a section at the beginning of the Discussion dealing with these technical aspects/limitations that may affect interpretations of their observations, including further addressing any potentially confounding effects of insufficient correction of signals for movement artifacts- an issue raised again in this review.

We have added a section to the beginning of the Discussion to address these points.

(3) The issue of the consistency of calcium dynamic response patterns for a given neuron with repeated measurements to establish whether a given neuron truly belongs to any particular group as classified. How the authors establish consistency by applying some specific metrics should be more thoroughly addressed. A more rigorous assessment of the response consistency in individual neurons would go a long way to address technical and interpretive problems (including those associated with movement artifacts).

We have repeated recordings on separate days and in separate mice. Although for the most part we cannot identify the same neuron from day to day, we see very consistent patterns of responses across recording sessions, we now include these data in Figure 1 Figure Supplement 5. We explain the technical difficulties of identifying the *same* neuron between recording sessions on different days (possible in a few cases and presented in Figure 1 Figure Supplement 6), and why the possible issue of GCaMP6 bleaching constrained use to a single recording session per day. Our reasoning has been laid out in the response to Reviewer 2.

To justify the results of our classification further, we have plotted the change in Ca^2+^ response from baseline to 6% CO_2_ versus the response to 3% inspired CO_2_. This simple two component analysis would predict that E_A_ neurons, as they have a stronger response to 3% than 6%, should fall below the x=y line on the graph. Conversely, E_G_ neurons as they have a stronger response to 6% than 3%, should fall above the x=y line. E_NC_ neurons as their activity does not change with CO_2_ should be clustered around the origin, and I neurons should fall in the negative quadrant of the graph. We include a new figure supplement to demonstrate that this analysis shows our groupings follow the predicted pattern (Figure 1 Figure Supplement 7). Out of the 124 neurons from the RTN and Raphe included in our dataset, this further analysis caused us to reassign only 7 neurons (2 from RTN: 1 from I to NC and 1 from E_G_ to NC; and 5 from Raphe: 2 from E_s_ to E_G_ and 3 from E_s_ to E_A_).

(4) The authors need to rectify inconsistencies in the data presented and address questions about how they interpret some of their observations, as noted in the reviews.

We have corrected some errors pointed out to us that were inadvertently introduced during revision.

Reviewer #1 (Recommendations for the authors):This revised paper provides supporting evidence demonstrating the technical application of deep endoscopic fluorescence imaging with head-mounted miniscopes and genetically encoded calcium sensors in awake behaving mice for multicellular calcium activity imaging from key medullary structures involved in respiratory control in freely behaving mice, which is novel and important for the respiratory neurobiology field. The authors address some of the major problems and concerns about characterizing multi-neuronal activity by dynamic calcium imaging within medullary regions (RTN, raphe magnus and pallidus nuclei, pFL) proposed to have important chemosensory functions for the regulation of respiratory responses and homeostatic control of breathing with hypercapnia.They have revised the paper in a number of important aspects in response to the original reviews: (1) they have removed the data on astroglial calcium dynamics and now just focus on the neuronal datasets obtained from these regions; (2) they have addressed the technical issues related to the depth of field with the GRIN lens and miniscope system used, which is critical for understanding the adequacy of sampling calcium dynamics in the targeted neuronal populations; (3) they have provided additional information on the neuromedin B identity of the possibly imaged neurons, although there are still concerns about the specificity of the antibody labeling and how the molecular identity has been established for the imaged neurons; and (4) they have discussed more directly how their regional calcium imaging data may be reconciled with available information on electrophysiological behavior and its variability in different experimental approaches, including comparisons with measurements in their awake freely behaving mice vs measurements in anesthetized and in vitro cellular measurements.

We would like to thank Reviewer 1 for their positive stance on our revisions.

Reviewer #2 (Recommendations for the authors):This revised manuscript reports the results of GCaMP imaging of neurons from the RTN and raphe of unanesthetized mice using a mini-microscope to determine how hypercapnia affects neuronal calcium levels. The authors have made a number of major changes, including removing the data on glia.There is a lot of potential in this approach to determine how neurons in these two regions respond to hypercapnia in vivo. However, as currently presented some errors have been introduced, there are some inconsistencies in the data, and there are new concerns about some artifacts contaminating some of the results. The authors also make some conclusions that are not supported by their data.It is important for the authors to specifically state that for the two specific types of neurons of most interest here the authors cannot convert the size of the calcium signals measured to firing rate. The relationship between calcium levels and firing rate is unlikely to be linear and is likely to be different for the two neuron types. One or both neuron types may not increase calcium levels very much in response to an increase in firing rate. The relationship could only be determined by simultaneous imaging and extracellular recordings of spikes. The authors should discuss this issue.

We agree and have never attempted to convert the Ca^2+^ signal to firing rate in any version of the paper. Nevertheless, the dynamics of the Ca^2+^ signal are likely to reflect the dynamics of firing. We can be confident that when the firing rate increases, intracellular Ca^2+^ will also increase. The precise relationship will have distortions introduced by the slower nature of the Ca^2+^ signal and the dynamics of intracellular buffering. Nevertheless, the use of Ca^2+^ as a proxy for activity is widely accepted.

Movement artifacts influencing the interpretation of resultsThere is serious concern that movement has confounded some of the data. The authors provide arguments against the role of movement in altering the results, but many neurons had to be excluded due to movement artifacts (Figure 1A), indicating it is sometimes a problem.

These data were presented in first version of the manuscript. For a neuron to be included in the analysis it had to adhere to the rules set out in the methods. In some instances, we were further able to clean up the recordings by removing the background subtraction. If the cells were not able to meet our criteria for being free of movement and we were unable to remove movement through background subtraction the cells were excluded from further analysis. We provide recordings of every single recorded neuron and movies of each subtype, so that the reader may make their own judgement.

Line 155 – "[under anesthesia]…movement artifacts cannot be a potential confounding factor."This statement is not true. Convulsive seizures involve a large amount of movement. This experiment was done without a paralytic agent, so it does not rule out movement as a cause of the calcium signal recorded in the neurons during seizures.

The way we performed the experiments does indeed rule out movement as a cause of the signals. We stated in the methods of the previous version paper that the dose of Kainate was 8 mg·kg-1 IP, titrated to avoid behavioural seizures, but sufficient to induce electrographic seizures in the absence of movement. We have emphasized and clarified this point in the revised text and now provide a supplementary video to document this.

Figure 1 Figure Supplement 2 and 3For several groups of neurons recorded from the same mice, the pattern of calcium transients is so nearly identical that it does not seem biologically plausible. For example, that is true for neurons from mouse 7 for the RTN EA group. It is also true for the top two traces for Figure 2H (both extracted from mouse 5 in Suppl figure2), and for mouse 6 for the Raphe I group. How can the authors verify that the calcium activity is due to an increase in firing rate in these examples, and not movement or other artifact shared across neurons recorded at the same time? It would be possible for neurons to have similar patterns of activity, but not to be essentially identical.

We disagree with this comment. The response types are consistent across mice. For this to be a movement artefact, every mouse would have to move and generate this proposed artefact in the same way. This seems exceedingly unlikely and the simpler hypothesis that the Ca^2+^ responses reflect the biological responses of these neurons to hypercapnia seems far more probable to us. Were movement artefact to be a confounding factor, then all neurons from a single session would show the same profile. However, we show that multiple functional types were recorded from the same session. We have provided movies of all the neuronal types so readers can see the signals for themselves and judge accordingly.

We agree that the responses are very similar but do not see the problem -hypercapnia is a global stimulus and will thus evoke a global response, likely to be very similar in all responding neurons. In principle, synchrony may be additionally enhanced by synaptic and electrical coupling between neurons. Finally, using GRIN lens technology in vivo, several publications show neuronal recordings with near identical synchronous pattern/activity e.g PMIDs 35110409 (Video S1, Figure 2), 31184589.

There are a number of inconsistencies in the dataFigure 1 Figure Supplement 2The trace labeled "Average EA" (7th trace down from the top in 1st column) does not correspond with the individual traces shown, where there should be a transient increase in calcium upon exposure to CO2 instead of occurring midway through the 3% CO2 exposure.

This was a mistake introduced during revision where we changed the timescale of the individual traces to make them clearer but omitted to do the same for the average. Thank you for pointing this out and we have now corrected this.

Figure 2H – Why does the Average Waveform EA look different than the Average EA in Figure 1 Figure Supplement 2? Shouldn't they be the same?

Yes they should -once again our thanks, and we have corrected this

There are several observations that are not consistent with the literature.In Figure 2H, the VE trace shows adaptation in 3% CO2, but not in 6% CO2. In Figures 2H and 9C of the previous version of this paper, there was no adaptation at either CO2 level.

Figures 2H and 9C of the original paper were an average of the integrated and smoothed plethysmographic waveform over all mice. As the reviewers were unconvinced by this and asked us to document this point more clearly, we have replaced it in panel H of the revised Figure 2 by the V_E_ record for the mouse from which the neurons of this panel were recorded to facilitate direct comparison. This shows the adaptation more clearly and we have done the same for the V_T_ traces in three more individual mice in panel J. The reason for showing individual mice is that the extent of adaptation to the initial stimulus varies from mouse to mouse and an average over all mice obscures this.

The authors state on line 452 that adaptation to step changes in inspired CO2 is well known to occur in the literature, but they do not give a reference. I don't think that is true, with the existing literature instead showing a sustained increase in CO2 with continuous exposure to hypercapnia. The authors should cite literature if it exists.

We seem to be talking at cross purposes. We of course agree that there is a sustained increase in breathing with hypercapnia. However, superimposed on this is a transient adapting response -this is easy to see from the recordings we present in the paper. The vast body of literature that we have examined never shows measurements across the transition from normocapnia to hypercapnia (90 publications appeared in a PubMed search with the keywords: mice, hypercapnia and tidal volume). There is a simple reason for this -over the transition period, ventilation increases to a peak that then falls back somewhat to a stable value (still above that of normocapnia). This is simply an undocumented well-known fact of the field. Clearly there is a sustained ventilatory response to hypercapnia as measured at the end of the episode, but there is also an adapting response evident over the first minute that is superimposed on top of this sustained response. We refer the referee to panel 2H and J that show this adapting response in 4 different mice. Typically, the last minute of a hypercapnic episode is analysed to avoid this adapting period around the transition.

With regard to the literature, we have only managed to find one paper out of 90 that actually documents the transition and indeed shows an adapting response on top of the sustained response, see figure 1B in PMID: 27018763.

Line 257 – "Only a minority of neurons, 13% (5/38), encoded the magnitude of hypercapnia."The sizes of the responses were much less than expected based on previous literature. This should be mentioned.

The reviewer earlier suggested that we cannot convert Ca^2+^ signals to firing rate -we agree and have never attempted to do this. Given this, we do not understand how this point can be made, and for this reason will not discuss it further. Nevertheless, the available data shows that Phox2b+ chemosensory neurons do not exhibit very high firing rates to a strong stimulus (<12Hz). As these rates are low, we would not expect huge changes in the Ca2+.

The point that intrigues us is that the proportion of neurons that behave like a “classical Phox2b+ chemoreceptor neuron” as expected from the literature is very low. We extensively discuss the reasons for this on page 11.

Line 367 "…type 1 neurons … display an adapting response to acidification."The data in Lazarenko et al., 2009 is very unconvincing for an adapting response, and therefore the analogy with the EA neurons is as well.

We beg to differ. The Lazarenko et al. paper, being from an earlier era, does not provide access to their original data to settle this point. However, we take it as a matter of good faith that the authors showed representative data in this figure. Taking this as our starting point we have made further measurements from this paper:

Following a change in pH from 7.4 to 7.0, it takes a type 1 neuron approx. 23 ± 6s (n=5, mean ± SD) to reach its peak firing rate. Over the next 27s (from peak) the firing rate reduces to 52% of its peak. Following a change from pH 7.4 to 7.3, a type 1 neuron takes 39s to reach peak firing and then declines to 84% of this peak rate over 38s.

We additionally include annotated illustrations from the two papers to make our point (B1 shows a Type 1 neuron, and C1 and Type 2 neuron).

This adapting response in type 1 neurons is also evident in the isolated Phox2b+ neurons reported in Wang et al. 2013 J Neurosci doi: 10.1523/JNEUROSCI.5550-12.2013 e.g. Figure 2A. Although the acid stimulus is only 1 minute long, allowing for the time taken for the stimulus to wash on and wash off (about 30s), the firing appears to adapt.

What is interesting about the values that we have measured from these papers is that they are not dissimilar to the dynamics of the E_A_ neurons we describe. This similarity is evident even though they were obtained to a pH change in brain slices from P6-10 mice compared to in vivo recordings from adult mice and a response to CO_2_.

Coupled with the fact that ventilation does not adapt in most literature in response to sustained hypercapnia, and other chemoreceptors do not adapt to CO2, I am not convinced that the EA neurons are important for chemoreception.

There is clearly an adapting component to the ventilation response over the initial phase -this is documented in Figure 2H and J. We have now shown it correlates very nicely with the response pattern of the E_A_ neurons. This is only a correlation, but since the RTN is linked to the generation of ventilatory responses to hypercapnia, it seems plausible that the E_A_ neurons are involved in generating the adapting component of the response.

There is also no clear mechanism for why neurons should adapt to 3% CO2 but not 6%.

We agree, but inability to imagine a mechanism does not affect our observations.

Line 569 – "[raphe neurons] are likely to take over from RTN neurons under pathophysiological levels of CO2."That conclusion is not consistent with the observation that nearly half of raphe neurons have a graded or sustained response to CO2 of only 3%. That is not a pathophysiological stimulus.

Fair point -we have amended the text

It should be pointed out that sampling of both groups of neurons was limited and the authors may have not recorded from all subtypes of neuron.

This is possible, but we have included 98 neurons from 9 mice in RTN. We have extensively discussed this and will include this point in a new section of the Discussion.

The authors left labels off of the axes of a number of figures, e.g. 3B, 4C (fR), 7G.

We have rechecked the figures and added any missing labels.

Figure 1 Figure Suppl 3 – The EA trace does not look convincingly adapting.

We agree that this example does not look similar to the E_A_ neurons of the RTN, but it shows a larger response (in terms of frequency of transients) at the beginning of the 3% stimulus than in the 6% stimulus and we therefore classified as showing an adapting response. This is backed up by our 2-component analysis presented in Figure 1 Figure Supplement 7. This new analysis also led us to reclassify 3 E_S_ neurons as E_A_, hence there are now 4 neurons in this category.

Reviewer #3 (Recommendations for the authors):The authors have attempted to address the concerns from the previous reviews.1. In dealing with the major issue of identifying the recorded neurons as RTN cells, the authors argue that the use of a PRSX8 promoter-driven viral approach was ineffective. But, this is not the only solution. An alternative is using a Phox2b-Cre mouse and a DIO virus for GCaMP delivery. These reagents are readily available and should be attempted if they want to say anything about RTN neurons.

This mouse was not available at the time we did the experiments. We disagree that we cannot say anything about RTN neurons. We are clearly recording activity from neurons located within the RTN as evidenced by the landmarks, facial nucleus, pyramids, and SP-5, and the presence of NMB positive neurons within the region.

2. The authors need to use different terminology for what they are calling RTN. The initial description of RTN neurons in cats was based on their projections to VRG (and a lesser extent, DRG; see Smith et al., 1989). Since then, the RTN cell group has been defined functionally and neurochemically with increasingly greater precision (e.g., Guyenet et al., 2019). In the current study, these authors have not convincingly demonstrated that the actual recorded cells meet any of the established criteria for being called RTN neurons. Therefore, they should follow the more general "parafacial" terminology to describe this region, and refer to these as "unidentified neurons located in the parafacial region."

We disagree on this point. We are quite willing to accept that the precise identity of the neurons has not been established but using the term “parafacial” is likely to introduce further confusion. The retrotrapezoid nucleus (RTN) is a nucleus. The point being made by the reviewer conflates anatomical localisation with function. In common with many other nuclei in the brain, the RTN contains multiple sub-populations of neuron. Nearby nuclei such as the preBotC contain neuronal subpopulations with NK1R+ glutamatergic, Sst glutamatergic, and glycinergic chemical phenotypes. The Raphe contains neurons with serotonergic and GABAergic chemical phenotypes. To look further afield, the VTA contains dopaminergic, GABAergic and glutamatergic neurons.

As far as we are aware, no-one is seeking to redefine the preBotC along the lines implied by this comment. This line of reasoning would imply that as Sst neurons are output neurons and not rhythm generating neurons, they should no longer be considered preBotC neurons. Sensibly, the field refers to these neurons as a defined subtype within the preBotC. Similarly, no-one is promoting dopaminergic neurons as the “true” VTA. We strongly argue that the naming convention for neurons in the RTN should adhere to the well-established norm in neuroscience: Phox2b+/NMB+ chemosensory neurons are a subpopulation within the RTN, but do not by themselves constitute the RTN.

3. There are issues remaining with respect to Nmb antibody validation and immunostaining. It is still the case that no experimental controls for the Nmb immunohistochemistry are provided in this paper.Instead, two references are cited. In Yamagata et al. (eLife 2021), however, it appears that a different antibody was used (monoclonal antibody from Developmental Studies Hybridoma Bank) – rendering this an irrelevant citation. The Li et al. (Nature, 2014) citation indeed reports on the same polyclonal antibody used here, but they used it to detect Nmb-IR terminals in the preBötC (see Ext. Data Figure 2 in Li et al.) and not cell somata in the RTN. The problems of detecting neuropeptides in neuronal cell bodies by antibody staining are well recognized and the images provided are unconvincing. For example, there appears to be staining where one would expect to find few, if any, Nmb-expressing neurons – e.g., see upper right in current Figure 2D, just below the lens lesion. This is not to say that this Ab staining approach is impossible, but at least some validation is required. For example, the authors might combine IHC with FISH to demonstrate that the Nmb antibody specifically recognizes neurons that also express Nmb transcripts and, more importantly, that it does not non-specifically stain cells that do not express Nmb.

Souza et al. 2018 show NMB+ neurons in the locations mentioned by the reviewer “midline NMB” neurons Figure 1 of PMID: 29667182. Staining right next to the lens tract is likely to be artefactual.

We have removed the Yamagata reference, and now provide further validation of antibody specificity by comparing the NMB immunostaining with the pattern of ISH from the Allen Brain Atlas across several brain areas. This shows lack of staining in areas that do not express NMB (from ISH) and staining in areas where it would be expected from the ISH.

(It should be noted that, even with more appropriate validation, this strategy of post hoc identification will not be able to ensure that any particular GCaMP-imaged cell was an actual RTN neuron – i.e., no attempt is made to relate individual responsive neurons to the cells that were subsequently immunostained.)

We agree with this comment to the extent that no recording can be attributed to a particular subtype of RTN neuron. But we have been appropriately cautious in this regard. (1) We do not equate any response of a neuron to a specific subtype, but merely report the different functional subtypes. (2) In the Discussion, we attempt to reconcile our findings with the prior literature -for example, we “tentatively suggest” that the E_G_ subtypes may represent Phox2b+ neurons. It seems to us that this a perfectly acceptable point to make, and it would be odd if we did not put our findings in this wider context. We now state that sub-type specific drivers of GCaMP would be needed before any associations of functional subtypes with neurons of known chemical phenotype can be definitively made.

4. The issue of response "patterns" remains. The post hoc grouping of different neurons selected for similar responses to use for averaging (Figure 1, Supp. 2) does not say anything about whether the response in an individual neuron represents its own characteristic "pattern." It is circular reasoning to select neurons based on a type of response to the stimulus, and then view the averaged data from those pre-selected neurons aligned to the stimulus as evidence for a consistent "pattern."

We are puzzled by this comment. The reviewers specifically requested peristimulus averaging in their previous review to show that responses were repeatable and reproducible. We have chosen to show across mice that the same types of neuronal response are seen. As we show every recording in the dataset, we leave it up to readers to decide whether these response patterns are reproducible and consistent. The additional 2 component analysis we have performed (plotting the change in activity at 3% vs 6%) we have performed backs up our categorisation (see Figure 1 Figure Supplement 7).

Moreover, the new anecdotal data presented of recordings from the same cell over multiple days (Figure 1, Supp. 5) with no analysis beyond the "eye test" is unconvincing and does not adequately address this issue. What precisely does it mean to say that "activity patterns were remarkably similar between separate recording sessions" when no quantification is provided to assess that professed similarity. Why not simply repeat the stimulus protocol and verify a consistent response pattern?

We have revised Figure 1 Figure Supplement 6 (previously Figure Supplement 5) to bring out the similarity of the activity in the two sessions. We have now added a third panel in which the traces from different days are superimposed. The key issue is not how similar the activity of the cells is in each recording session (one would not expect them to be identical) but whether one would classify the patterns of neuronal activity differently across sessions. These neurons were classified as non-coding (NC) in both sessions.

Identification of the *same* neurons from recording session to recording session (on different days) was possible only in a few cases. The reasons for why we could not reliably identify the same neurons on different days are not completely clear. Most likely it was not possible to place the camera on the headstage in the exact same turret position. This would cause the focal plane to vary from session to session -this has been explained in the text on pp 5.

We chose to perform one recording session per day i.e. one complete protocol, rather than giving repeated stimuli. This is because the GCaMP6 significantly bleaches and we wanted to be able to repeat the recording sessions across multiple days. However, every mouse was recorded on more than one day. In our dataset (Figure 1 supplements) we only include recordings from one session (i.e. each mouse is represented only once, to avoid pseudoreplication, and all the neurons from a mouse are from the same session). We now make this explicitly clear. However, we additionally present data to show that the same types of neuronal responses to hypercapnia are present in multiple recording sessions from the same mouse, even although these cannot be attributed to the same neurons (Figure 1 Figure Supplement 5).

Our additional analysis in Figure 1 Supplement 7 shows that the different neuronal types we identified fall into distinct quantitative groupings.

5. Aside from issues with the data mentioned above, it is worth pointing out that some of the interpretations presented in the Discussion are questionable and/or fail to adequately consider the existing literature. A few examples are provided below.As mentioned in the previous review, the responses recorded (and any associated response properties such as graded, adapting, etc.) could represent either an intrinsic sensory response or a complex amalgam of the presumed intrinsic property with extrinsic inputs to the recorded neuron. Although the authors claim to have clarified this at various points in the text (e.g., one instance is noted on p. 13, when discussing CO2-inhibited raphe neurons), a reader is likely to get the overall sense from the presentation that the responses are mostly presumed to be intrinsic (e.g., with phrasing such as "cells encoded the level of inspired CO2.") An explicit acknowledgement of this interpretive confound, which is unavoidably inherent to studies that correlate neuronal activity with behavior, should be provided at the outset of the Discussion (preferably in a general standalone section that deals with the various experimental and interpretive limitations.)

We appreciate that bringing these matters to one place in the discussion would be helpful so have added a section to do this.

The statements that: "The majority of evidence for the involvement of the RTN, and in particular the Phox2B+ neurons, in CO2 chemoreception comes from neonatal or young juvenile animals" and "much of prior evidence for RTN chemosensory responses depends heavily on recordings from anaesthetized animals" ignores strong evidence from the Guyenet group for RTN involvement in CO2 responses of unanesthetized adult rodents (e.g., Basting et al., 2015).

Our phrasing may not have been sufficiently precise. The evidence for the activity of RTN neurons in response to chemosensory stimuli comes mainly from neonatal or young juvenile animals and recordings from anaesthetized animals. The paper quoted does not document recordings of activity of RTN neurons in the absence of anaesthesia. There are of course papers that show the effect of optogenetic stimulation in non-anaesthetized animals or the effects of various lesions. But these tell us nothing about the activity patterns, only that stimulation of RTN neurons is sufficient to give an enhancement of breathing, or that destruction of the RTN alters hypercapnic responses. Without knowing the activity of these neurons in the absence of anaesthesia, the precise significance of optogenetic stimulation or lesions is not known.

We have modified our wording to avoid any further confusion.

Further, the suggestion that a normal developmental feature is that "chemosensory response becomes less dependent on Phox2B+ neurons of the RTN by 3 months of age" is based on a study that examined adaptations following genetic RTN destruction – hardly a normal developing mouse – and it ignores the profound effects of acute RTN neuron ablation in adult rats on chemosensory responses (Souza et al., 2018). A more balanced interpretation of the present results that considers all the relevant literature is advisable.

We now include discussion of chemical lesioning experiments.

As a final example, it would appear to be a stretch to relate E-A and E-G responses in vivo to two types of RTN neurons recorded in vitro, based apparently on a visual inspection ("careful examination") of a few example cells presented in figures from other papers in which those particular response characteristics were neither identified nor characterized. It does not seem appropriate to build a substantial and speculative part of the discussion on this subjective interpretation from those example cells.

We disagree with this point. We take it on trust that the authors showed representative data and refer the reviewer to our response to reviewer 2. That they never identified or characterized this adapting response, may reflect the priorities that the authors had at the time. The emphasis in the Lazarenko paper was very much on documenting pH sensitivity and the differences in this characteristic between the type 1 and type 2 neurons. It is noteworthy that this same difference (adapting versus graded) was evident when the Phox2B+ neurons were isolated and was published some 4 years after the Lazarenko paper.

[Editors’ note: further revisions were suggested prior to acceptance, as described below.]

1) There is still a serious concern that there is a problem with precisely interpreting what the measured calcium signals are encoding due to uncertain, possibly nonlinear, relationships between calcium imaging from various neurons and their firing rate/pattern. The authors respond to this criticism by saying: "… the dynamics of the Ca^2+^ signal are likely to reflect the dynamics of firing. We can be confident that when the firing rate increases, intracellular Ca^2+^ will also increase." While this has been shown for some types of neurons (especially hippocampal and neocortical neurons), the reviewers are not aware that it has been shown for RTN or raphe neurons, and the authors offer no evidence for that.

It seems to us inadvisable to argue that RTN and raphe neurons have very different properties with respect to their Ca^2+^ dynamics from other neurons in many different regions of the brain (e.g. additionally hypothalamus, LDT/PPT). While this is possible in principle, there is no evidence to support this contention. In fact, the reverse is true. We note that the use of *c-fos* expression to identify neurons activated by chemosensory stimuli including those of the RTN has been widely published and accepted by the field (e.g., PMIDs 8173977; 26068853, Figures 1 and 4). As *c-fos* expression depends on activity dependent accumulation of intracellular Ca^2+^, these studies and data strongly support the Ca^2+^ dynamics of RTN and Raphe neurons as reflecting their activity patterns.

There are many mechanisms by which calcium levels could become dissociated from the firing rate. For example, in some neurons calcium levels may be relatively insensitive to an increase in firing rate due to low calcium current density or a high level of calcium buffering. An increase in firing rate or intracellular acidosis could have a stimulatory effect on calcium extrusion or sequestration. The "adaptation" of calcium levels the authors point out may reflect augmentation of calcium regulation rather than adaptation of firing rate.

Unfortunately, we do not think that these points have mechanistic support or evidence to back them:

1. As Ca^2+^ buffering is finite, a high buffering capacity might slow the onset of the increase of Ca^2+^ but would not prevent it from occurring.

2. Low Ca^2+^ current densities would mean that the magnitude of the signal might be small, even too small to be resolved. Cells that did not show appreciable Ca^2+^ signals in the baseline were excluded from our dataset as being uninterpretable.

3. Extrusion/sequestration rates for Ca^2+^ are much slower than entry rates. The electrochemical driving force for Ca^2+^ entry is very high, and channels have much faster transport rates than energy dependent transporters/exchangers. This is apparent in neurons throughout the nervous system where Ca^2+^ transients have a very fast onset (reflecting kinetics of entry through a variety of Ca^2+^ permeable channels -not just voltage-gated Ca^2+^ channels) and a much slower exponentially decaying phase that reflects sequestration/extrusion processes. This pattern is clearly evident in our own recordings, in which the decay time constant of Ca^2+^ transients is roughly 2s, and the 10-90% rise time of the transients is roughly 0.5s.

To specifically address the point about acidosis or some other feature of the hypercapnic stimulus changing Ca^2+^ dynamics and thus accounting for the adapting pattern, we have taken example recordings E_A_ neurons and tonic firing neurons and plotted individual Ca^2+^ transients from before, during and after the hypercapnic episodes. The kinetics of these transients are in effect real-time probes of the entry and extrusion/sequestration rates for Ca^2+^ in the cell. Neither the decay time constant nor the rise time of these signals in either of these cell types differs between these conditions. For the reviewers’ alternative hypothesis (that the apparent adapting pattern of Ca^2+^ activity is due to changes in influx and efflux rates caused by intracellular acidosis rather than reflecting neuronal firing patterns) to have merit, the decay time of the Ca^2+^ transients, which reflects buffering/sequestration/extrusion would have to be faster than the rise time of the transients during the hypercapnic period. Since neither change in an appreciable way, and the decay time constant remains much longer than the rise time, we can confidently discount this alternative explanation. We have added text to pp7 and pp10 and created a new Figure 2—figure supplement 2 to document this point.

The response/decay kinetics of GCaMP6s is also a problem in terms of encoding the temporal characteristics of neuronal firing (e.g., Dana et al. Nat Methods 16, 649-657, 2019). A disconnect between calcium levels and firing rate could also explain why some neurons did not have a linear response to a graded increase to CO2 – their firing rate may have increased linearly while calcium levels fell off.

The Dana et al. paper does not support this contention. Dana et al. show that while GCaMP6s cannot accurately resolve *individual* action potentials, it does nevertheless provide a smoothed envelope that reflects the firing dynamics. This is analogous to passing a high frequency electrical signal through a low pass filter. Dana et al. do not provide evidence of a disconnect between Ca^2+^ levels as measured by GCaMP6s and firing rates (they show quite the opposite: that the Ca^2+^ signal parallels the firing rates). Other papers agree on the difficulty of resolving single action potentials but also document the ability of GCaMP6 to accurately represent multispike firing dynamics in a variety of neurons and model systems (including zebrafish and *Drosophila*) – PMIDs 23868258, 33683198. As the responses of chemosensory neurons to hypercapnia are multispiking events, our use of GCaMP6 as an indicator of activity in these cells seems entirely appropriate given the current state of genetically encoded Ca^2+^ sensors.

The authors say they "have never attempted to convert Ca^2+^ signals to firing rate," but they lead the reader to believe that their measurements reflect neuronal firing rate/patterns throughout the paper by describing their measurements of calcium fluorescence as "activity of neurons", "neuronal firing", "activity patterns", or "activity." Neurons were categorized as Inhibited, Excited, or Tonic. Neurons are described as being "silenced." All of these terms lead the reader to think that the measurements presented are reliable surrogates of neuronal electrophysiological activity. The authors state that "the use of Ca^2+^ as a proxy for activity is widely accepted." That doesn't make it right. The authors need to be more precise in their terminology throughout the text. They are measuring calcium signals, not necessarily firing rate/activity patterns.

We have reviewed how we have presented the data, which is in line with the great consensus in the field that Ca^2+^ levels do reflect a reasonable proxy of firing rates and activity patterns. We note that throughout the Results section we refer to Ca^2+^ activity patterns (e.g., pp 5-6). We have extensively discussed the limitations in the Discussion. In the response above we have shown by reference to the literature and further analysis of our own data that these points of the reviewers lack evidential support. We have added some further references in the limitations section on pp10 to support these points.

2) The authors' responses have not adequately addressed the major and overarching concerns that: a) the recordings were ultimately from unidentified RTN neurons even if we allow that they were in the region of the RTN; and, b) we cannot know whether the recorded responses were a true characteristic CO2 response of the recorded neurons without repeated measurements in the same neurons (as opposed to some random fluorescence changes that were found represented among the population of recorded cells in different mice). The authors need to emphasize these problems more clearly in the manuscript.

We are pleased that the reviewers now accept that we recorded from neurons in the RTN.

In response to (b): from a statistical viewpoint, repeated recordings from a single cell would be regarded as pseudoreplication that tells nothing about the activity of a wider sample. We have used genuine replication by recording from separate neurons from separate mice. We have found consistency of activity patterns that is time-linked to the hypercapnic stimulus between recording sessions and between mice. This tells us that these cells exist and that our sample recordings represent the population.

To turn argument around, let us suppose that we were trying to claim that the observed responses were random and did not reflect any meaningful biological reality. The problem is that we would then have to “explain away” the following observations: the patterns are time locked to the stimulus; they are seen in multiple recording sessions from the same mouse; and they are seen in different mice recorded at different times. We would argue that it would be extremely unlikely that this alternative interpretation would gain significant support from our scientific peers.

In summary, we believe that we have already devoted considerable time and space to addressing these concerns by adding new data and figures in the two previous revisions. We also note that we have been extremely transparent and have shown all of our data, allowing readers to assess patterns in our data for themselves. Consequently, we are not aware of any further changes that could meaningfully address this point other than an additional sentence which we have added to the limitations section stating the desirability of repeated recordings.

3) Reviewer #3 still questions the specificity of the NMB immunostaining. While it is appreciated that the authors tried to match their immunolabeling with the Allen Brain Atlas, this is unconvincing and appears to be mostly comparisons of the type of non-specific labeling that is often seen in areas of high cell density (e.g., Figure 2, Figure Suppl. 1, Panel Ei, cortex; Panel Eii, piriform cortex). The most obvious example they provide of "real" strong in situ labeling from the Allen Atlas is in Panel Eii (in the hilus of the dentate gyrus?) – and in this case, the NMB immunostaining they show is not any stronger than the non-specific labeling noted above.

We have previously found the Allen Brain Atlas to be quite accurate in its localisation of a number of other markers and have no reason to believe that it does not represent NMB expression accurately. Most relevant for our study is that the neurons stained in the RTN with this antibody have the positions and cell body shapes and punctate staining pattern that have been previously described with use of FISH (PMID 29066557). This suggests that in this area, at least, it is staining the correct cells. As we are not using this antibody to describe a new population of cells but to confirm the location of the lens and that we are indeed imaging from the RTN, in conjunction with other landmarks and ChAT staining we suggest that this level of verification is adequate.

Other specific issues that should be addressed:1) Abstract, lines 26-28. Given the authors' acknowledgements of the potential limitations of their calcium signal measurements in terms of encoding temporal patterns of neuronal activity, the statement that their "analysis revises understanding of chemosensory control in awake adult mouse" should be modified.

As we argue above Ca^2+^ imaging does give an accurate picture of the activity envelope even if it cannot resolve individual spikes, so we think that this wording is appropriate.

2) Introduction: "brain imaging techniques have been developed to allow recording of activity of defined cell populations in awake, freely-moving animals … require: the expression of genetically encoded Ca^2+^ indicators such as GCaMP6 in the relevant neurons …" This specific requirement was not met for the RTN since there was no "defined cell population" targeted with the GCaMP6. Consider rewording.

As the reviewers agreed in point 2 above, we were recording from neurons in the RTN. We have made it clear at many points that we used the hSyn promoter and we would argue that we are indeed recording from the relevant neurons, even if we have not identified the subtype.

3) Page 5, lines 151-153: "When we were able to identify the same neurons their activity patterns were remarkably similar between separate recording sessions (Figure 1—figure supplement 6)." This cursory analysis, based on a few neurons, is not very convincing. The need for repeated measurements of the same neuron to be sure about the characteristic dynamic pattern of the calcium signal should be clearly stated in the limitations section in the Discussion.

We have added a sentence to the limitations section.

4) Line 369: "are reproducible between recording sessions from a single mouse …" Change to "are reproducible across the imaged neuronal population between recording sessions from a single mouse" to clarify that these general activity patterns were observed, but not within the same individual neurons.

Done

5) There seem to be two different definitions applied for the EA subtype of response. For one, they responded to the initial increase in inspired CO2 but did not maintain their activation, and for the other, they respond to 3% CO2 more robustly than 6% CO2. Are these the same?

Yes -the wording has been modified to avoid confusion.

6) Lines 447-448: "Chemosensory responses become less dependent on Phox2B+ neurons of the RTN by 3 months of age (Ramanantsoa et al., 2011) … should add "when those neurons are genetically ablated" or some such qualifier.

Done

7) Line 515: This statement should include a reference to Cleary et al. (PMID: 34013884), in which CO2-inhibited SST-expressing interneurons were recorded in the parafacial (RTN) region. This paper should also be cited with reference to the diversity of neuronal subtypes in that region, many of which may have been sampled in the current GCaMP6 recordings.

Done

8) Page 15, para. 1 and 2 and Figure 8. The calcium signal measurements presented in this paper and summarized in panel C do not directly provide information on neuronal firing patterns and associated synaptic interactions implied in these diagrams. Given the uncertainty, the authors should emphasize that the regional interactions postulated are based on what is generally proposed in the literature, and it is currently unknown if any specific type of neuron that the authors have classified from the calcium signals and represented in the diagram has the connections indicated. Some readers may find these diagrams excessively speculative.

We agree that we have not identified synaptic connections from these neurons and that the interconnections are purely literature based we have reworded the legend and text on pp15 to bring this out more clearly.

9) Line 619: "The neuronal responses to CO2 were more heterogeneous in both the RTN and Raphe than would be expected from the prior literature" should be clarified to state that "The neuronal responses to CO2 were heterogeneous for unidentified neurons in the RTN and for serotonergic neurons in Raphe."

Reworded.